# Liberals and conservatives respond divergently to stereotype portrayals of race and gender
Elizabeth Q. Jiang ✉ & Margaret J. Shih

Representation in the media has become a polarizing issue dividing conservatives and liberals in the U.S. In four experiments ($N = 5125$), we find that stereotype portrayal elicits divergent attitudinal, economic, and behavioral reactions from liberals and conservatives. Notably, these reactions differ when portrayals feature racial minority (Study 1, $n = 958$ & Study 2, $n = 900$) versus white models (Study 3, $n = 783$ & Study 4, $n = 2484$). Our findings demonstrate consistent divergence in responses to stereotype congruent versus incongruent portrayals between liberals and conservatives, although the direction and magnitude of differences vary. Liberals and conservatives display both variability and consistency in their divergent evaluations: liberals endorse portrayals of minority races and of incongruency but withhold this endorsement for solely white models, whereas conservatives typically prefer congruent portrayals, but show an openness towards incongruency when white models are featured. Understanding these dynamics is crucial for navigating the current sociopolitical landscape, especially in contexts where representations of race and gender identities are contentious.

Political polarization has resulted in divisions in the United States[1], particularly on issues surrounding equity, diversity, and inclusion (EDI)[2]. The growth of social justice movements related to race and gender—such as #MeToo, active anti-racism, LGBTQ+ pride, and gender pronoun signifiers—has also spawned a backlash among conservatives[3,4]. For instance, while liberals often prioritize addressing perceived social injustices and advocate for change (i.e., stay woke), conservatives frequently emphasize preserving traditional values and institutions they view as foundational to society (i.e., war on woke)[5]. These liberal-driven movements and their backlash from conservatives reflect an intensifying, negative spiral: growing attention to the way individuals are represented in popular media reinforces beliefs and, in turn, leads to behavioral responses such as boycotts, cancellations, and accusations of cancel culture[6–8]. The liberal versus conservative ideological discord on issues surrounding EDI is especially salient in the consideration of representation and representativeness of marginalized individuals.

Popular mass media portrayals of marginalized groups in the U.S.[9], such as racial minorities and women, have incurred strongly divergent reactions from liberal- and conservative-minded viewers, wherein the former tend to advocate for more diverse representations, while the latter often express concerns, penalize, or boycott representations that may not reflect authentic characterizations or merit-based qualifications. Over the past few years, for instance, food product branding that is in line with stereotypes and film/TV cast lists that are not in line with stereotypes have incited sharp

attention from liberals and conservatives, respectively[10,11]. The same stimulus (i.e., an advertisement, a movie trailer, etc.) may incite strongly divergent actions such as praise and increased support from those who view it as positive progress, as opposed to criticism, penalizing review bombing, and other backlash from those who see it as undermining important cultural traditions or standards. Importantly, these stimuli—the portrayals of racial minorities and women—evoke the culture wars and, specifically, the struggle with EDI in America.

Although understanding psychological processes underlying these polarized viewpoints could help to defuse these palpable real-world tensions, comprehensive research on the intersection of political ideology, EDI, and marginalized group stereotypes is limited. Notable work has been conducted on each component of this intersection. For instance, research on political ideology has investigated the nuances underlying political disagreement, citing intolerance, between-group affect, heated dislike of political out-group members, and distinct cognitions and belief systems as important drivers of political polarization[12–17]. Such polarization contributes to the way conservatives and liberals differentially perceive stereotypes of social identity groups. However, past research at the intersection of political ideology and social identity stereotypes has predominantly focused on how conservatives and liberals differ in their evaluations of *traditional* stereotypes[18–20], showing differences between liberals and conservatives in their preference, endorsement, or perpetuation of commonly-held and pervasive stereotypes. This body of research suggests that liberals may be more implicitly malleable in

Department of Management, University of California, Los Angeles (UCLA), Los Angeles, CA, USA. ✉e-mail: elizabeth.jiang.phd@anderson.ucla.edu

their responses to traditional stereotypes[18], that conservatives are more likely to endorse race and gender-based stereotypes[19], and that liberals and conservatives differ in both explicit evaluations and implicit perceptions of a series of social identity-based stereotypes[20]. Some research has studied reactions to counterstereotypical portrayals of marginalized identities, but these often do not study differences in reactions by political ideology[21–26], thus leaving open the critical question of how political ideology and stereotype portrayal interact.

Very little empirical work has simultaneously studied how political ideology affects reactions to explicit portrayals of marginalized identities that either maintain or diverge from traditional stereotypes, despite the considerable interconnection between political ideology and stereotype portrayal. As such, we seek to provide a thorough examination not only of how responses to stereotypical portrayals of race and gender may or may not differ from reactions to stereotype-incongruent portrayals of the same subjects, but also of how these reactions depend upon the political ideology of the viewer. Our testing of the interactive dynamics of political ideology and stereotype portrayal on attitudes, economic attributions, and downstream behaviors will contribute to a fuller understanding of the relationships between these important and socially contentious topics. Furthermore, we incorporate racial representation as an additional factor in understanding the effects of political ideology and stereotype portrayal on explicit evaluations and perceptions. Specifically, we vary the racial make-up in explicit stereotype congruent and explicit stereotype incongruent portrayals to assess the extent to which conservatives and liberals diverge in their direct evaluations of such representations.

Further, updated research on explicitly divergent or stereotype-incongruent portrayals of marginalized identities that reflect the changing portrayals in current films, brand advertisements, and mainstream media is necessary in advancing our understanding of stereotypes over time. The theoretical and empirical attention given to stereotypes has historically focused on explicit stereotypes and extended through contemporary frameworks[27,28]. As research developed, the field expanded significantly with increased attention to implicit social cognition[29,30], focusing on implicit bias measurement and other measures beyond explicit self-report[31]. Recent research has examined the relationship between these explicit and implicit domains[20,32–35]. In our work, we argue it is important to directly study explicitly stereotype-congruent and stereotype-incongruent portrayals, given the greater media coverage of explicit representations and the heightened attention that the larger society is taking to how marginalized groups are portrayed. Indeed, in wider American society, explicit biases and preferences are frequently expressed, oftentimes to staunch or extreme ends. Directly studying responses to explicit stereotype portrayals from opposing political ideologies reflects the current social moment. Moreover, focusing on explicit stereotypes and explicit reports of reactions to stereotype portrayal may reveal a newfound comfort or willingness among participants to openly express their potentially preferential or biased beliefs, in turn updating the research on explicit stereotypes. Thus, the present work investigates both explicitly stereotype-congruent and explicitly stereotype-incongruent portrayals of varying race and gender identities.

Here we investigate the simultaneous forces of political ideology, stereotype portrayal, and racial representation on attitudinal, economic, and behavioral outcomes. We test whether conservatives and liberals have similar or divergent judgments of stimuli that manipulate stereotype portrayal. To evaluate this, we conducted four preregistered experiments ($N = 5125$) with U.S. citizens in which we operationalized stereotype portrayal as either stereotype congruent (i.e., depicting commonly-held stereotypes) or stereotype incongruent (i.e., absence of commonly-held stereotypes, including depiction of divergence from commonly-held stereotypes). In Studies 1 & 2, we present experimental stimuli that feature Black and Asian racial minorities; in Studies 3 & 4, we present experimental stimuli that include white models. The procedure was the same in each of the studies. First, participants were randomly assigned to either the stereotype-congruent condition or the

stereotype-incongruent condition. They viewed pre-tested custom advertisements according to their condition assignment. As such, participants would view advertisements portraying Black and Asian racial minorities and/or women in contexts associated with their assigned stereotype portrayal condition (e.g., custom athletic gear modeled by Black men (congruent) versus modeled by Asian women (incongruent); software/IT services modeled by Asian men (congruent) versus Black women (incongruent); tutoring services modeled by white women (congruent) versus white men (incongruent)) (see Table 1 for experimental stimuli; see Supplementary Materials Appendix B for more details on pre-testing of the advertisement stimuli). Then, after viewing the stimuli, participants answered questions along our dependent measures: attitudinal perceptions, economic perceptions, and behavioral outcomes. Finally, participants responded to demographic questions measuring their political ideology, race, gender, age, income, and educational attainment.

## Methods
We conducted one Pilot Study experiment, one Preliminary Study experiment, and four main preregistered quantitative experiments (Studies 1–4) using online samples of liberal and conservative participants. Preregistration documents for all studies can be found at our OSF project repository, which is publicly accessible at the following link: https://osf.io/cxpgk/.

Please see the Supplementary Materials Appendix A: Study Design Details and Sample Information for detailed methods for Pilot Study and Preliminary Study.

Below, we provide methodological details for our main experiments, Studies 1–4.

### Research sample
The research sample consists of U.S. adults who are strongly and consistently liberal or conservative in their self-reported political ideology. In recruiting and obtaining responses from our sample, we have complied with all relevant ethical regulations. This research protocol was approved by UCLA's Institutional Review Board. All participants provided informed consent to participate, and no deception was utilized in this study.

### Sampling strategy
We used convenience sampling via online survey panels (TurkPrime, Prolific, and CloudResearch Connect), given their ease and ability in reaching participants by ideology (and, in particular, difficult-to-reach Conservative participants). We measured distinct samples across four separate studies consisting of unique participants (i.e., no participants had taken a previous study). Participants were paid $2 to $5 for their participation.

A note on recruiting samples of politically liberal and politically conservative participants using online panels: The online survey panels feature pre-screening tools based on participants' demographics. We used the panels' pre-screening questions, capturing participants' political ideology, in an attempt to recruit balanced samples of liberals and conservatives. We do not use the online panels' pre-screening tools alone, which may be outdated, to identify participants' political ideology in our analyses; to identify participants' political ideology, we use our demographic questions, which are asked after the main dependent variable measures and at the end of each survey. Our online surveys were advertised in terms of general perceptions, attitudes, behavior, and decision-making, and in no way mentioned political ideology, political beliefs, stereotypes, race, or gender. Additionally, our usage of the pre-screening tool provided by the online survey panel is not shared with participants. We expect no priming effect of political ideology on our results.

### Data collection
Data was collected online using online survey panels, TurkPrime, Prolific, or CloudResearch Connect. Participants were online workers who completed an online experimental survey, which was hosted on Qualtrics.

Regarding sample size calculations:

**Table 1 | Experimental stimuli manipulating stereotype portrayal (stereotype congruent vs. stereotype incongruent), by study**

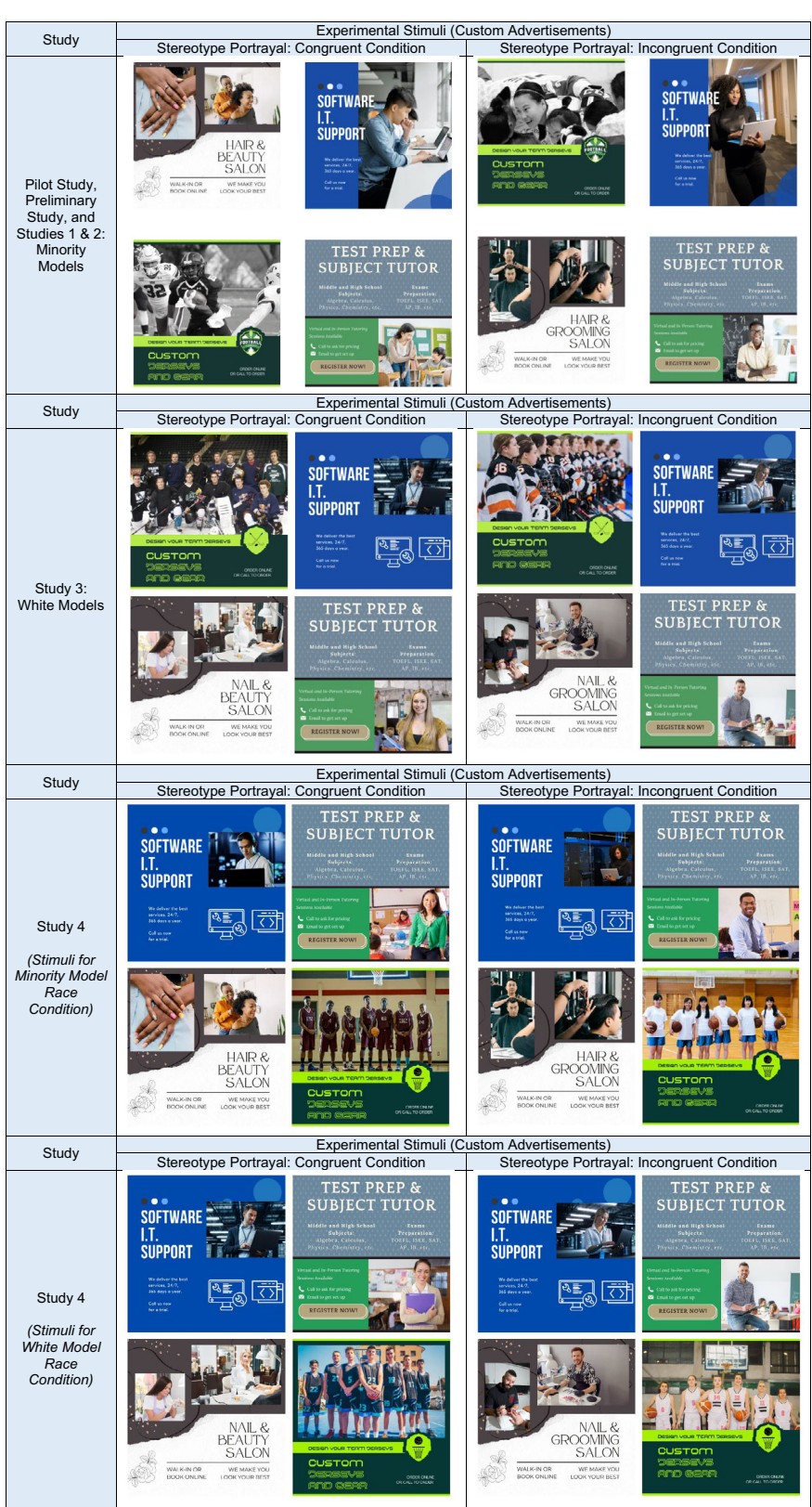

The experimental stimuli were created by the study team and pre-tested for realism and stereotype alignment. Please see the Supplementary Materials Appendix B for complete details on the pre-test methods, analyses, and results.

For Studies 1–3, we did not perform a sample size calculation and instead preregistered large numbers of liberal and conservative participants, given our dual desires to recruit a balanced liberal and conservative sample and achieve high statistical power.

Our preregistration for Study 1a (February 24, 2023) notes: "We will seek an increase in statistical power by recruiting a larger $N = 800$ sample size. Additionally, we will seek to evenly balance political ideology ($n = 400$ politically liberal and $n = 400$ politically conservative). Due to attention check and failure-to-finish rate of about 15%, we will recruit $N = 920$ to get to clean $N = 800$," and our preregistration for Study 1b (May 5, 2023) notes: "We will seek an increase in statistical power by recruiting a larger $N = 200$ sample size. Additionally, we will seek to evenly balance political ideology ($n = 100$ politically liberal and $n = 100$ politically conservative). Due to attention check and failure-to-finish rates, we will recruit $N = 300$ to get to clean $N = 200$…. We will merge Study [1]b's clean data with Study [1] (a)'s clean data in order to increase statistical power. This will be similar to a mini meta-analysis (e.g., Goh et al., 2016)."

Our preregistration for Study 2 (November 7, 2023) notes: "We will seek $n = 525$ liberal and $n = 525$ conservative participants. First, we are seeking a sample balanced on political ideology. Second, we are seeking to obtain as many participants as is feasible and realistic using Prolific, which has a smaller and less active conservative population."

Our preregistration for Study 3 (January 18, 2024) notes: "We will seek strong statistical power by recruiting a large sample size. Additionally, we will seek to evenly balance political ideology (50% politically liberal and 50% politically conservative). Due to attention check and failure-to-finish rates, as well as the difficulty of recruiting conservative participants, we will recruit as large a sample size as we can, aiming for at least $n = 250$ conservatives." This was amended in a subsequent preregistration posted to OSF (January 30, 2024), stating: "We will seek strong statistical power by recruiting a large $N = 900$ sample size. Additionally, we will seek to evenly balance political ideology ($n = 450$ politically liberal and $n = 450$ politically conservative). Due to attention check and failure-to-finish rates, we will recruit $N = 1000$ to get to clean $N = 900$."

For Study 4, which preregistered a three-way factorial experimental design, we calculated the sample size required to detect an exploratory three-way interaction effect. Our preregistration for Study 4 (February 27, 2025) notes: "We will aim for $2500 + 300$ participants, as we seek a sample that will be statistically powerful enough to detect an exploratory three-way interaction (2500; this sample size was calculated based on the average effect size $f$ from previous studies ($f = 0.1105$) with 80% statistical power and alpha level $a = 0.05$) and we are accounting for a 12% exclusion rate of participants (300). Therefore, we will aim to recruit 2800 participants, balanced between liberals (1400) and conservatives (1400). We will run two concurrent and separate surveys using CloudResearch Connect's pre-screeners for liberals and conservatives. We will monitor progress across both surveys such that participants accrue at roughly the same rate and such that we can close both surveys at the same time. We may run a second wave of surveys using higher incentives or using other platforms in case we are not able to reach our balanced sample in the first set of CloudResearch Connect surveys."

Preregistration documents for all studies can be found at our OSF project repository, which is publicly accessible at the following link: https://osf.io/cxpgk/.

## Timing

Please note for Studies 1–4, we launched concurrently running surveys that targeted liberals (Libs) and conservatives (Cons) using the online survey panel pre-screeners that enabled us to directly seek specific political ideologies. Analysis was done with the merged study data (i.e., Libs and Cons combined).

Timing for each study is as follows:

Study 1: Study 1 is a mini-meta study of Study 1a and Study 1b; we merged Study 1a (March 2023) and Study 1b (May/June 2023) data in order to increase our sample size and statistical power. We preregistered the additional recruitment of participants for Study 1b and preregistered our mini-meta analysis approach. Study 1a: March 6, 2023—March 9, 2023 (Libs) and March 6, 2023—March 20, 2023 (Cons). Study 1b: May 9, 2023—May 15, 2023 (Libs) and May 9, 2023—June 5, 2023 (Cons). In general, there are fewer active participants who are politically conservative on the online panels, and so we allowed the Cons surveys to run longer in order to capture as many conservative participants as possible.

Study 2: October 18, 2023—November 1, 2023 (Libs) and a similar period of October 18, 2023—November 18, 2023 (Cons).

Study 3: January 26, 2024—February 1, 2024 (Libs) and the same period of January 26, 2024—February 1, 2024 (Cons).

Study 4: February 28, 2025—March 19, 2025 (Libs) and the same period of February 28, 2025—March 19, 2025 (Cons).

## Data exclusions

Study 1: As preregistered, we excluded data from participants who did any of the following: did not complete the full survey including the final question (more details below in the Non-participation section, $n = 163$), were duplicate respondents ($n = 7$), or who did not respond correctly to attention check questions ($n = 13$). We also excluded data from participants who spent less than 300 seconds (5 minutes) on the entire survey ($n = 22$), which we did not preregister but view as a best practice for attempting to, firstly, exclude bots impersonating real human participants and, secondly, ensure higher quality participant data. In total, this led to excluding data from $n = 205$ participants. Additionally, we excluded from analysis data from individuals who were not consistently and strongly politically self-identified (i.e., excluded participants who were politically moderate or politically independent, $n = 181$; more details in our Main Manuscript and Supplementary Materials). This yielded a final sample of $n = 958$ liberal and conservative participants.

Study 2: As preregistered, we excluded data from participants who did any of the following: did not complete the full survey including the final question (more details below in the Non-participation section, $n = 277$), were duplicate respondents ($n = 0$), or who did not respond correctly to attention check questions ($n = 4$). We also excluded data from participants who spent less than 300 seconds (5 minutes) on the entire survey ($n = 4$), which we did not preregister but did for the same reasons as noted in Study 1, and for consistency between studies. In total, this led to excluding data from $n = 285$ participants. Additionally, we excluded from analysis data from individuals who were not consistently and strongly politically self-identified (i.e., excluded participants who were politically moderate or politically independent, $n = 144$; more details in our Main Manuscript and Supplementary Materials). This yielded a final sample of $n = 900$ liberal and conservative participants.

Study 3: As preregistered, we excluded data from participants who did any of the following: did not complete the full survey including the final question (more details below in the Non-participation section, $n = 147$), were duplicate respondents ($n = 0$), or who did not respond correctly to attention check questions ($n = 7$). We also excluded data from participants who spent <300 seconds (5 minutes) on the entire survey ($n = 3$), which we did not preregister but did for the same reasons as noted in Study 1, and for consistency between studies. And, we removed data from participants whom we used to pilot test this survey ($n = 45$), which we mentioned in our preregistration; we excluded this data in the best practice of maintaining similar timing for the main sample. Further, we removed data from participants who responded affirmatively to a question on whether they had taken similar survey on this topic before ($n = 42$), as an additional check of removing those who were familiar with the survey; we did not ask this question in previous studies, and instead relied on the online survey panel's (i.e., TurkPrime's or Prolific's) exclusion tool, which excludes participants who had taken previous studies from seeing or entering the present study. In total, this led to excluding data from $n = 244$ participants. Additionally, we excluded from analysis data from individuals who were not consistently and

**Table 2 | Study sample sizes by participant ideology and experimental conditions**

| Study | $N$, total clean study sample (post-exclusions) | $n$, sample by participant political ideology (PI) | $n$, sample by stereotype portrayal (SP) conditions | $n$, sample by PI x SP | $n$, sample by model race (MR) conditions | $n$, sample by PI x MR | $n$, sample by PI x SP x MR |
|---|---|---|---|---|---|---|---|
| Pilot study | $N = 220$ | N/A | $n_{Cong} = 115$ <br> $n_{Incong} = 105$ | N/A | N/A | N/A | N/A |
| Preliminary study | $N = 196$ | $n_{Cons} = 61$ <br> $n_{Libs} = 135$ | $n_{Cong} = 100$ <br> $n_{Incong} = 96$ | $n_{Cons*Cong} = 29$ <br> $n_{Cons*Incong} = 32$ <br> $n_{Libs*Cong} = 71$ <br> $n_{Libs*Incong} = 64$ | N/A | N/A | N/A |
| Study 1 | $N = 958$ | $n_{Cons} = 423$ <br> $n_{Libs} = 535$ | $n_{Cong} = 476$ <br> $n_{Incong} = 482$ | $n_{Cons*Cong} = 204$ <br> $n_{Cons*Incong} = 219$ <br> $n_{Libs*Cong} = 272$ <br> $n_{Libs*Incong} = 263$ | N/A | N/A | N/A |
| Study 2 | $N = 900$ | $n_{Cons} = 351$ <br> $n_{Libs} = 549$ | $n_{Cong} = 438$ <br> $n_{Incong} = 462$ | $n_{Cons*Cong} = 171$ <br> $n_{Cons*Incong} = 180$ <br> $n_{Libs*Cong} = 267$ <br> $n_{Libs*Incong} = 282$ | N/A | N/A | N/A |
| Study 3 | $N = 783$ | $n_{Cons} = 377$ <br> $n_{Libs} = 406$ | $n_{Cong} = 380$ <br> $n_{Incong} = 403$ | $n_{Cons*Cong} = 185$ <br> $n_{Cons*Incong} = 192$ <br> $n_{Libs*Cong} = 195$ <br> $n_{Libs*Incong} = 211$ | N/A | N/A | N/A |
| Study 4 | $N = 2484$ | $n_{Cons} = 1161$ <br> $n_{Libs} = 1323$ | $n_{Cong} = 1256$ <br> $n_{Incong} = 1228$ | $n_{Cons*Cong} = 582$ <br> $n_{Cons*Incong} = 579$ <br> $n_{Libs*Cong} = 674$ <br> $n_{Libs*Incong} = 649$ | $n_{Minor} = 1255$ <br> $n_{white} = 1229$ | $n_{Cons*Minor} = 579$ <br> $n_{Cons*White} = 582$ <br> $n_{Libs*Minor} = 676$ <br> $n_{Libs*White} = 647$ | $n_{Cons*Cong*Minor} = 279$ <br> $n_{Cons*Cong*White} = 303$ <br> $n_{Cons*Incong*Minor} = 300$ <br> $n_{Cons*Incong*White} = 279$ <br> $n_{Libs*Cong*Minor} = 348$ <br> $n_{Libs*Cong*White} = 326$ <br> $n_{Libs*Incong*Minor} = 328$ <br> $n_{Libs*Incong*White} = 321$ |

For pilot and preliminary studies, unlike for studies 1–4, we did not seek a sample balanced by political ideology.

strongly politically self-identified (i.e., excluded participants who were politically moderate or politically independent, $n = 108$; more details in our Main Manuscript and Supplementary Material). This yielded a final sample of $n = 783$ liberal and conservative participants.

Study 4: As preregistered, we excluded data from participants who did any of the following: did not complete the full survey including the final question (more details below in the Non-participation section, $n = 407$), who were not strong in their political ideology ($n = 397$), or who did not respond correctly to attention check questions ($n = 22$). We also checked participant data for participants who spent <300 seconds (5 minutes) on the entire survey ($n = 0$) and for participants who were duplicate respondents ($n = 1$), which we did not preregister but used as exclusion criteria for the same reasons as noted in previous studies and for consistency between studies. Further, we excluded data from participants who responded affirmatively to a question on whether they had taken a similar survey on this topic before ($n = 34$) as an additional check to remove participants who were familiar with the survey and may have taken a survey for a previous study hosted on another participant recruitment platform. In total, this led to excluding data from $n = 861$ participants. This yielded a final sample of $n = 2484$ liberal and conservative participants.

### Non-participation

Study 1: 163 participants did not complete the full survey including the final question. We received raw data from $n = 1344$ participants, yielding a 87.9% completion rate.

Study 2: 277 participants did not complete the full survey, including the final question. We received raw data from $n = 1329$ participants, yielding a 79.2% completion rate.

Study 3: 147 participants did not complete the full survey including the final question. We received raw data from $n = 1135$ participants, yielding a 87.0% completion rate.

Study 4: 407 participants did not complete the full survey including the final question. We received raw data from $n = 3355$ participants, yielding a 87.9% completion rate.

### Randomization

In Studies 1–3, participants were randomly assigned to one of two Stereotype Portrayal conditions (stereotype congruent portrayal OR stereotype incongruent portrayal). We set the randomization to be evenly populated, such that 50% of participants are assigned to the congruent condition and 50% are assigned to the incongruent condition. In Study 4, which used a 2 (participant political ideology) × 2 (stereotype portrayal) × 2 (model race) three-factor design, participants were randomly assigned to one of two Stereotype Portrayal conditions at a 50–50% even distribution and to one of two Model Race conditions (minority OR white models) at a 50–50% even distribution.

### Stimuli

The custom advertisements used in these experiments were created by the study team and pre-tested for realism and stereotype alignment. Please see the Supplementary Materials Appendix B subsections "Pilot Study: Testing Advertisement Stimuli" and "Study 4 Supplementary Results" for complete details on the pre-test methods, analyses, and results.

### Final sample sizes

Study 1 participants ($n = 1139$ U.S. adults; $n_{liberal} = 535$, $n_{conservative} = 423$, $n_{moderate/independent} = 181$) were recruited from TurkPrime (https://www.cloudresearch.com/).

Study 2 participants ($n = 1044$ U.S. adults; $n_{liberal} = 549$, $n_{conservative} = 351$, $n_{moderate/independent} = 144$) were recruited from Prolific (https://app.prolific.com/).

Study 3 participants ($n = 891$ U.S. adults; $n_{liberal} = 406$, $n_{conservative} = 377$, $n_{moderate/independent} = 108$) were recruited from TurkPrime (https://www.cloudresearch.com/).

Study 4 participants ($n = 2881$ U.S. adults; $n_{liberal} = 1323$, $n_{conservative} = 1161$, $n_{moderate/independent} = 397$) were recruited from CloudResearch Connect (https://connect.cloudresearch.com).

Please see Table 2 for a summary table of all studies' sample size by condition. We focus our sample reporting and main analyses on liberal and

## Table 3 | Study 1: participant race and gender

| Participant gender | Participant race | # | % of sample |
|---|---|---|---|
| Men | White | 263 | 27.5% |
| | Black | 28 | 2.9% |
| | Asian | 17 | 1.8% |
| | Hispanic | 23 | 2.4% |
| | Other | 6 | 0.6% |
| | Missing | 1 | 0.1% |
| Women | White | 512 | 53.4% |
| | Black | 50 | 5.2% |
| | Asian | 13 | 1.4% |
| | Hispanic | 22 | 2.3% |
| | Native Hawaiian or Pacific Islander | 2 | 0.2% |
| | Other | 10 | 1.0% |
| Other | White | 9 | 0.9% |
| | Hispanic | 2 | 0.2% |

## Table 4 | Study 2: participant race and gender

| Participant gender | Participant race | # | % of sample |
|---|---|---|---|
| Men | White | 295 | 32.8% |
| | Black | 44 | 4.9% |
| | Asian | 25 | 2.8% |
| | Hispanic | 33 | 3.7% |
| | American Indian or Alaska Native | 1 | 0.1% |
| | Native Hawaiian or Pacific Islander | 1 | 0.1% |
| | Other | 7 | 0.8% |
| Women | White | 402 | 44.7% |
| | Black | 33 | 3.7% |
| | Asian | 12 | 1.3% |
| | Hispanic | 24 | 2.7% |
| | American Indian or Alaska Native | 2 | 0.2% |
| | Other | 8 | 0.9% |
| Other | White | 7 | 0.8% |
| | Black | 1 | 0.1% |
| | Asian | 1 | 0.1% |
| | Hispanic | 1 | 0.1% |
| | Other | 3 | 0.3% |

## Table 5 | Study 3: participant race and gender

| Participant gender | Participant race | # | % of sample |
|---|---|---|---|
| Men | White | 284 | 36.3% |
| | Black | 22 | 2.8% |
| | Asian | 22 | 2.8% |
| | Hispanic | 31 | 4.0% |
| | American Indian or Alaska Native | 1 | 0.1% |
| | Other | 3 | 0.4% |
| Women | White | 329 | 42.0% |
| | Black | 35 | 4.5% |
| | Asian | 21 | 2.7% |
| | Hispanic | 18 | 2.3% |
| | American Indian or Alaska Native | 2 | 0.3% |
| | Native Hawaiian or Pacific Islander | 1 | 0.1% |
| | Other | 6 | 0.8% |
| Other | White | 4 | 0.5% |
| | Asian | 1 | 0.1% |
| | Hispanic | 2 | 0.3% |
| | Other | 1 | 0.1% |

## Table 6 | Study 4: participant race and gender

| Participant gender | Participant race | # | % of sample |
|---|---|---|---|
| Men | White | 758 | 30.5% |
| | Black | 90 | 3.6% |
| | Asian | 74 | 3.0% |
| | Hispanic | 75 | 3.0% |
| | American Indian or Alaska Native | 1 | 0.0% |
| | Native Hawaiian or Pacific Islander | 3 | 0.1% |
| | Other | 13 | 0.5% |
| Women | White | 1143 | 46.0% |
| | Black | 114 | 4.6% |
| | Asian | 75 | 3.0% |
| | Hispanic | 84 | 3.4% |
| | American Indian or Alaska Native | 3 | 0.1% |
| | Native Hawaiian or Pacific Islander | 1 | 0.0% |
| | Other | 21 | 0.8% |
| Other | White | 15 | 0.6% |
| | Black | 3 | 0.1% |
| | Asian | 1 | 0.0% |
| | Hispanic | 4 | 0.2% |
| | Other | 6 | 0.2% |

conservative participants; supplementary results including politically moderate/independent for each study are available in the Supplementary Materials Appendix B: Supplementary Results, under subsections "Study [1, 2, 3, 4] Supplementary Results."

### Sample demographics
Participants provided their self-identification for sex/gender and race/ethnicity. Please see Tables 3–6 for participant race and gender breakdowns for each study.

### Composite index measures
We created separate composite index measures for participant political ideology, attitudinal perceptions, and economic perceptions. Please see the Supplementary Materials Appendix B subsection "Index Measures" for complete details on all index measures.

Below, we report preregistered and exploratory analyses of primary interest. Additional preregistered analyses of secondary interest are reported in the Supplementary Materials Appendix B: Supplementary Results.

### Results
All statistics reported in the result section were conducted using preregistered between-groups factorial analyses of variance. We also conducted follow-up and exploratory linear regression models that control for participants' age, race, gender, income, and educational attainment. In probing

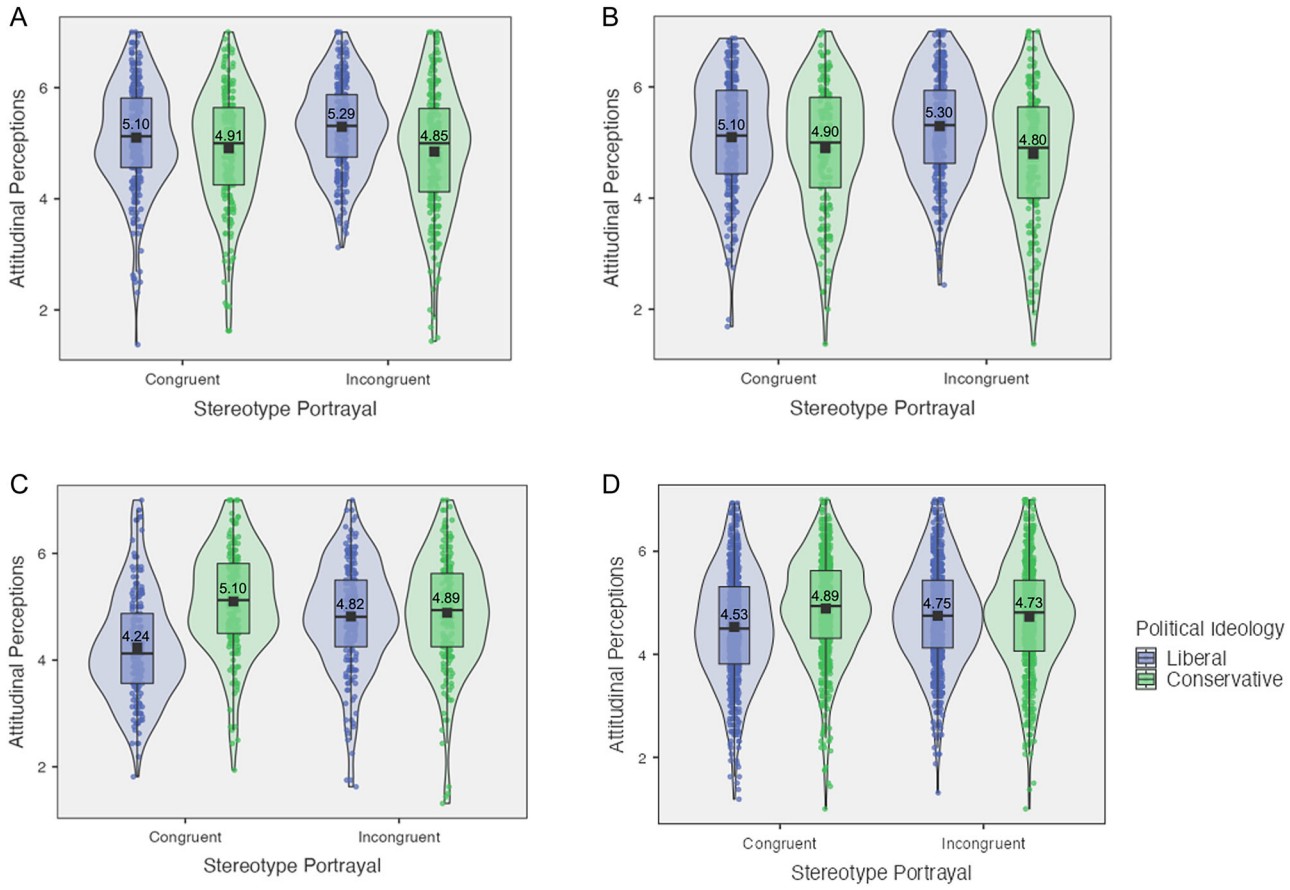

**Fig. 1 | Violin plots for attitudinal perceptions by stereotype portrayal and political ideology, studies 1–4.** Violin plot for Study 1 is contained in (**A**); Study 2 is contained in (**B**); Study 3 is contained in (**C**); Study 4 is contained in (**D**). These violin plots map the mean (i.e., black square point), raw data (i.e., circular points), and box-plot elements (i.e., center line, median; box limits, upper and lower quartiles; whiskers, 1.5× interquartile range). Figure legend: Blue = Liberals, Green = Conservatives.

interaction effects to better understand pairwise differences, we conduct follow-up simple effects analysis and *t* tests when appropriate.

We conducted four preregistered experiments, Studies 1–4. Prior to Study 1, we conducted a stimulus pre-test Pilot Study and a preregistered Preliminary Study. Study details and results for the Pilot Study and Preliminary Study are reported in the Supplementary Materials Appendix A (study and sample details) and Appendix B (results reporting). For all studies, we leave secondary and exploratory test results in the Supplementary Materials Appendix B subsections "Study [1, 2, 3, 4] Supplementary Results." These exploratory tests include linear regression models on our outcome measures that include the insubstantial politically moderate participants who completed our studies and exploratory mediation analyses that are not central to our central research question or hypotheses.

We presently organize our results by outcome variable for our first two studies, Studies 1 & 2.

## Attitudinal outcomes: perceptions of liking, comfort, and persuasiveness

We conducted experiments with politically conservative (cons) and liberal (libs) U.S. citizens who were randomly assigned to stereotype-congruent (cong) vs. incongruent (incong) conditions that depicted Black and Asian racial minorities and that manipulated related race and gender stereotypes in workplace advertisements. We asked participants to rate their liking, comfort with, and persuasiveness of the advertisement, which we collapsed into an index measure of attitudinal perceptions (see Supplementary Materials Appendix B subsection "Index Measures" for more details on individual items and index outcomes).

In Study 1, we conducted two preregistered waves of sample collection and used a mini meta-analysis approach in merging the data[36] in order to obtain a large sample that balanced conservative and liberal participants. Running a confirmatory factorial ANOVA, we found a significant interaction effect of stereotype portrayal and political ideology on attitudes ($M_{\text{cons*cong}} = 4.91$, $SD_{\text{cons*cong}} = 1.05$, $M_{\text{cons*incong}} = 4.85$, $SD_{\text{cons*incong}} = 1.12$, $M_{\text{libs*cong}} = 5.10$, $SD_{\text{libs*cong}} = 0.95$, $M_{\text{libs*incong}} = 5.29$, $SD_{\text{libs*incong}} = 0.80$; $F_{(1,954)} = 3.98$, $p = 0.046$, $\eta^2_p < 0.01$, 95% CI = [0.00, 1.00]) (see Fig. 1A). This result held in an exploratory linear regression ($b = 0.25$, $SE = 0.13$, $p = 0.046$, $\eta^2_p < 0.01$, 95% CI = [0.00, 1.00]). In an exploratory linear regression holding participants' age, race, gender, income, and educational attainment constant, we find similar directional results, though this result was not statistically significant ($b = 0.22$, $SE = 0.12$, $p = 0.081$, $\eta^2_p < 0.01$, 95% CI = [0.00, 1.00]). Exploratory simple effects analyses showed a significant difference among liberals evaluating congruent versus incongruent portrayals ($t_{(954)} = 2.29$, $p = 0.022$, Cohen's $d = 0.15$, 95% CI = [0.02, 0.28]), a significant difference between liberals and conservatives evaluating congruent portrayals ($t_{(954)} = 2.13$, $p = 0.033$, Cohen's $d = 0.14$, 95% CI = [0.01, 0.26]), and a significant difference between liberals and conservatives evaluating incongruent portrayals ($t_{(954)} = 5.00$, $p < 0.001$, Cohen's $d = 0.32$, 95% CI = [0.20, 0.45]), such that liberals preferred incongruent portrayals significantly more than congruent portrayals, and that liberals preferred both stereotype congruent and stereotype incongruent portrayals significantly more than conservatives. As we conducted two preregistered waves of participant recruitment, we also calculate and report $p_{\text{augmented}}$ for Study 1's results in the Supplementary Materials Appendix B subsection "Study 1 Supplementary Results."[37]

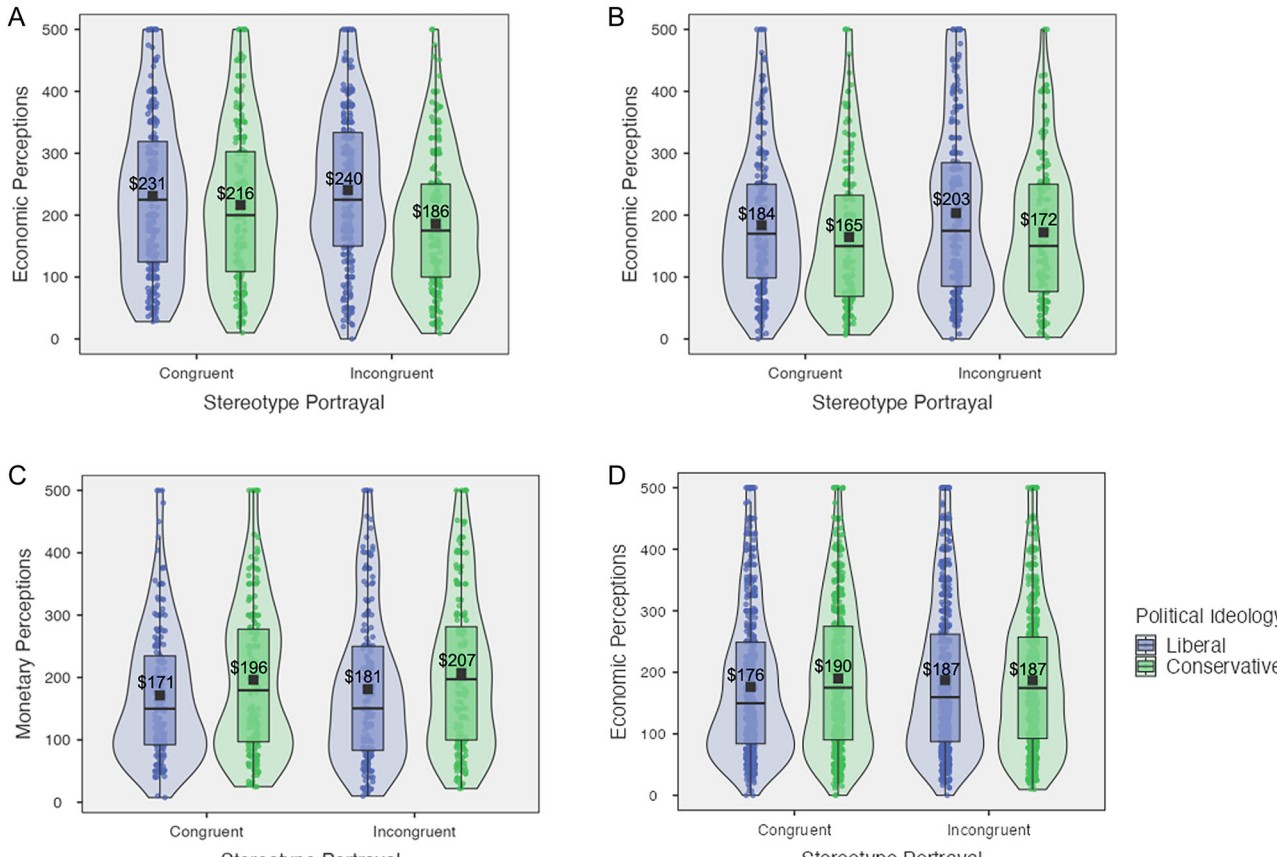

**Fig. 2 | Violin plots for economic perceptions by stereotype portrayal and political ideology, studies 1–4.** Violin plot for Study 1 is contained in (**A**); Study 2 is contained in (**B**); Study 3 is contained in (**C**); Study 4 is contained in (**D**). These plots map the mean (i.e., black square point), raw data (i.e., circular points), and box-plot elements (i.e., center line, median; box limits, upper and lower quartiles; whiskers, 1.5× interquartile range). Figure legend: Blue = Liberals, Green = Conservatives.

Next, in Study 2, we found the same pattern of significant interaction results ($M_{\text{cons*cong}} = 4.90$, $SD_{\text{cons*cong}} = 1.12$, $M_{\text{cons*incong}} = 4.80$, $SD_{\text{cons*incong}} = 1.16$, $M_{\text{libs*cong}} = 5.10$, $SD_{\text{libs*cong}} = 1.01$, $M_{\text{libs*incong}} = 5.30$, $SD_{\text{libs*incong}} = 0.92$; $F_{(1,896)} = 4.62$, $p = 0.032$, $\eta^2_p < 0.01$, 95% CI = [0.00, 1.00]) (see Fig. 1B). These results held in a linear regression with and without holding constant participants' age, race, gender, income, and educational attainment ($b = 0.30$, $SE = 0.14$, $p = 0.034$, $\eta^2_p < 0.01$, 95% CI = [0.00, 1.00]; $b = 0.30$, $SE = 0.14$, $p = 0.032$, $\eta^2_p < 0.01$, 95% CI = [0.00, 1.00]). Simple effects analyses showed a significant difference among liberals evaluating congruent versus incongruent portrayals ($t_{(896)} = 2.24$, $p = 0.025$, Cohen's $d = 0.15$, 95% CI = [0.02, 0.28]) and between liberals and conservatives evaluating incongruent portrayals ($t_{(896)} = 5.06$, $p < 0.001$, $d = 0.34$, 95% CI = [0.21, 0.47]), revealing again that liberals favored stereotype incongruent portrayals significantly more than conservatives. New in Study 2 is the finding that liberals seem to like all portrayals more than conservatives, which we seek to better understand in Studies 3 & 4.

Across Studies 1 & 2, we found interaction effects of political ideology and stereotype portrayal on attitudinal measures. We find consistent divergence in reactions to stereotype portrayals between conservatives and liberals, showing that participants' attitudinal preferences depended on both their political orientation and the way the advertisements were portrayed. This was specifically apparent in the incongruent stereotype portrayal conditions, wherein conservatives rated incongruent portrayals significantly lower than liberals, who showed higher attitudes towards incongruent portrayals.

Notably, our findings in Studies 1 & 2 show that liberals have higher attitudinal ratings than conservatives, regardless of stereotype portrayal. We did not hypothesize or preregister this underlying main effect of political ideology on attitudinal perceptions, but suspect that it is driven by the stimuli. The advertisements featured in Studies 1 & 2 feature only racial

minorities and no white models. Participants may be reacting to this in different ways. Upon seeing diverse racial minority models, liberals may be amplifying their endorsement of minority representation. Alternatively (or additionally), upon seeing only racial minority models, conservatives may be expressing their backlash towards the exclusion of white models. We follow up on this finding in Study 3, in which we employ advertisements that feature only white models and manipulate gender-based stereotypes, and in Study 4, in which we employ advertisements that manipulate model race and manipulate race and gender stereotypes.

**Economic outcomes: perceptions of monetary value**

We also included measures of economic perceptions of value (i.e., willingness to pay for product/service and predicted monetary worth to others) (see Supplementary Materials Appendix B subsection "Index Measures" for more details on individual items and index outcomes).

In Study 1, we found a significant interaction effect along economic perceptions ($M_{\text{cons*cong}} = \$216$, $SD_{\text{cons*cong}} = \$127$, $M_{\text{cons*incong}} = \$186$, $SD_{\text{cons*incong}} = \$107$, $M_{\text{libs*cong}} = \$231$, $SD_{\text{libs*cong}} = \$128$, $M_{\text{libs*incong}} = \$240$, $SD_{\text{libs*incong}} = \$124$; $F_{(1,954)} = 6.23$, $p = 0.013$, $\eta^2_p < 0.01$, 95% CI = [0.00, 1.00]) (see Fig. 2A). This finding was robust in a linear regression ($b = 39.72$, $SE = 15.91$, $p = 0.013$, $\eta^2_p < 0.01$, 95% CI = [0.00, 1.00]), and was directionally similar in a linear regression controlling for participants' age, race, gender, income, and educational attainment, though this result did not show evidence of statistical significance ($b = 34.88$, $SE = 15.70$, $p = 0.027$, $\eta^2_p < 0.01$, 95% CI = [0.00, 1.00]). Exploratory simple effects analyses showed a significant difference among conservatives evaluating congruent versus incongruent portrayals ($t_{(954)} = 2.55$, $p = 0.011$, Cohen's $d = 0.17$, 95% CI = [0.04, 0.29]), and a significant difference between liberal and conservative responses to incongruent portrayals ($t_{(954)} = 4.88$, $p < 0.001$,

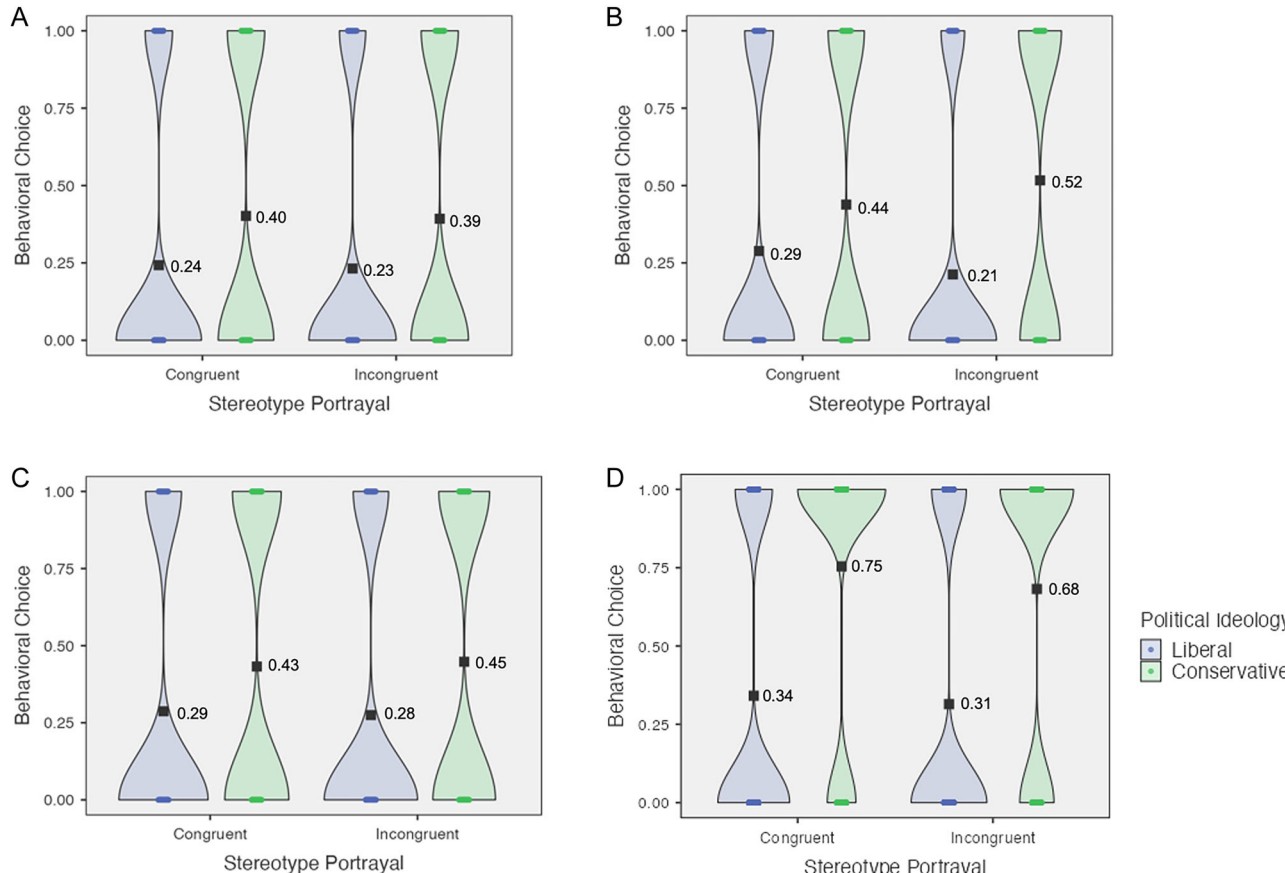

**Fig. 3 | Violin plots for behavioral choices by stereotype portrayal and political ideology, studies 1–4.** Violin plot for Study 1 is contained in (**A**); Study 2 is contained in (**B**); Study 3 is contained in (**C**); Study 4 is contained in (**D**). These plots map the mean (i.e., black square point) and raw data (i.e., circular points). *Y* axis Behavioral Choice: 0 = Stereotype Incongruent Candidate Hired, 1 = Stereotype Congruent Candidate Hired. Figure legend: Blue = Liberals, Green = Conservatives.

Cohen's $d = 0.32$, 95% CI = [0.19, 0.44]). Thus, participants' economic perceptions—similar to their attitudinal perceptions—depended on the interaction between their political orientation and the advertisements' portrayal, with a particularly consistent divergence between liberals and conservatives evaluating incongruent stereotype portrayals.

In Study 2, we found no evidence for a statistically significant interaction effect of political ideology and stereotype portrayal on economic perceptions ($M_{cons*cong} = \$165$, $SD_{cons*cong} = \$115$, $M_{cons*incong} = \$172$, $SD_{cons*incong} = \$117$, $M_{libs*cong} = \$184$, $SD_{libs*cong} = \$114$, $M_{libs*incong} = \$203$, $SD_{libs*incong} = \$139$; $F_{(1, 896)} = 0.53$, $p = 0.467$, $\eta^2_p < 0.01$, 95% CI = [0.00, 1.00]) (see Fig. 2B). Probing this result, we ran an independent samples *t* test and found a significant main effect of political ideology on economic perceptions ($M_{cons} = \$169$, $SD_{cons} = \$116$, $M_{libs} = \$194$, $SD_{libs} = \$128$; $t_{(898)} = 3.00$, $p = 0.003$, Cohen's $d = 0.20$, 95% CI = [0.07, 0.34]), revealing that liberals spend significantly more than conservatives across all stereotype portrayal conditions. While Study 2 failed to show a significant interaction effect, perhaps due to timing or sample constraints, the significant main effect of political ideology on economic perceptions is worth noting. We consider the pattern of liberals attributing significantly higher value to all advertised goods/services than conservatives, which is in line with the main effect of political ideology on attitudinal outcomes previously reported, to be interesting and worth following up on in our Studies 3 & 4 that serve as paradigm replications.

**Behavioral outcomes: downstream hiring selection**

For our third and final outcome measure, we presented participants with a software hiring context and asked participants which job candidate they would hire, specifying that the performance of their candidate would reflect on their own performance (see Supplementary Materials Appendix B

subsection "Vignette Hiring Stimuli (Studies 1–3)" for more details on the vignette-style measure).

In Study 1, we found no evidence for a statistically significant interaction effect of political ideology and stereotype portrayal on job candidate choice ($M_{cons*cong} = 0.40$, $SD_{cons*cong} = 0.49$, $M_{cons*incong} = 0.39$, $SD_{cons*incong} = 0.49$, $M_{libs*cong} = 0.24$, $SD_{libs*cong} = 0.43$, $M_{libs*incong} = 0.23$, $SD_{libs*incong} = 0.42$; $F_{(1, 954)} < 0.01$, $p = 0.981$, $\eta^2_p < 0.01$, 95% CI = [0.00, 1.00]) (see Fig. 3A). However, following our preregistration, we tested for and found a significant main effect by political ideology ($F_{(1, 956)} = 29.10$, $p < 0.001$, $\eta^2_p = 0.03$, 95% CI = [0.01, 1.00]), showing that conservative participants ($M_{cons} = 0.40$, $SD_{cons} = 0.49$) were more likely than liberal participants ($M_{libs} = 0.24$, $SD_{libs} = 0.43$) to choose the stereotype congruent Asian Male candidate.

In Study 2, we found a significant interaction effect ($M_{cons*cong} = 0.44$, $SD_{cons*cong} = 0.50$, $M_{cons*incong} = 0.52$, $SD_{cons*incong} = 0.50$, $M_{libs*cong} = 0.29$, $SD_{libs*cong} = 0.45$, $M_{libs*incong} = 0.21$, $SD_{libs*incong} = 0.41$; $F_{(1, 896)} = 5.99$, $p = 0.015$; $\eta^2_p < 0.01$, 95% CI = [0.00, 1.00]) (see Fig. 3B), which held in linear regression models with and without controlling for participants' age, race, gender, income, and educational attainment ($b = 0.16$, $SE = 0.06$, $p = 0.013$, $\eta^2_p < 0.01$, 95% CI = [0.00, 1.00]; $b = 0.15$, $SE = 0.06$, $p = 0.015$, $\eta^2_p < 0.01$, 95% CI = [0.00, 1.00]). Simple effects analyses showed a significant difference between liberals and conservatives in both the congruent condition ($t_{(896)} = 3.34$, $p < .001$, Cohen's $d = 0.22$, 95% CI = [0.09, 0.35]) and the incongruent condition ($t_{(896)} = 6.93$, $p < 0.001$, Cohen's $d = 0.46$, 95% CI = [0.33, 0.60]). There was also a significant main effect of political ideology ($M_{cons} = 0.48$, $SD_{cons} = 0.50$, $M_{libs} = 0.25$, $SD_{libs} = 0.43$; $t_{(898)} = 7.05$, $p < 0.001$, Cohen's $d = 0.50$, 95% CI = [0.36, 0.63]) on software candidate selection. The results from Study 2 show that the behavioral selection of a stereotype congruent software candidate is more likely to occur among

conservative participants in general and is most likely to occur among conservative participants who viewed incongruent stereotype portrayals, which suggests a penalization effect of incongruent ads on conservatives who later acted more often in line with their preference for endorsing traditional stereotypes. Additionally, while liberals in general were already less likely than conservatives to choose the stereotype-congruent software candidate, liberals who had viewed incongruent stereotype portrayals were even less likely to choose the congruent candidate, which suggests a boosting effect of incongruent ads for liberals to further act upon their preference for progressive portrayals of minorities. The same incongruent ads led to even more amplified and divergent responses from conservatives and liberals, who acted upon their previous preferences and were more likely to endorse stereotype congruency or stereotype incongruency, respectively, after viewing the ads.

Overall, our results on behavioral outcomes suggest that the effect of stereotype portrayal and political ideology was inconsistent along behavioral measures—perhaps due to the timing or sample constraints of our studies, but at times did show either a penalization or boosting effect on participants' behavioral choices. Further, political ideology alone consistently predicted the choices and endorsements of stereotypes.

### Paradigm replication: white models and gender stereotypes
Next, in Study 3, we tested our paradigm for replication of previous interaction effects by using advertisements that depicted solely white models, maintaining the same contexts used in previous studies (see Table 1 for experimental stimuli).

Beyond paradigm replication, another rationalization for running this paradigm test was to investigate a pattern of results from preceding studies wherein liberals reported more favorable attitudinal and economic perceptions than conservatives overall. Our intuition is that liberals reacted positively to the advertisements in the prior studies because the ads featured all racial minorities, which perhaps matches liberals' ideological beliefs, while conservatives responded negatively to those same advertisements, perhaps because they featured no white models and violated their ideological beliefs. As such, we run Study 3 to probe whether a similar liberal boost effect would be present when the advertisements did not make race-based stereotypes salient, but rather portrayed and manipulated gender-based stereotypes, and when the models were all white.

**Attitudes.** We found a significant interaction effect of stereotype portrayal and political ideology on attitudinal perceptions ($M_{cons*cong} = 5.10$, $SD_{cons*cong} = 0.97$, $M_{cons*incong} = 4.89$, $SD_{cons*incong} = 1.06$, $M_{libs*cong} = 4.24$, $SD_{libs*cong} = 1.00$, $M_{libs*incong} = 4.82$, $SD_{libs*incong} = 1.00$; $F_{(1,779)} = 30.18$, $p < 0.001$, $\eta^2_p = 0.04$, 95% CI = [0.02, 1.00]) (see Fig. 1C), which held in a linear regression ($b = 0.79$, $SE = 0.14$, $p < 0.001$, $\eta^2_p = 0.04$, 95% CI = [0.02, 1.00]) as well as a linear regression controlling for participants' age, race, gender, income, and educational attainment ($b = 0.81$, $SE = 0.14$, $p < 0.001$, $\eta^2_p = 0.04$, 95% CI = [0.02, 1.00]). Exploratory simple effects analyses revealed significant pairwise differences among conservatives in the congruent versus incongruent conditions ($t_{(779)} = 2.05$, $p = 0.041$, Cohen's $d = 0.15$, 95% CI = [0.01, 0.29]), among liberals in the congruent versus incongruent conditions ($t_{(779)} = 5.79$, $p < 0.001$, Cohen's $d = 0.41$, 95% CI = [0.27, 0.56]), and between liberals and conservatives evaluating congruent portrayals ($t_{(779)} = 8.34$, $p < 0.001$, Cohen's $d = 0.60$, 95% CI = [0.45, 0.74]). Exploratory t-tests revealed significant differences between liberals and conservatives ($M_{cons} = 4.99$, $SD_{cons} = 1.02$, $M_{libs} = 4.54$, $SD_{libs} = 1.04$; $t_{(781)} = 6.15$, $p < 0.001$, Cohen's $d = 0.44$, 95% CI = [0.30, 0.58]), and between congruent and incongruent conditions ($M_{cong} = 4.66$, $SD_{cong} = 1.08$, $M_{incong} = 4.85$, $SD_{incong} = 1.02$; $t_{(781)} = 2.57$, $p = 0.010$, Cohen's $d = 0.18$, 95% CI = [0.04, 0.32]). Interestingly, in this study, which featured solely white models, conservatives preferred all advertisements more than liberals, thus flipping the pattern from Studies 1 & 2 in which liberals preferred all advertisements more than conservatives. Indeed, Study 3 shows that liberals perceived all advertisements less favorably than conservatives, but still preferred incongruent portrayals to congruent

portrayals, while conservatives preferred congruent portrayals significantly more than liberals. Furthermore, incongruent portrayals were rated significantly higher than congruent portrayals by the entire sample, underscoring how liberals continue to prefer incongruent portrayals and how conservatives' new, higher attitudes towards incongruent portrayals collectively contribute to this main effect of stereotype portrayal. Thus, along our attitudinal perceptions outcome, we replicate the general interaction pattern and findings of divergence from Studies 1 & 2, but show that the racial make-up of the models in advertisements affects the directions of divergence.

**Economic perceptions.** We did not find a significant interaction effect along economic perceptions ($M_{cons*cong} = \$196$, $SD_{cons*cong} = \$120$, $M_{cons*incong} = \$207$, $SD_{cons*incong} = \$124$, $M_{libs*cong} = \$171$, $SD_{libs*cong} = \$106$, $M_{libs*incong} = \$181$, $SD_{libs*incong} = \$120$; $F_{(1, 779)} < 0.01$, $p = 0.959$, $\eta^2_p < .001$, 95% CI = [0.00, 1.00]) (see Fig. 2C). An exploratory $t$ test showed a significant difference between conservatives and liberals ($M_{cons} = \$202$, $SD_{cons} = \$122$, $M_{libs} = \$176$, $SD_{libs} = \$113$; $t_{(781)} = 3.01$, $p = 0.003$, Cohen's $d = 0.22$, 95% CI = [0.07, 0.36]). On average, conservatives attributed greater perceived value to all advertised goods/services compared to liberals, reversing the direction of preference from Studies 1 & 2 while maintaining a main effect of political ideology. With this, Study 3 reveals another reversal in outcomes between conservatives and liberals, such that conservatives report higher favorable perceptions that now supersede liberals' perceptions across the suite of advertisement stimuli, which is in response to the advertisements portraying all white models.

**Downstream hiring.** Additionally, we found no evidence for a statistically significant interaction effect of political ideology and stereotype portrayal on job candidate choice ($M_{cons*cong} = 0.43$, $SD_{cons*cong} = 0.50$, $M_{cons*incong} = 0.45$, $SD_{cons*incong} = 0.50$, $M_{libs*cong} = 0.29$, $SD_{libs*cong} = 0.45$, $M_{libs*incong} = 0.28$, $SD_{libs*incong} = 0.45$; $F_{(1,779)} = 0.17$, $p = 0.682$, $\eta^2_p < .001$, 95% CI = [0.00, 1.00]) (see Fig. 3C); however, we found a significant main effect by political ideology ($t_{(781)} = 4.71$, $p < .001$, Cohen's $d = 0.34$, 95% CI = [0.20, 0.48]), showing that conservative participants ($M_{cons} = 0.44$, $SD_{cons} = 0.50$) were still more likely than liberal participants ($M_{libs} = 0.28$, $SD_{libs} = 0.45$) to choose the congruent Asian Male candidate, regardless of stereotype portrayal, which is consistent with previous studies.

As such, we find that our paradigm of manipulating stereotype portrayal to be either stereotype congruent or stereotype incongruent will indeed lead to divergent responses on attitudinal outcomes based on the political ideology of the viewer. Of note, in the case where we altered the race of the models being portrayed (i.e., all white models replacing all racial minority models), responses from conservatives and liberals indeed diverged, but in a reversed pattern compared to previous studies.

### Full replication: stereotype portrayal, model race, and political ideology
Lastly, we conducted a final experiment in which we tested our entire paradigm for replication and robustness. We used a factorial design specifying three factors: 2(*stereotype portrayal*: congruent, incongruent) x 2(*model race*: minority, white), x 2(*participant political ideology*: conservative, liberal). The stimuli we used in Study 4 are very similar to previous studies', with some modifications to enhance similarity in model facial expression, body language, and environment across conditions (see Table 1 for experimental stimuli). As in our Pilot Study, we pre-tested target advertisements for realism and stereotype alignment (see Supplementary Materials Appendix B subsection "Study 4 Supplementary Results" for pretest results). As preregistered, we focus on testing the interaction effects of *political ideology x stereotype portrayal*, of *political ideology x model race*, and of *political ideology x stereotype portrayal x model race* on our dependent measures. We leave the exploratory tests of *stereotype portrayal x model race* on our outcome measures to the Supplementary Materials Appendix B

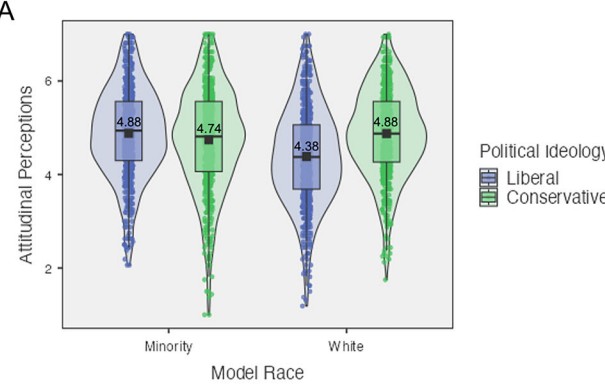

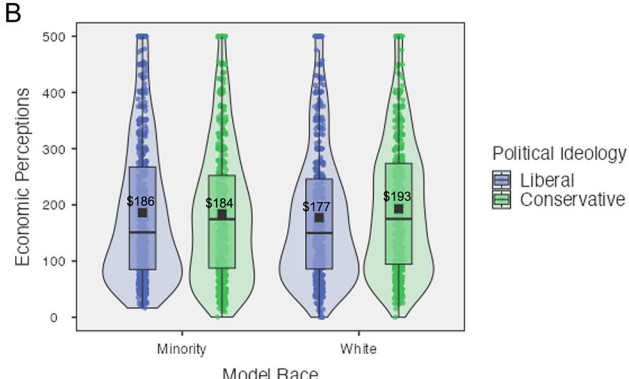

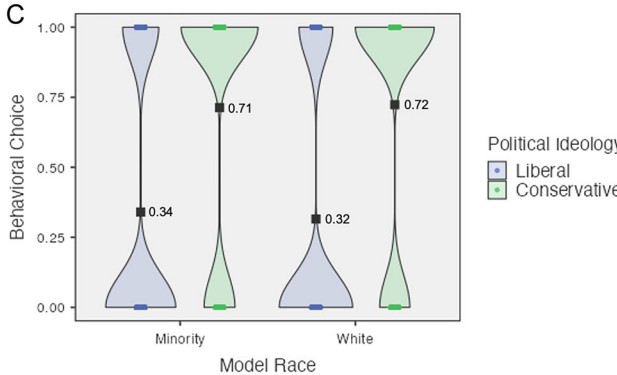

**Fig. 4 | Violin plots for outcome measures by stereotype portrayal and model race, study 4.** Violin plot for attitudes is contained in (**A**); economic perceptions are contained in (**B**); downstream hiring is contained in (**C**). These plots map the mean (i.e., black square point), raw data (i.e., circular points), and box-plot elements (i.e., center line, median; box limits, upper and lower quartiles; whiskers, 1.5× inter-quartile range). **C** *Y* axis Behavioral Choice: 0 = Stereotype Incongruent Candidate Hired, 1 = Stereotype Congruent Candidate Hired. Figure legend: Blue = Liberals, Green = Conservatives.

subsection "Study 4 Supplementary Results," as these are not *a priori* confirmatory analyses.

**Attitudes**. We sought to understand how attitudes towards the target advertisements would change based on the advertisement's stereotype portrayal, the model's race depicted in the advertisement, and participant's political ideology.

**Confirmatory two-way factorial ANOVAs.** First, we conducted a confirmatory two-way factorial ANOVA investigating the effect of *political ideology* and *stereotype portrayal* on attitudes. We found a significant interaction effect on attitudes ($M_{cons*cong} = 4.89$, $SD_{cons*cong} = 1.01$, $M_{cons*incong} = 4.73$, $SD_{cons*incong} = 1.04$, $M_{libs*cong} = 4.53$, $SD_{libs*cong} = 1.07$, $M_{libs*incong} = 4.75$, $SD_{libs*incong} = 0.98$; $F_{(1, 2480)} = 20.70$, $p < .001$, $\eta^2_p < 0.01$, 95% CI = [0.00, 1.00]) (see Fig. 1D), which held in a linear regression ($b = 0.38$, $SE = 0.08$, $p < .001$, $\eta^2_p < .01$, 95% CI = [0.00, 1.00]) as well as in a linear regression controlling for participants' age, race, gender, income, urban density, and educational attainment ($b = 0.38$, $SE = 0.08$, $p < .001$, $\eta^2_p < 0.01$, 95% CI = [0.00, 1.00]).

In simple effects analyses investigating the effect of stereotype portrayal at each level of political ideology, we found significant contrasts among liberals evaluating stereotype congruent versus incongruent stimuli ($t_{(2480)} = 3.84$, $p < 0.001$, Cohen's $d = 0.15$, 95% CI = [0.08, 0.23]) and among conservatives evaluating stereotype congruent versus incongruent stimuli ($t_{(2480)} = 2.63$, $p = 0.009$, Cohen's $d = 0.11$, 95% CI = [0.03, 0.18]), such that liberals preferred incongruent portrayals over congruent portrayals, while conservatives preferred congruent portrayals over incongruent portrayals. Evaluating the effect of political ideology at each stereotype portrayal

condition, we found significant contrasts between liberal and conservative participants viewing stereotype congruent portrayals ($t_{(2480)} = 6.15$, $p < .001$, Cohen's $d = 0.25$, 95% CI = [0.17, 0.33]) such that conservatives expressed greater preference than liberals for congruent portrayals, but found no evidence for significant contrasts between liberal and conservative participants viewing incongruent portrayals ($t_{(2480)} = 0.31$, $p = 0.756$, Cohen's $d = 0.01$, 95% CI = [−0.07, 0.09]).

Second, we conducted a confirmatory two-way factorial ANOVA investigating the effect of *political ideology* and *model race* on attitudes. We found a significant interaction effect ($M_{cons*minor} = 4.74$, $SD_{cons*minor} = 1.10$, $M_{cons*white} = 4.88$, $SD_{cons*white} = 0.94$, $M_{libs*minor} = 4.88$, $SD_{libs*minor} = 0.96$, $M_{libs*white} = 4.38$, $SD_{libs*white} = 1.04$; $F_{(1, 2480)} = 59.91$, $p < .001$, $\eta^2_p = 0.02$, 95% CI = [0.01, 1.00]) (see Fig. 4A). This result held in a linear regression model ($b = 0.63$, $SE = 0.08$, $p < .001$, $\eta^2_p = 0.02$, 95% CI = [0.01, 1.00]) and when controlling for participants' demographics ($b = 0.64$, $SE = 0.08$, $p < .001$, $\eta^2_p = 0.02$, 95% CI = [0.02, 1.00]).

Simple effects analyses showed significant contrasts between model race conditions among liberals ($t_{(2480)} = 8.94$, $p < 0.001$, Cohen's $d = 0.36$, 95% CI = [0.28, 0.44]) and among conservatives ($t_{(2480)} = 2.23$, $p = 0.026$, Cohen's $d = 0.09$, 95% CI = [0.01, 0.17]), revealing that liberals prefer advertisements featuring minority models significantly more than advertisements featuring white models, whereas conservatives prefer advertisements featuring white models significantly more than advertisements featuring minority models. Additionally, simple effect analyses revealed significant contrasts between conservatives and liberals evaluating advertisements with minority models ($t_{(2480)} = 2.41$, $p = 0.016$, Cohen's $d = 0.10$, 95% CI = [0.02, 0.18]) and advertisements with white models ($t_{(2480)} = 8.51$, $p < .001$, Cohen's $d = 0.34$, 95% CI = [0.26, 0.42]), such that

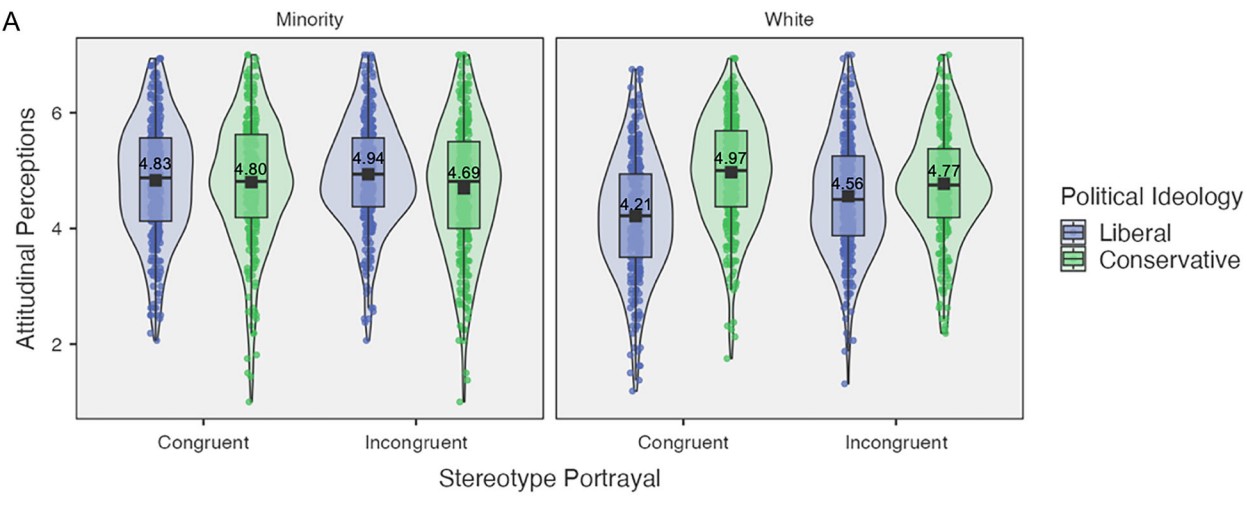

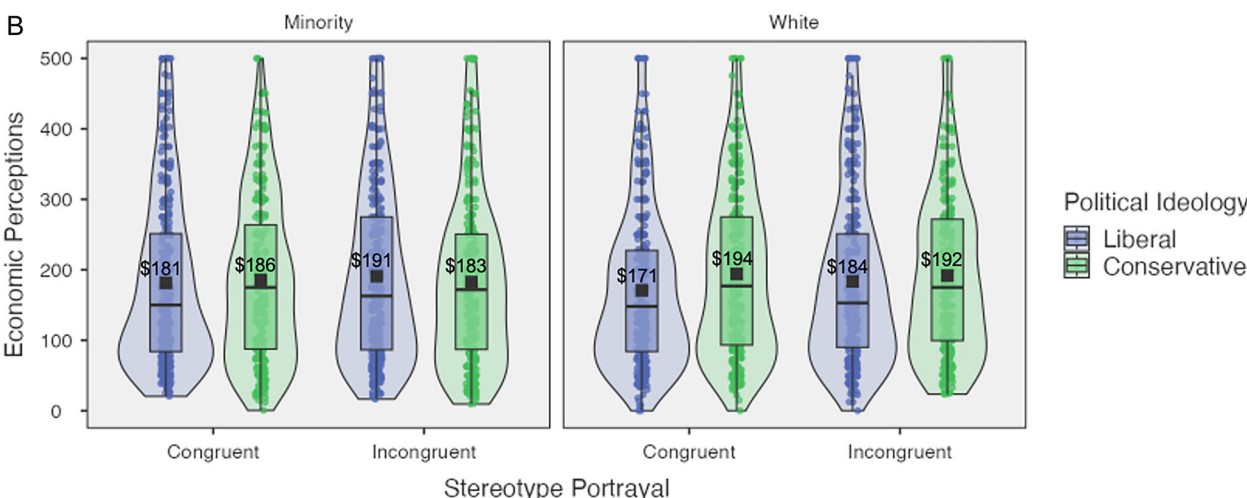

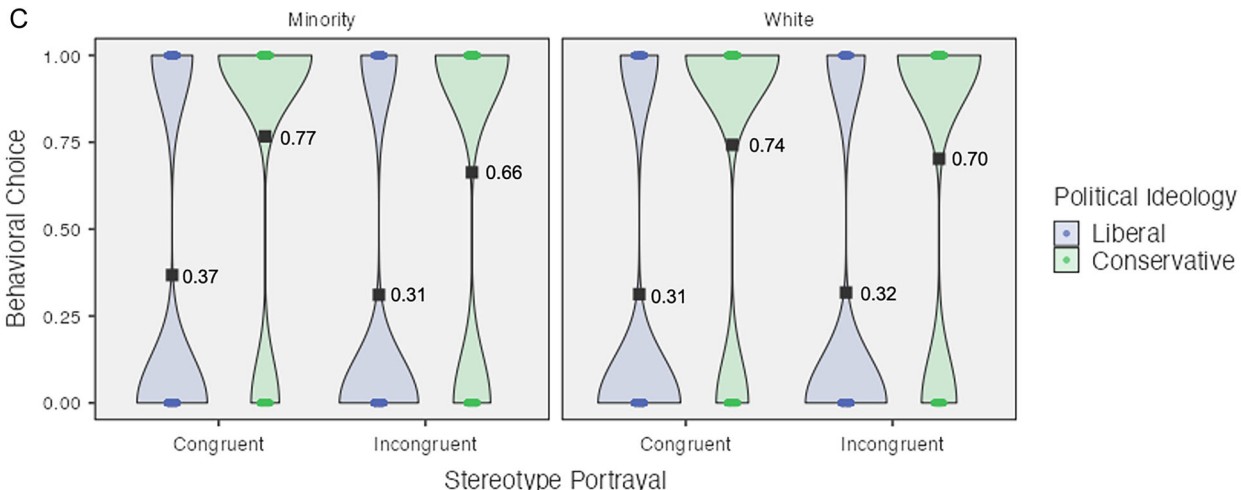

**Fig. 5 | Violin plots for outcome measures by three-way interaction (political ideology, stereotype portrayal, and model race), study 4.** Violin plot for attitudes is contained in (**A**); economic perceptions are contained in (**B**); downstream hiring is contained in (**C**). These plots map the mean (i.e., black square point), raw data (i.e., circular points), and box-plot elements (i.e., center line, median; box limits, upper and lower quartiles; whiskers, 1.5× interquartile range). **C** Y axis Behavioral Choice: 0 = Stereotype Incongruent Candidate Hired, 1 = Stereotype Congruent Candidate Hired. Figure legend: Blue = Liberals, Green = Conservatives.

liberals preferred minority models significantly more than conservatives, while conservatives prefer white models significantly more than liberals.

Please reference the Supplementary Materials Appendix B subsection "Study 4 Supplementary Results" for the secondary results of an exploratory two-way ANOVA testing for an interaction effect of *stereotype portrayal* and *model race* on attitudes.

**Exploratory three-way factorial ANOVAs.** We were curious to explore whether an interaction effect of all three factors was significant and influenced attitudinal perceptions. As such, we conducted an exploratory three-way factorial ANOVA investigating the effect of *political ideology*, *stereotype condition*, and *model race* on attitudes. We found a significant three-way interaction effect on attitudes ($M_{cons*cong*minor} = 4.80$, $SD_{cons*cong*minor} = 1.08$, $M_{cons*cong*white} = 4.97$, $SD_{cons*cong*white} = 0.93$, $M_{cons*incong*minor} = 4.69$, $SD_{cons*incong*minor} = 1.12$, $M_{cons*incong*white} = 4.77$, $SD_{cons*incong*white} = 0.95$, $M_{libs*cong*minor} = 4.83$, $SD_{libs*cong*minor} = 1.00$, $M_{libs*cong*white} = 4.21$, $SD_{libs*cong*white} = 1.04$, $M_{libs*incong*minor} = 4.94$, $SD_{libs*incong*minor} = 0.92$, $M_{libs*incong*white} = 4.56$, $SD_{libs*incong*white} = 1.01$; $F_{(1, 2476)} = 4.06$, $p = 0.044$, $\eta^2_p < .01$, 95% CI = [0.00, 1.00]) (see Fig. 5A). Exploring this significant three-way interaction, we conducted a simple effects analysis.

We first examined how the effect of political ideology varied across the levels of stereotype portrayal and model race. Among congruent portrayals, political ideology had a significant effect when white models were featured ($t_{(2476)} = 9.42$, $p < 0.001$, Cohen's $d = 0.38$, 95% CI = [0.30, 0.46]) such that conservatives expressed significantly higher attitudes toward white models than liberals. Among congruent portrayals, we found no evidence for a significant difference between conservatives and liberals evaluating ads featuring minority models ($t_{(2476)} = 0.38$, $p = 0.707$, Cohen's $d = 0.02$, 95% CI = [−0.06, 0.09]). Among incongruent portrayals, political ideology had a significant effect both when white models were featured ($t_{(2476)} = 2.62$, $p = 0.009$, Cohen's $d = 0.11$, 95% CI = [0.03, 0.18]) and when minority models were featured ($t_{(2476)} = 3.05$, $p = 0.002$, Cohen's $d = 0.12$, 95% CI = [0.04, 0.20]), revealing that, under incongruent conditions, conservatives significantly preferred white models more than liberals and that liberals significantly preferred minority models more than conservatives.

We also examined the effect of stereotype portrayal across the levels of political ideology and model race. For liberals, stereotype portrayal had a significant effect when white models were featured ($t_{(2476)} = 4.35$, $p < 0.001$, Cohen's $d = 0.17$, 95% CI = [0.10, 0.25]), but not when minority models were featured ($t_{(2476)} = 1.35$, $p = 0.176$, Cohen's $d = 0.05$, 95% CI = [−0.02, 0.13]). Liberals significantly preferred incongruent depictions of white models more than congruent depictions of white models, but did not show any differences in attitudes toward minority models on the basis of stereotype portrayal. For conservatives, stereotype portrayal also had a significant effect on attitudes when white models were featured ($t_{(2476)} = 2.36$, $p = 0.019$, Cohen's $d = 0.09$, 95% CI = [0.02, 0.17]), but not when minority models were featured ($t_{(2476)} = 1.31$, $p = 0.190$, Cohen's $d = 0.05$, 95% CI = [−0.03, 0.13]). Conservatives significantly preferred congruent portrayals of white models more than incongruent portrayals of white models and did not show any differences in attitudes toward minority models in congruent versus incongruent portrayals.

Further, we examined the effect of model race at each level of political ideology and stereotype portrayal. Among liberals, model race had a significant effect on attitudes under congruent stereotype portrayals ($t_{(2476)} = 7.96$, $p < 0.001$, Cohen's $d = 0.32$, 95% CI = [0.24, 0.40]) and under incongruent stereotype portrayals ($t_{(2476)} = 4.78$, $p < .001$, Cohen's $d = 0.19$, 95% CI = [0.11, 0.27]). Liberals prefer minority models significantly more than white models across both congruent and incongruent portrayal conditions. Among conservatives, model race had a significant effect on attitudes only under stereotype congruent portrayals ($t_{(2476)} = 2.03$, $p = 0.042$, Cohen's $d = 0.08$, 95% CI = [0.00, 0.16]) and not under stereotype incongruent portrayals ($t_{(2476)} = 0.99$, $p = 0.323$, Cohen's $d = 0.04$, 95% CI = [−0.04, 0.12]). Conservatives prefer white models significantly more than minority models under stereotype-congruent portrayals, and showed no

evidence of a significant difference in attitudes between minority or white models under stereotype-incongruent portrayals.

**Economic perceptions.** Next, we tested for interaction effects of our experimental factors to assess whether our manipulations affect economic perceptions of the target advertisements.

**Confirmatory two-way factorial ANOVAs.** We first sought to evaluate two-way factorial interaction effects. In a confirmatory two-way factorial ANOVA investigating the effect of *political ideology* and *stereotype portrayal* on economic perceptions, we found no evidence for a significant interaction effect ($M_{cons*cong} = $190, $SD_{cons*cong} = $117, $M_{cons*incong} = $187, $SD_{cons*incong} = $118, $M_{libs*cong} = $176, $SD_{libs*cong} = $118, $M_{libs*incong} = $187, $SD_{libs*incong} = $123; $F_{(1, 2480)} = 2.10$, $p = 0.147$, $\eta^2_p < .01$, 95% CI = [0.00, 1.00]) (see Fig. 2D). We also found no evidence for significant main effects of political ideology ($M_{cons} = $189, $SD_{cons} = $117, $M_{libs} = $182, $SD_{libs} = $120; $F_{(1, 2480)} = 2.01$, $p = 0.156$, $\eta^2_p < .01$, 95% CI = [0.00, 1.00]) or of stereotype portrayal ($M_{cong} = $183, $SD_{cong} = $118, $M_{incong} = $187, $SD_{incong} = $120; $F_{(1, 2480)} = 0.87$, $p = 0.351$, $\eta^2_p < .01$, 95% CI = [0.00, 1.00]).

Similarly, in a confirmatory two-way factorial ANOVA investigating the effect of *political ideology* and *model race* on economic perceptions, we found no evidence for a significant interaction effect ($M_{cons*minor} = $184, $SD_{cons*minor} = $118, $M_{cons*white} = $193, $SD_{cons*white} = $117, $M_{libs*minor} = $186, $SD_{libs*minor} = $124, $M_{libs*white} = $177, $SD_{libs*white} = $117; $F_{(1, 2480)} = 3.41$, $p = 0.065$, $\eta^2_p < .01$, 95% CI = [0.00, 1.00]) (see Fig. 4B). We found no evidence for significant main effects of political ideology ($M_{cons} = $189, $SD_{cons} = $117, $M_{libs} = $182, $SD_{libs} = $120; $F_{(1, 2480)} = 2.04$, $p = 0.154$, $\eta^2_p < 0.01$, 95% CI = [0.00, 1.00]]) or of model race ($M_{minor} = $185, $SD_{minor} = $121, $M_{white} = $185, $SD_{white} = $117; $F_{(1, 2480)} = 0.01$, $p = 0.916$, $\eta^2_p < .01$, 95% CI = [0.00, 1.00]).

Please refer to the Supplementary Materials Appendix B subsection "Study 4 Supplementary Results" for the results of an exploratory two-way ANOVA testing for an interaction effect of *stereotype portrayal* and *model race* on economic perceptions.

**Exploratory three-way factorial ANOVAs.** Finally, we investigated whether a three-way interaction of all three factors exists and influences economic perceptions. In an exploratory three-way factorial ANOVA, we found no evidence for a significant interaction effect of *political ideology*, *stereotype portrayal*, and *model race* on economic perceptions ($M_{cons*cong*minor} = $186, $SD_{cons*cong*minor} = $117, $M_{cons*cong*white} = $194, $SD_{cons*cong*white} = $118, $M_{cons*incong*minor} = $183, $SD_{cons*incong*minor} = $129, $M_{cons*incong*white} = $192, $SD_{cons*incong*white} = $116, $M_{libs*cong*minor} = $181, $SD_{libs*cong*minor} = $123, $M_{libs*cong*white} = $171, $SD_{libs*cong*white} = $112, $M_{libs*incong*minor} = $191, $SD_{libs*incong*minor} = $125, $M_{libs*incong*white} = $184, $SD_{libs*incong*white} = $121; $F_{(1, 2476)} = 0.02$, $p = 0.888$, $\eta^2_p < .01$, 95% CI = [0.00, 1.00]) (see Fig. 5B).

**Downstream hiring.** For our last outcome measure, we conducted confirmatory and exploratory factorial ANOVAs to test for interaction effects on downstream hiring.

**Confirmatory two-way factorial ANOVAs.** We first tested for two-way interaction effects on downstream hiring. Investigating our behavioral measure, we conducted a confirmatory two-way factorial ANOVA investigating the effect of political ideology and stereotype portrayal on downstream hiring. We found no evidence for a significant interaction effect ($M_{cons*cong} = 0.75$, $SD_{cons*cong} = 0.43$, $M_{cons*incong} = 0.68$, $SD_{cons*incong} = 0.47$, $M_{libs*cong} = 0.34$, $SD_{libs*cong} = 0.47$, $M_{libs*incong} = 0.31$, $SD_{libs*incong} = 0.46$; $F_{(1, 2480)} = 1.49$, $p = 0.222$, $\eta^2_p < 0.01$, 95% CI = [0.00, 1.00]) (see Fig. 3D). We did find significant main effects of political ideology ($M_{cons} = 0.72$, $SD_{cons} = 0.45$, $M_{libs} = 0.33$, $SD_{libs} = 0.47$; $F_{(1, 2480)} = 446.03$, $p < 0.001$, $\eta^2_p = 0.15$, 95% CI = [0.13, 1.00]) and of stereotype portrayal ($M_{cong} = 0.53$, $SD_{cong} = 0.50$, $M_{incong} = 0.49$, $SD_{incong} = 0.50$; $F_{(1, 2480)} = 5.90$, $p = 0.015$, $\eta^2_p < 0.01$, 95% CI = [0.00, 1.00]). Overall, conservatives were

significantly more likely to hire the stereotype-congruent software candidate (i.e., the Asian Male candidate as opposed to the Black Female candidate) than liberals, and participants in the congruent condition were more likely than participants in the incongruent condition to hire the congruent candidate.

Next, in a confirmatory two-way factorial ANOVA investigating the effect of political ideology and model race on downstream hiring, we found no evidence for a significant interaction effect ($M_{cons*minor} = 0.71$, $SD_{cons*minor} = 0.45$, $M_{cons*white} = 0.72$, $SD_{cons*white} = 0.45$, $M_{libs*minor} = 0.34$, $SD_{libs*minor} = 0.47$, $M_{libs*white} = 0.32$, $SD_{libs*white} = 0.46$; $F_{(1, 2480)} = 0.89$, $p = 0.345$, $\eta^2_p < 0.01$, 95% CI = [0.00, 1.00]) (see Fig. 4C). We did find a significant main effect of political ideology on hiring ($M_{cons} = 0.72$, $SD_{cons} = 0.45$, $M_{libs} = 0.33$, $SD_{libs} = 0.47$; $F_{(1, 2480)} = 444.06$, $p < 0.001$, $\eta^2_p = 0.15$, 95% CI = [0.13, 1.00]), but no evidence for a significant effect of model race on hiring ($M_{minor} = 0.51$, $SD_{minor} = 0.50$, $M_{white} = 0.51$, $SD_{white} = 0.50$; $F_{(1, 2480)} = 0.04$, $p = 0.837$, $\eta^2_p < 0.01$, 95% CI = [0.00, 1.00]). Again, conservatives were significantly more likely to hire the stereotype-congruent software candidate than liberals, but model race had no standalone effect on downstream hiring.

Please see the Supplementary Materials Appendix B subsection "Study 4 Supplementary Results" for the results of an exploratory two-way ANOVA testing for an interaction effect of *stereotype portrayal* and *model race* on downstream hiring.

**Exploratory three-way factorial ANOVAs.** Finally, we examined whether a three-way factorial interaction effect existed to affect downstream hiring. In an exploratory three-way factorial ANOVA, we found no evidence for a significant interaction effect of political ideology, stereotype portrayal, and model race on downstream hiring ($M_{cons*cong*minor} = 0.77$, $SD_{cons*cong*minor} = 0.42$, $M_{cons*cong*white} = 0.74$, $SD_{cons*cong*white} = 0.44$, $M_{cons*incong*minor} = 0.66$, $SD_{cons*incong*minor} = 0.47$, $M_{cons*incong*white} = 0.70$, $SD_{cons*incong*white} = 0.46$, $M_{libs*cong*minor} = 0.37$, $SD_{libs*cong*minor} = 0.48$, $M_{libs*cong*white} = 0.31$, $SD_{libs*cong*white} = 0.46$, $M_{libs*incong*minor} = 0.31$, $SD_{libs*incong*minor} = 0.46$, $M_{libs*incong*white} = 0.32$, $SD_{libs*incong*white} = 0.47$; $F_{(1, 2476)} = 0.00$, $p = 0.979$, $\eta^2_p < 0.01$, 95% CI = [0.00, 1.00]) (see Fig. 5C).

## Discussion

The polarization between politically liberal and conservative individuals on issues related to equity, diversity and inclusion has reached a recent peak in the United States[1,2]. This ideological split has important consequences for the way race and gender identities are perceived and for how stereotypes are perpetuated. In this work, we examined how political ideology affects the way U.S. citizens evaluate portrayals of race and gender identities that are either in line and congruent with traditional stereotypes or divergent and incongruent with traditional stereotypes.

Across four studies ($N = 5125$), we found evidence of an interaction effect wherein evaluations of advertisements depend on the viewer's political ideology, the stereotype congruency (or lack thereof) depicted in the ad, and the model race(s) featured in the ad. In particular, along our attitudinal perceptions outcome, we find consistent results and provide a fuller understanding of how individuals with different political beliefs respond to stereotype portrayals and to varying racial representation. As for our economic and behavioral outcomes, we find initial, but insubstantial, evidence for interaction effects and consistent main effects of political ideology.

Importantly, our results show not just a consistency in attitudinal responses from liberals who exhibit stronger preference for incongruent portrayals than their conservative counterparts and, amongst themselves, significantly prefer incongruent over congruent portrayals (Studies 1–4), but also a malleability in responses from both liberals and conservatives: liberals typically prefer incongruent portrayals and are willing to withhold favorable attitudes from congruent portrayals, while conservatives typically prefer congruent portrayals, but express higher attitudes towards incongruent portrayals than liberals under certain conditions (Studies 3 & 4). Further, we found that the pattern from Studies 1 & 2, wherein liberals preferred all advertisements more than conservatives, was flipped in our

results for Study 3, such that conservatives preferred all advertisements more than liberals when only white models and a manipulation of gender stereotypes were portrayed. Indeed, liberals consistently prefer all portrayals more than conservatives when the models are racial minorities (Studies 1, 2, and 4), whereas conservatives prefer all portrayals more than liberals when the models are white (Studies 3 & 4). Notably, we also found that incongruent portrayals were rated significantly higher than congruent portrayals by the entire sample (Study 3), underscoring not only how liberals consistently prefer incongruent portrayals overall (Studies 1, 3, and 4), but also how conservatives' attitudes can flip towards preferring incongruent portrayals more than liberals when only white models are presented (Studies 3 & 4); these two forces collectively contribute to a significant main effect of stereotype portrayal on attitudes.

Our findings that liberals and conservatives are sensitive to the ways stereotype incongruency is portrayed and that their attitudes are malleable under different conditions are worth noting and following up on. Both groups of participants' attitudes change when portrayals feature models that may inherently underscore or contradict their different worldviews: conservatives are more reluctant to favor incongruent portrayals but show an openness towards endorsing incongruency when white models are featured, while liberals endorse portrayals of minority races and of incongruency but withhold this endorsement for solely white models. Future work will explore and conduct additional paradigm tests featuring socially dominant models, higher status models, or other depictions of political worldviews in various contexts. While our attitudinal findings are robust and consistently manipulated, our economic perceptions and downstream behavioral outcomes fail to provide substantial evidence of interaction effects, though they do reveal interesting and reliable main effects of political ideology.

We found an initial significant interaction effect of political ideology and stereotype portrayal on monetary evaluation in Study 1, such that liberals (compared to conservatives) expect to spend more on incongruently depicted professional services, which lines up with our attitudinal finding that liberals prefer incongruent portrayals significantly more than conservatives do (Studies 1 & 2). However, this interaction effect did not replicate in studies beyond Study 1. We did find a significant main effect of political ideology on economic perceptions across Studies 1–3, such that liberals spend significantly more on all professional services than conservatives when the services are modeled by racial minorities (Studies 1 & 2); interestingly, this main effect reverses and reveals that conservatives spend significantly more than liberals when the services are modeled by white individuals (Study 3). We found no notable interaction effects on economic perceptions in our three-way factorial experiment in Study 4.

Additionally, we found an initial significant interaction effect of political ideology and stereotype portrayal on downstream behaviors in Study 2. These results reveal that stereotype portrayal seems to act as an enhancer of pre-existing political ideology on downstream hiring selections: conservatives who viewed incongruent stereotype portrayals were more likely to hire the stereotype congruent job candidate than those who viewed congruent portrayals, while liberals who viewed incongruent portrayals were even less likely to choose the congruent candidate than when they viewed congruent portrayals. The same incongruent portrayals led to more amplified and divergent responses from conservatives and liberals, who reacted to those incongruent portrayals in ways that penalize such portrayals (i.e., conservatives) or endorse such portrayals (i.e., liberals). While the interaction effect did not replicate in other studies, we provide consistent evidence that behavioral choices are directly influenced by political ideology, wherein conservatives were more likely than liberals to hire the congruent job candidate (Studies 1–4).

Taken together, our findings build upon and advance the intersection of social and political psychology by directly testing stereotype incongruent portrayals of varied race and gender identities, through which we reveal politically divergent responses. Unlike previous work, which either focused on traditional stereotypes among political ideologies or on

counterstereotypes without regard to political ideology, our work centers on both stereotype congruence and political ideology.

We establish a clear divergence across political ideologies in attitudinal perceptions of explicit stereotypes, thus providing competing evidence to previous work, which suggested that liberal and conservatives held the same implicit preferences but that one group's preferences may simply be weaker than the other's[20]. Perhaps explicit reports in response to explicit stereotypes effectively extract the divergence across political ideology. Certainly, our participants were willing to directly express their like or dislike of the representations and portrayals they viewed, showing that affective attitudinal responses, while not directed at immediate political opponents, continue to be openly expressed at targets that seem to represent opposing political views; this expands prior work[12,13] and posits that extreme affective polarization can be transferred to objects or inanimate objects that merely represent opposing political views. Future work might consider testing both implicit and explicit preferences to stimuli that range from the obvious to the implicit.

Moreover, we test our interactive dynamic not just on attitudes, but also across economic attributions and downstream behaviors, finding initial evidence of interaction effects and underscoring the consistency of political ideology as a main driver of these outcomes. Critically, we also manipulate and evaluate the effect of racial representation, or model race, on politically conservative and politically liberal participants' responses to stereotype portrayal, finding that liberals continue to favor portrayals that feature racial minorities while conservatives favor portrayals that feature only white individuals. In fact, depicting all white models is key in revealing a conservative malleability and higher attitudinal score towards incongruent portrayals: when advertisements are portrayed by only white individuals, conservatives show increased attitudes and heightened preference for stereotype incongruent (Studies 3 & 4). Taken together, we show a significant and robust divergence between political ideologies, and the divergence in reactions is characterized by both consistency and variability among liberals and conservatives. The variability among conservatives' evaluations contends with prior work that both conservatives and liberals are equally intolerant and staunch in their views[16,17]. Indeed, our work provides support for the argument that conservatives effectively and considerably respond to certain stimuli. Just as prior work has shown that conservatives typically penalize counterstereotypical people but will only do so when the counterstereotypicality is functional or provides utility[21], our work shows that conservatives respond favorably to portrayals of white models, perhaps because there is a functionality or usefulness for them in those representations. Importantly, our work advances past research on political ideology, stereotype congruency, and racial representation by providing a thorough investigation into the effect of these three factors on attitudes across a variety of target advertisements.

## Limitations

We note several limitations in this work. Primarily, our studies use convenience samples of online participants. While there are various benefits to this approach, including the ability to pre-screen for U.S. citizenship (thus targeting participants who are familiar with stereotypes in the United States, who had the potential to be politically active, and who represent American political ideologies), the ability to obtain a large and balanced sample of conservatives and liberals, and the speed with which data can be collected, there is a need to conduct an experiment either in the field or with a field sample, albeit challenges in resource and access to recruit a large field sample may exist. We expect a field study would replicate the results we found in our four studies and would extend the generalizability of our findings, as our findings were consistent across time and participant samples. Of note, future work may also consider exploring how the demographic make-up beyond the political ideology of the participants may moderate results. For instance, recruiting only non-dominant identities (i.e., only racial minorities or only female liberal and conservative participants) and assessing their reactions to congruent versus incongruent portrayals of their own non-dominant group may reveal findings important to understanding further intersections of identity and representation. Further, future studies may consider dissecting

and breaking down stereotype incongruence into two subcategories, strict counterstereotypes (i.e., depictions of individuals that directly counter commonly-held stereotypes) versus non-stereotypes (i.e., depictions of individuals that are neither blatantly stereotypical nor counterstereotypical; depictions that are unaffiliated with stereotypes), and investigating how conservative and liberal viewers respond to such depictions.

Our findings highlight the consequences of stereotype content and political belief on evaluations of race and gender in popular media and workplace contexts. Our work advances the study of political ideology, stereotypes, and social perceptions and brings into sharp focus the reactions to stereotype-congruent versus stereotype-incongruent portrayals of race and gender. This work provides a clear starting point on the differences between conservative and liberal responses to portrayals of varying race and gender identities. We hope these findings will inspire future inquiry into how and why differences in political ideology influence the way racial minorities, white individuals, men and women are perceived. Given the intense polarization of the U.S.'s political landscape, especially with respect to diversity issues, it is imperative for researchers and organizations to continue investigating and understanding the interaction between political ideology, stereotype portrayal, and racial representation.

## Data availability

All study materials, anonymized raw participant survey data, and pre-registrations are in the Open Science Framework project repository, which is publicly accessible at the following link: https://osf.io/cxpgk/ (identifier: https://doi.org/10.17605/OSF.IO/CXPGK).

## Code availability

The analyses herein were preregistered, and the RStudio analysis code files are available in the Open Science Framework project repository, which is publicly accessible at the following link: https://osf.io/cxpgk/ (identifier: https://doi.org/10.17605/OSF.IO/CXPGK).

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

## Acknowledgements

The authors would like to thank the Morrison Center for Marketing and Data Analytics at UCLA Anderson for partial funding of this study. The funders had no role in study design, data collection and analysis, decision to publish or preparation of the manuscript. Additionally, the authors would like to thank the Social and Identity Lab (SAIL) at UCLA Anderson for their feedback.

## Author contributions

E.Q.J. proposed the original research questions and broad study design, conducted the analyses, drafted the original manuscript, and prepared all public data, analysis code, and materials. M.J.S. supervised the project, contributed to theory development, data interpretation, provided resources, and developed the manuscript. All authors contributed to the refinement of the study design, the hypotheses, and the planned analysis. CRediT author statement: E.Q.J.: conceptualization, methodology, formal analysis, investigation, writing—original draft, visualization, project administration, funding acquisition; M.J.S.: conceptualization, writing—review & editing, supervision, funding acquisition.

## Competing interests

The authors declare no competing interests.
