## [Transparent Peer Review file · Communications Psychology]

Liberals and conservatives respond divergently to stereotype portrayals of race and gender

Corresponding Author: Ms Elizabeth Jiang

Version 0:

Decision Letter:

Dear Ms Jiang,

Thank you for your patience during the peer-review process. Your manuscript titled "Liberals and conservatives respond divergently to stereotype portrayals of racial minorities and women" has now been seen by 2 reviewers, whose comments are appended below. You will see that they find your work of some potential interest. However, they have raised quite substantial concerns that must be addressed. In light of these comments, we cannot accept the manuscript for publication, but would be interested in considering a revised version that fully addresses these serious concerns.

We hope you will find the Reviewers' comments useful as you decide how to proceed. Should additional work allow you to address these criticisms, we would be happy to look at a substantially revised manuscript. If you choose to take up this option, please highlight all changes in the manuscript text file, and provide a detailed point-by-point reply to the reviewers.

Editorially, we consider it important to conduct a fully pre-registered replication, as suggested by Reviewer 1. This would help to ensure the robustness of your findings and respond to Reviewer 1's concerns surrounding the existing pre-registrations. The policy for pre-registrations at Communications Psychology is as follows. Authors must disclose all deviations from the preregistered protocol and explain the rationale for deviation (e.g., flaw, feasibility, suboptimality). In cases of deviation from the preregistered analysis plan for reasons other than fundamental flaw or feasibility, the originally planned analyses must also be reported (see <https://www.nature.com/commpsychol/editorial-policies/preregistration-policy>).

I am also attaching a checklist that details critical reporting requirements for the revised manuscript. Please attend to each item and ensure your manuscript is fully compliant. We are requesting that your manuscript aligns with these requirements as this facilitates the evaluation of your manuscript, reducing delays in re-review and potential future acceptance. If your revised manuscript is not aligned with these requests on major issues, such as those concerning statistics, it may be returned to you for further revisions without re-review. Additional information can be found in our style and formatting guide <https://www.nature.com/documents/commpsychol-style-formatting-guide-accept.pdf> Communications Psychology formatting guide.

If the revision process takes significantly longer than five months, we will be happy to reconsider your paper at a later date, provided it still presents a significant contribution to the literature at that stage.

Please use the following link to submit your
- revised manuscript,

- point-by-point response to the referees' comments,
- cover letter (as a separate document),
- the Editorial Policy Checklist (see below),
- the Reporting Summary (see below), and
- the completed Editorial Request Table (attached):

Link Redacted

Thank you for the opportunity to review your work.

Best regards,

Patricia Lockwood

Patricia Lockwood, PhD
Editorial Board Member
Communications Psychology
orcid.org/0000-0001-7195-9559

REVIEWER EXPERTISE:

Reviewer #1 stereotypes, social psychology,
Reviewer #2 stereotypes, social psychology, intersectionality

REVIEWER REPORTS:

Reviewer #1 (Remarks to the Author):

This manuscript presents a series of four studies examining how the stereotype congruency of advertisements for various services affects participants' attitudes toward the advertisements, their subjective economic value, and behavioral intentions. Importantly, the manuscript explores the impact of political ideology (US liberals versus conservatives) on these measures and its interaction with stereotype congruency. Studies 1-3 feature ads with racial minority characters, whereas study 4 features White characters and manipulates solely gender stereotypes. Using relatively large convenience samples, the manuscript reports consistent effects of an interaction between political ideology and stereotype congruency on attitudes, and variable effects on economic perceptions and behavioral hiring intentions. Overall, the manuscript features an interesting potential addition to the relatively small number of studies that focused on the impact of political ideology on attitudes toward stereotypical and counterstereotypical targets. However, some issues need to be addressed to ensure the interpretation, reliability, and robustness of the findings.

Major themes:

As the manuscript notes, the most consistent finding across all studies is the interaction between ideology and stereotypicality for the attitudinal dispositions outcome. However, unlike the authors' emphasis on the increased preference of conservatives for stereotype-congruent ads, the reported statistical results indicate that only liberals statistically favored stereotype-incongruent over stereotype-congruent ads. Similar interpretive issues repeat for the economic perceptions outcome, in which the manuscript claims liberals consistently preferred stereotype-incongruent ads even though only study 3 featured a relevant interaction and statistical difference in which conservatives valued stereotype-congruent over stereotype-incongruent ads. Put differently, the abstract and discussion feature an overinterpretation of descriptive findings and need to provide a more accurate description of the results.

In reviewing the studies, the most consistent finding across studies and stimuli seems to be that conservatives barely change their attitudes—the numerical values seem very similar across all studies (the same applies, albeit to a lesser extent, to conservatives' behavioral hiring intentions). Only liberals change their attitudinal perceptions across studies and conditions. This consistency is interesting in regard to current models and scholarly work and needs to be better addressed. The manuscript mentions all studies were pre-registered. However, the OSF link provides access to files that were uploaded on June 2024, well after all studies were completed, without any time stamps on any of the preregistration files (excluding Study 3). As such, it is impossible to know when the preregistration files were last modified. More importantly, all studies feature significant deviations from the preregistrations, effectively making all reported results non-preregistered. Of note, not each and every study needs to be preregistered for it to have value. Here are some of the most important deviations I identified:

- (1) Sample size. The pilot study preregistered a sample size is 300, but the data files contain 330 participants who completed the study in two waves, while the manuscript reports only 220 participants. Study 1's preregistered sample size is 300, but the total collected N is 377 with 303 who completed the study, and the same goes for the other studies.
- (2) All studies exclude participants with moderate political views without preregistering this exclusion and without a-priori defining how the exclusion would be determined (without mentioning the specific scoring computations and criteria).
- (3) All studies in the manuscript deviate from the preregistered analyses and use t-tests and post-hoc Tukey HSD instead of

the preregistered analyses.

(4) All preregistrations feature additional hypotheses that are never reported in the manuscript (pilot study: neutral ads; studies 1 and 2: a memory hypothesis and secondary analyses; studies 3 and 4: an expectation hypothesis and secondary analyses).

(5) In the pilot study, the preregistration reports the stereotype congruency measure to have 10 items (although 1 repeating), but the supplementary materials report using only 5 items for the composite measure of stereotype congruency.

(6) In study 1, the preregistration specifies an irrelevant power analysis with 3 groups and a one-way anova, whereas the main pre-registered analysis features only 2 groups (stereotype-congruent and stereotype-incongruent) and the hypotheses feature either a comparison between two groups (H1 & H3) or a 2 (political ideology) * 2 (stereotypicality) interaction.

(7) Uniquely to Study 2, it is unclear when and why the study teams decided to supplement the original data collection with a second wave. A month and a half passed between the first and the second wave, raising the concern that the additional round of data collection was initiated after the study team had already viewed the results, which would require an increased severity of the statistical tests (Lakens, 2023), for example by correcting the critical p of each of the statistical tests for multiple stops using Paugment (Sagarin et al., 2014). This is prudent for Study 2, given the small effect size observed.

(8) The preregistrations for studies 3-4, carrying the dates of 1/17 and 1/30 respectively, seem to have been completed only after data collection has already begun, 10/18 and 1/26, respectively. This violates the statements in the preregistration, saying “no data have been collected for this study”, again reducing the value of these preregistrations.

The manuscript reports findings for ads generated by the authors. Study 4 features stimuli in two conditions that are roughly equivalent in all features but the manipulated feature of gender: models are presented in the same pose, using relatively similar visual expressions, in similar environments and situations, et cetera. This is not the case for the stimuli used in studies 1-3. For example, the I.T. support ad in the congruent condition features an East Asian man sitting in what appears to be a concentrated facial expression, whereas the ad in the incongruent condition features a Black Woman standing and smiling, perhaps conveying more warmth and less competence (and thus reinforcing population-level stereotypes). Such differences could offer several alternative explanations to the obtained results. To test the impact of stereotype congruity more cleanly, the target ads need to more closely resemble each other across conditions, as they do in Study 4.

One of the potentially interesting findings in this manuscript pertains to the differences between the results obtained in study 4 compared to the other studies. However, given the incidental nature of this (not previously hypothesized) difference and all the previous comments, the most convincing way to demonstrate such a difference and to bolster the robustness of the other findings would be to run a new, truly preregistered, study. Such a study could include a 2 (political ideology) by 2 (stereotypicality) by 2 (racial identity of targets) between-participant experimental array using ad stimuli that resemble each other as much as possible on all dimensions. If properly preregistered (in line with the 8 comments on preregistration above, including hypotheses and analyses) and assuming an adequate sample size, such a study would significantly contribute to the demonstration of the reported effects.

On a technical note, the raw data on the OSF contain potentially identifiable data – they include IP addresses and geographical coordinates for many of the observations. These should be removed.

Contextualization of the study:

The Introduction section includes many news items (e.g., NPR, The Atlantic). Had such sources been included in addition to academic sources there would have been no issue, but they cannot be relied on as a primary source. Ample relevant studies exist.

The introduction says that “The theoretical and empirical attention given to stereotypes has historically been centered on implicit stereotypes (e.g., 20-21), and has slowly expanded to include focus on explicit stereotypes”. This is not an accurate description of the trajectory of research on stereotypes. The so-called “implicit revolution” in social cognition started in the ‘90s with the application of cognitive priming research to social constructs such as stereotypes and the introduction of implicit measures such as the implicit association test by Greenwald and Banaji. These measures were introduced to uncover perceptions that were hypothesized to be undetectable with explicit measures. Research up to that point focused exclusively on explicit stereotypes, perhaps starting with the classic Princeton trilogy (Katz & Braley 1933) and all the way to the more contemporary stereotype content model by Fiske and colleagues, with many fantastic contributions by many scholars. The authors may want to significantly edit this point.

The novelty of this work in examining the impact of political ideology on attitudes vis-à-vis perceptions of stereotype congruity and incongruity would be highlighted if relevant previous studies would be discussed more in-depth. For example, Stern et al. (2015) already characterized how liberals and conservatives differ in assessing stereotype-congruent and -incongruent stimuli (in a different context).

The introduction portrays liberals as advocating for progress and fighting social injustice, whereas conservatives favor tradition, which, at least the way the sentence is structured, is opposed to reducing social injustice (and hence, tradition is necessarily for social injustice). Some may say this is a biased characterization of conservative values and should be edited to reflect a more neutral view.

Technical issues/suggestions:

The provided data files, although reported as raw, seem to be minimally processed in some way. The timestamps for the data entries do not feature the expected chronological order in most of the files. Please provide fully raw data files (and, ideally, the code used to process them).

Given the unbalanced sample and exclusion rates, it is important to provide a full description of the number of participants allocated to each condition to reliably evaluate the robustness of the findings.

Relatedly, the study does not report the demographic characteristics of the samples. Although the study reports no a-priori hypothesis regarding the impact of such characteristics, the interpretation of the results would be vastly different if, for example, the samples were predominantly White versus a mixed racial/ethnic background. The same applies to the other demographic details collected, especially with regard to gender.

The manuscript reports the results of regression models in all studies. It might be interesting to include participants with

moderate political views in these analyses to examine the effect and potentially increase the statistical power, particularly if only liberals show a difference between the stereotypicality condition.

Reviewer #2 (Remarks to the Author):

Nature Communications Psychology

Liberals and conservatives respond divergently to stereotype portrayals of racial minorities and women

The authors set out to examine liberal and conservative reactions to stereotypical and counter-stereotypical portrayals of women and racial minorities. I had a few thoughts regarding the operationalization of stereotypical and counter-stereotypical portrayals, as well as a desire for a bit more initial setup of the research question. I hope my thoughts are constructive and help move their research forward.

One comment that kept arising for me was the conflation between increased diversity and counter-stereotypicality, or incongruity versus unaffiliated. The authors assumed that all diversity is inherently counter-stereotypical, when that simply can't be true. Just because I don't have a representation for all race/gender combinations does not make the representations I don't have inherently *counter-stereotypical*. For a stereotype to be counter-stereotypical, it should be explicitly counter to a known stereotype. For example, the authors use Asian women athletes as a counter-stereotypical exemplar. It seems as if the stereotype they are counter-acting is Asian women being nerds. But that feels like a stretch versus me simply not having a stereotypical representation of Asian women as athletes at all. They don't enter into the space as stereotypical or counter-stereotypical. I simply don't have a cognitive representation for them in that space. Relatedly, some of their examples were fascinatingly intersectional and other ones weren't. For example, there are several papers out there showing how Black women get a pass due to intersectional invisibility (Purdie-Vaughns & Eibach, 2008) when engaging in more dominant, masculine behaviors. To what extent is a Black woman software engineer seen as counter-stereotypical or there aren't strong representations due to the perceived masculinity of Black women (Hall et al., 2019; Livingston et al., 2012; Rosette et al., 2016)? These types of dynamics seem to be glossed over in the creation of the stimuli as well as in the authors' theorizing, but likely influenced results. Another example of intersectional dynamics that seemed to be missing in nuance is why the tutoring services modeled by White men were deemed incongruent. If tutoring is meant to be more feminine, might participants have read gayness onto the men? If so, the example is counter-stereotypical in one sense, but stereotypical in another. The domain versus the actor within the domain gets a bit muddled and confused in terms of this stereotypical versus counter-stereotypical language. Now to be fair, I know the authors pretested the stimuli by asking people "The way the models are depicted in the ads does not align with common stereotypes." I still feel the stimuli are not clear depictions of the stereotypical versus counter-stereotypical dimension the authors hoped to test. I can say that the models don't align with common stereotypes but that does not mean I believe they are counter-stereotypical.

Secondly, in the introduction, the authors note that there hasn't been much research on how "conservatives and liberals differ in their evaluations of traditional stereotypes." However, they don't fully motivate why we wouldn't just expect the opposite (their hypotheses feel almost like a tautology at this point) nor why we should study this beyond "it hasn't been studied yet". I don't find that all research that hasn't been done should be done, so I encourage the authors to motivate why this work is important to do.

Smaller points

- The sentence on page 2 is confusing as it relates to the "intensifying, negative spiral." The way it reads is that awareness reinforces systematic inequalities, which doesn't quite make sense.
- Also on page 2, the authors say that "review bombing" was part of negative backlash. But review bombing can also happen in a positive direction, where people purposefully flood a space with positive reviews to counteract backlash.
- A bit confused by the difference between Study 2 and Study 3. Is Study 3 the second wave of Study 2? How was Study 3 different from Study 2?
- Were the economic outcomes findings robust to outliers? I assume there were some with the large SDs.
- The authors measured political ideology using social and fiscal questions. Should we expect fiscal conservatives to react to stereotype differences by identity? It seems like we should only expect effects for social conservatives, not fiscal ones.

In response to the specific questions asked by NCP:

- Does the paper represent an advance in understanding which may influence thinking in the field? If you have concerns about the advance in relation to specific studies, we appreciate references to this work.
 - o It is unclear exactly how to think about where to go next with these findings. I didn't think it advanced theory as much as showed an addendum to a phenomenon we could have guessed.
- Does the article present an original study, new analysis, new model, or a direct or extended replication of previous work?
 - o Original study, although it's more of a conceptual replication of past work and adding in one additional moderator (e.g., stereotypical versus counter-stereotypical)
- Are the data and analysis technically sound? Are they appropriate to answer the research question, e.g., are causal research questions addressed on the basis of causal, rather than correlational evidence?
 - o The analyses are fine. I had some questions about the operationalization, as outlined above.
- Does the paper provide strong evidence for its conclusions?
 - o Yes

- Is the study question important to scientists for a sub-field of psychology?
 - o Yes. Stereotypes are hugely relevant for many sub-fields within psychology
- Are there any special ethical concerns arising from the use of animals or human subjects?
 - o Nope!
- Was the study preregistered and if so, did the authors follow the preregistration?
 - o They did pre-register their studies

Overall, love new work on stereotypes! The novelty and insights from the studies were difficult for me to discern. I believe the insights are there!

Hall, E. V., Hall, A. V., Galinsky, A. D., & Phillips, K. W. (2019). MOSAIC: A Model of Stereotyping Through Associated and Intersectional Categories. *Academy of Management Review*, 44(3), 643–672. <https://doi.org/10.5465/amr.2017.0109>

Livingston, R. W., Rosette, A. S., & Washington, E. F. (2012). Can an Agentic Black Woman Get Ahead? The Impact of Race and Interpersonal Dominance on Perceptions of Female Leaders. *Psychological Science*, 23(4), 354–358. <https://doi.org/10.1177/0956797611428079>

Purdie-Vaughns, V., & Eibach, R. P. (2008). Intersectional Invisibility: The Distinctive Advantages and Disadvantages of Multiple Subordinate-Group Identities. *Sex Roles*, 59(5–6), 377–391. <https://doi.org/10.1007/s11199-008-9424-4>

Rosette, A. S., Koval, C. Z., Ma, A., & Livingston, R. (2016). Race matters for women leaders: Intersectional effects on agentic deficiencies and penalties. *The Leadership Quarterly*, 27(3), 429–445. <https://doi.org/10.1016/j.leafqua.2016.01.008>

Reviewed by Sa-kiera Hudson, PhD
 Assistant Professor at University of California, Berkeley Haas School of Business
sa.kiera.hudson@berkeley.edu

I intentionally sign all my reviews and any and all typos can be blamed on COVID being an ongoing and exhausting pandemic, the systematic dismantling of LGBTQ rights, the weariness and heartbreak associated with constant bombing and killings, and police brutality in America. If my signature must be removed by journal policy, I request that the note [signature redacted] be included in its stead.

EDITORIAL POLICIES

We ask that you ensure your manuscript complies with our editorial policies and reporting requirements.

To that end, we require revised manuscripts to be accompanied by two completed items: a reporting summary that collects information on study design and procedure, and an editorial policy checklist that verifies compliance with all required editorial policies

- <https://www.nature.com/documents/nr-reporting-summary.zip>>Nature Research Reporting Summary
- <https://www.nature.com/documents/nr-editorial-policy-checklist.pdf>>Editorial Policy Checklist

All points on the policy checklist must be addressed. Your revised manuscript can only be sent back to the referees if these checklists are completed and uploaded with the revision.

Notes: If you have submitted a Stage 1 Registered Report, Review, Primer, Comment, or Perspective you do not need to submit these forms. If you have already submitted these forms, you may disregard this request.

** Visit Nature Research's author and referees' website at <http://www.nature.com/authors>>www.nature.com/authors for information about policies, services and author benefits**

Communications Psychology is committed to improving transparency in authorship. As part of our efforts in this direction, we are now requesting that all authors identified as 'corresponding author' create and link their Open Researcher and Contributor Identifier (ORCID) with their account on the Manuscript Tracking System prior to acceptance. ORCID helps the

scientific community achieve unambiguous attribution of all scholarly contributions. You can create and link your ORCID from the home page of the Manuscript Tracking System by clicking on 'Modify my Springer Nature account' and following the instructions in the link below. Please also inform all co-authors that they can add their ORCID to their accounts and that they must do so prior to acceptance.

Version 1:

Decision Letter:

Dear Ms Jiang,

Your manuscript titled "Liberals and conservatives respond divergently to stereotype portrayals of race and gender" has now been seen by our reviewers, whose comments appear below. In light of their advice I am delighted to say that we are happy, in principle, to publish a suitably revised version in Communications Psychology.

We therefore invite you to revise your paper one last time to address the remaining concerns of our reviewers and a list of editorial requests. At the same time we ask that you edit your manuscript to comply with our format requirements and to maximise the accessibility and therefore the impact of your work.

EDITORIAL REQUESTS:

SUBMISSION INFORMATION:

OPEN ACCESS:

*** TRANSPARENT PEER REVIEW:** Communications Psychology uses a transparent peer review system. On author request, confidential information and data can be removed from the published reviewer reports and rebuttal letters prior to publication. If you are concerned about the release of confidential data, please let us know specifically what information you would like to have removed. Please note that we cannot incorporate redactions for any other reasons.

*** CODE AVAILABILITY:** All Communications Psychology manuscripts must include a section titled "Code Availability" at the end of the methods section. We require that the custom analysis code supporting your conclusions is made available in a publicly accessible repository at this stage; please choose a repository that generates a digital object identifier (DOI) for the code; the link to the repository and the DOI must be included in the Code Availability statement. Publication as Supplementary Information will not suffice.

* DATA AVAILABILITY:

Link Redacted

Best regards,

Jennifer Bellingtier

Jennifer Bellingtier, PhD
Senior Editor
Communications Psychology

Patricia Lockwood, PhD
Editorial Board Member
Communications Psychology
orcid.org/0000-0001-7195-9559

REVIEWER EXPERTISE:

Reviewer #1 stereotypes, social psychology,
Reviewer #2 stereotypes, social psychology, intersectionality

REVIEWERS' COMMENTS:

Reviewer #1 (Remarks to the Author):

I was Reviewer #1 in the original submission.

The revised manuscript provides a very clear and robust set of results that are clearly described. I would like to applaud the authors for their efforts and their final outcome. Great job, I like this manuscript a lot, and will definitely cite it!

Two small issues that the authors might want to address before publication are"

(1) the "marginally significant" depiction should be corrected. Under NHST, results are either significant or insignificant. The manuscript could say that the descriptive direction is in line with, but it is not statistically significant.

(2) The discussion talks about the malleability of responses from conservatives, but the data suggest that liberals vary their responses more than conservatives. Across studies, the range of the mean response of conservatives is half of that of liberals:

Conservatives: 4.69 (Study 4, incongruent minority) to 5.1 (Study 3, White, congruent)

Liberals: 5.3 (Study 2, minority, incongruent) to 4.21 (Study 4, congruent White)

It would be interesting to discuss this point.

Reviewer 2 confirmed they had reviewed the revisions.

April 16, 2025

Dear Dr. Lockwood,

Thank you for inviting us to revise and resubmit our manuscript with an updated title, "Liberals and conservatives respond divergently to stereotype portrayals of race and gender" (COMMSPSYCHOL-24-0428A). We are grateful for the extensive feedback that you and the review team provided. We have revised the manuscript to address the comments raised by the expert reviewers.

In the remaining of this letter, we respond to each point made by all members of the review team, pointing out how we addressed each comment in our revised manuscript (and/or Supplementary Materials) or clarifying our reasoning. All of the comments from the review team appear in bold, and our responses appear in standard font.

Thank you for your patience during the peer-review process. Your manuscript titled "Liberals and conservatives respond divergently to stereotype portrayals of racial minorities and women" has now been seen by 2 reviewers, whose comments are appended below. You will see that they find your work of some potential interest. However, they have raised quite substantial concerns that must be addressed. In light of these comments, we cannot accept the manuscript for publication, but would be interested in considering a revised version that fully addresses these serious concerns.

RESPONSE: Thank you so much for finding our work interesting and for this opportunity to resubmit a revised manuscript and package of studies. We are very appreciative of your and the reviewers' time and feedback.

We hope you will find the Reviewers' comments useful as you decide how to proceed. Should additional work allow you to address these criticisms, we would be happy to look at a substantially revised manuscript. If you choose to take up this option, please highlight all changes in the manuscript text file, and provide a detailed point-by-point reply to the reviewers.

RESPONSE: In this response letter, we provide a reply to each of the points raised by the reviewers. We have bolded the original reviewer feedback, and our response is in standard text.

Editorially, we consider it important to conduct a fully pre-registered replication, as suggested by Reviewer 1. This would help to ensure the robustness of your findings and respond to Reviewer 1's concerns surrounding the existing pre-registrations. The policy for pre-registrations at Communications Psychology is as follows. Authors must disclose all deviations from the preregistered protocol and explain the rationale for deviation (e.g., flaw, feasibility, suboptimality). In cases of deviation from the preregistered analysis plan for reasons other than fundamental flaw or feasibility, the originally planned analyses must also be reported (see <https://www.nature.com/commpsychol/editorial-policies/preregistration-policy>).

RESPONSE: We agree that a fully pre-registered study with a three-way factorial design is necessary to evaluate the robustness of our findings. In this revision, we include a new and

highly-powered ($n = 2,484$) study specifying a 2 (political ideology: conservative, liberal) x 2 (stereotype portrayal: congruent, incongruent) x 2 (model race: minority, White) design to confirm two-way interactions and explore three-way interactions. Please see pages 10-14 in the manuscript for the results of this new study.

Additionally, for this additional study, following Review 1's recommendation, we used updated experimental stimuli with target advertisements that resemble each other as much as possible across conditions (see the images pasted in this Response Letter (Response Letter page 16), or Table 1 on pages 23-24 in the manuscript for those stimuli, and page 48 in the Supplementary Materials Appendix B subsection "Study 4 Supplementary Results" under "Pilot Testing Advertisement Stimuli for Study 4" for the stimuli pre-test results).

Overall, running preregistered/confirmatory two-way factorial ANOVAs and exploratory three-way factorial ANOVAs, followed by appropriate simple effects analyses, we find a replication of our results from our prior studies: First, attitudes are significantly impacted by the interaction effect of political ideology and stereotype portrayal, such that liberals prefer incongruent portrayals significantly more than congruent portrayals, conservatives prefer congruent portrayals significantly more than incongruent portrayals, and conservatives prefer congruent portrayals significantly more than liberals. Second, we found a significant interaction effect of political ideology and model race on attitudes, such that liberals prefer advertisements featuring only minority models significantly more than advertisements featuring only White models, conservatives prefer white models significantly more than minority models, liberals prefer minority models significantly more than conservatives, and conservatives prefer White models significantly more than liberals. Third, we found a significant three-way interaction effect of all three factors on attitudes.

Regarding our other measures, we found no significant main effects, two-way interaction effects, or three-way interaction effects on economic perceptions. For downstream hiring, we found no significant two-way or three-way interaction effects; we did find a significant main effect of political ideology and of stereotype portrayal on hiring, such that conservatives were significantly more likely to hire the stereotype congruent software candidate than liberals, and that participants in the congruent condition were significantly more likely than participants in the incongruent condition to hire the congruent candidate.

Taken together and in large part, these results replicate the results of Preliminary Study and Studies 1-3, thus strengthening our overall empirical contribution and the robustness of our findings.

Please note, since our initial submission, we have conducted a new three-factor experimental study based on your recommendation. With that new study, we have renamed the studies in our package as follows, with the provided rationale:

Response Letter Table RL1: Updated Study Names.

Initial Submission Study Names / #s	Revised Manuscript Study Names / #s	Rationale for Renaming
Pilot Study	Pilot Study	N/A
Study 1	Preliminary Study	This study is renamed to be Preliminary because it had a smaller sample size, measured attitudes, and the findings were replicated in Study 1. → Results are reported

		in Supplementary Materials Appendix B subsection "Preliminary Study: Testing Attitudes"
Study 2	Study 1	First main study with a complete suite of DVs: attitudes, \$, and hiring; sought large, balanced sample of Conservatives and Liberals.
Study 3	Study 2	This study maintains the complete suite of DVs with a large, balanced sample of participants.
Study 4	Study 3	This study maintains the complete suite of DVs with a large, balanced sample of participants.
N/A	Study 4	This is a new 2x2x2 study that did not exist in our initial submission. It also maintains the complete suite of DVs with a large, balanced sample.

I am also attaching a checklist that details critical reporting requirements for the revised manuscript. Please attend to each item and ensure your manuscript is fully compliant. We are requesting that your manuscript aligns with these requirements as this facilitates the evaluation of your manuscript, reducing delays in re-review and potential future acceptance. If your revised manuscript is not aligned with these requests on major issues, such as those concerning statistics, it may be returned to you for further revisions without re-review. Additional information can be found in our style and formatting guide Communications Psychology formatting guide.

If the revision process takes significantly longer than five months, we will be happy to reconsider your paper at a later date, provided it still presents a significant contribution to the literature at that stage.

Please use the following link to submit your

- revised manuscript,
- point-by-point response to the referees' comments,
- cover letter (as a separate document),
- the Editorial Policy Checklist (see below),
- the Reporting Summary (see below), and
- the completed Editorial Request Table (attached):

Thank you for the opportunity to review your work.

RESPONSE: Thank you so much for this opportunity to resubmit a revised manuscript and package of studies. We are very grateful for your and the reviewers' time and feedback.

Best regards,

Patricia Lockwood

Patricia Lockwood, PhD
Editorial Board Member
Communications Psychology
orcid.org/0000-0001-7195-9559

REVIEWER EXPERTISE:

Reviewer #1 stereotypes, social psychology,
Reviewer #2 stereotypes, social psychology, intersectionality

REVIEWER REPORTS:

Reviewer #1 (Remarks to the Author):

This manuscript presents a series of four studies examining how the stereotype congruency of advertisements for various services affects participants' attitudes toward the advertisements, their subjective economic value, and behavioral intentions. Importantly, the manuscript explores the impact of political ideology (US liberals versus conservatives) on these measures and its interaction with stereotype congruency. Studies 1-3 feature ads with racial minority characters, whereas study 4 features White characters and manipulates solely gender stereotypes. Using relatively large convenience samples, the manuscript reports consistent effects of an interaction between political ideology and stereotype congruency on attitudes, and variable effects on economic perceptions and behavioral hiring intentions. Overall, the manuscript features an interesting potential addition to the relatively small number of studies that focused on the impact of political ideology on attitudes toward stereotypical and counterstereotypical targets. However, some issues need to be addressed to ensure the interpretation, reliability, and robustness of the findings.

RESPONSE: Thank you so much for your time and thorough review of our initial submission. We are pleased you found our work interesting. We have seriously considered each of your points below and revised the manuscript and/or Supplementary Materials to address them, with particular attention to interpretation, preregistration transparency, and conducting an additional and preregistered 2x2x2 study following your recommendations. We believe these revisions thoroughly address your concerns.

Major themes:

As the manuscript notes, the most consistent finding across all studies is the interaction between ideology and stereotypicality for the attitudinal dispositions outcome. However, unlike the authors' emphasis on the increased preference of conservatives for stereotype-congruent ads, the reported statistical results indicate that only liberals statistically favored stereotype-incongruent over stereotype-congruent ads. Similar interpretive issues repeat for the economic perceptions outcome, in which the manuscript claims liberals consistently preferred stereotype-incongruent ads even

though only study 3 featured a relevant interaction and statistical difference in which conservatives valued stereotype-congruent over stereotype-incongruent ads. Put differently, the abstract and discussion feature an overinterpretation of descriptive findings and need to provide a more accurate description of the results.

RESPONSE: Thank you for raising this. After conducting our new three-way experimental study (more details on that in the relevant responses below), we have gone through our entire manuscript to thoroughly portray our interpretations and discussion of our results accurately. In each of our brief study discussions embedded within the Results section, we are careful to speak only to the pairwise differences that we explicitly test to probe significant interaction effects. As such, we have removed our inaccurate representations that conservatives preferred congruent more than incongruent portrayals, which was indeed overclaiming based on the significant interaction in general, rather than on specific pairwise tests.

To start, we have moved Preliminary Study (previously “Study 1” – please see page 7 in this response letter for more details about the renaming of our study package and Response Letter Table RL1) to the Supplementary Materials, as this was our first instance of overinterpretation in the original manuscript. Preliminary Study tested our attitudinal index on a smaller sample size and found a significant cross-over interaction but no significant pairwise tests, perhaps because it was statistically underpowered to detect such differences. Therefore, our claim that conservatives preferred congruent more than incongruent was an overinterpretation based on the cross-over interaction itself, with no empirical support in terms of pairwise tests. We have removed that overinterpretation from our Preliminary Study results discussion in Supplementary Materials Appendix B subsection “Preliminary Study: Testing Attitudes” (see page 41-42 for those results).

Further, for Study 1 (previously “Study 2”) attitudes, we write: *“Exploratory post-hoc TukeyHSD tests showed a significant difference between liberals evaluating congruent versus incongruent portrayals ($p < .05$), and a significant difference between liberal and conservative responses to incongruent portrayals ($p < .001$), such that liberals preferred incongruent portrayals significantly more than congruent portrayals, and liberals favored stereotype incongruent portrayals significantly more than conservatives”* (manuscript page 6), which accurately interprets and presents our results focused on the significant pairwise results for liberals without overclaiming on conservatives.

We do this again for Study 2 (previously “Study 3”) attitudes, *“Post-hoc TukeyHSD tests showed a significant difference between liberals and conservatives in general ($p < .001$) and between liberals and conservatives when evaluating incongruent portrayals ($p < .001$), revealing again that liberals favored stereotype incongruent portrayals significantly more than conservatives”* (manuscript page 6), underscoring the results that liberals prefer incongruent over congruent ads, and that the reverse preference for conservatives is not supported by our results.

We continue to do this for Study 2 and Study 3 results discussions, as well as in our general discussion and abstract. Our revised manuscript now avoids these overinterpretations.

In reviewing the studies, the most consistent finding across studies and stimuli seems to be that conservatives barely change their attitudes—the numerical values seem very similar across all studies (the same applies, albeit to a lesser extent, to conservatives’ behavioral hiring intentions). Only liberals change their attitudinal perceptions across studies and conditions. This consistency is interesting in regard to current models and scholarly work and needs to be better addressed.

RESPONSE: This observation is a fair one, especially when considering new Studies 1 & 2. In these first two studies (renamed from Studies 2 & 3 in initial submission, please see more details in the next response), both conservatives' and liberals' attitudes are very consistent; we replicate our findings. Starting in Study 3 that features only White models, we see that conservatives' attitudes increase in both congruent and incongruent conditions, with a particular spike of +0.20 attitudinal points in the congruent condition. To your observation, liberals also change their attitudes with a significant drop in ratings across both conditions. Then, in Study 4, we see conservatives' attitudes in the incongruent condition drop quite significantly from Studies 1 & 2's baseline, and liberals' attitudes across both conditions change meaningfully from Studies 1 & 2's baseline. This is all to say that, while liberals change their attitudinal perceptions across conditions a couple times more saliently than conservatives, it is not the case that conservatives do not change their attitudes at all. Indeed, we find in Studies 3 & 4 that conservatives are particularly sensitive to the racial representations featured in the target advertisements, and that previous studies' patterns are flipped in these studies. Further, we are primarily interested in comparing responses between liberals and conservatives in any given study, rather than comparing responses within liberals or within conservatives across studies, as we cannot control for timing effects, news cycle effects, etc. that may be present at the time that we ran any particular study.

The manuscript mentions all studies were pre-registered. However, the OSF link provides access to files that were uploaded on June 2024, well after all studies were completed, without any time stamps on any of the preregistration files (excluding Study 3). As such, it is impossible to know when the preregistration files were last modified. More importantly, all studies feature significant deviations from the preregistrations, effectively making all reported results non-preregistered. Of note, not each and every study needs to be preregistered for it to have value. Here are some of the most important deviations I identified:

RESPONSE: Thank you for highlighting the importance of this. We recognize that we did not follow best practices at the time of initiating our first studies, but over time we learned those best practices and duly implemented them (Study 2, Study 3, and new Study 4 all have AsPredicted preregistrations).

For our earlier studies (before Study 2), we used the honor system of dating our preregistration Word document with the date that we finalized the document. We then uploaded to our OSF repository as we prepared our manuscript for submission, thinking that the Word document metadata would provide transparency. However, as noted, we recognize that this was not best practice and have since improved our transparency in subsequent and the majority of studies in this package using clearly timestamped AsPredicted PDF documents (Study 2, Study 3, and new Study 4). For instance, in our new Study 4, we adhere to the standards of data and preregistration transparency by using only AsPredicted preregistration with clear time stamps, planned participant recruitment methods, and specifying confirmatory versus exploratory analyses; further, data collection did not occur before preregistration was registered on AsPredicted. We would also like to note that for none of our studies did data collection begin before preregistration; the start dates in the raw data files for each study are indeed after the date listed on preregistration documents (Word document and/or AsPredicted).

Next, we address in detail each of the comments you raise below.

Please note, since our initial submission, we have conducted a new three-factor experimental study based on your recommendation. With that new study (Study 4), we have renamed the studies in our package as follows, with the provided rationale:

Response Letter Table RL1: Updated Study Names.

Initial Submission Study Names / #s	Revised Manuscript Study Names / #s	Rationale for Renaming
Pilot Study	Pilot Study	N/A
Study 1	Preliminary Study	This study is renamed to be Preliminary because it had a smaller sample size, measured attitudes, and the findings were replicated in Study 1. → Results are reported in Supplementary Materials Appendix B subsection “Preliminary Study: Testing Attitudes”
Study 2	Study 1	First main study with a complete suite of DVs: attitudes, \$, and hiring; sought large, balanced sample of Conservatives and Liberals.
Study 3	Study 2	This study maintains the complete suite of DVs with a large, balanced sample of participants.
Study 4	Study 3	This study maintains the complete suite of DVs with a large, balanced sample of participants.
N/A	Study 4	This is a new 2x2x2 study that did not exist in our initial submission. It also maintains the complete suite of DVs with a large, balanced sample.

(1) Sample size. The pilot study preregistered a sample size is 300, but the data files contain 330 participants who completed the study in two waves, while the manuscript reports only 220 participants. Study 1’s preregistered sample size is 300, but the total collected N is 377 with 303 who completed the study, and the same goes for the other studies.

RESPONSE: We report exclusions and final sample sizes in Supplementary Materials Appendix A: Study Design Details and Sample Information (see pages 31-40). As noted in Appendix A subsection “Data Collection,” we recruit more participants than we need because we expect certain exclusion criteria to apply to the raw participant data (e.g., incorrect responses to attention check questions, participants who leave the survey open but do not finish the survey, duplicate participants) (see Data Collection subsection on page 31-32). Additionally, the raw participant sample includes participants that we are not targeting, such as those with moderate political views, who are subsequently excluded from analysis. We provide clear details on the number of participants that are excluded for each specified criteria (see “Data Exclusions” subsection on page 33-35) and provide a table of sample size condition counts (see Supplementary Table S1 on page 36) for every study in our package.

(2) All studies exclude participants with moderate political views without preregistering this exclusion and without a-priori defining how the exclusion would be determined (without mentioning the specific scoring computations and criteria).

RESPONSE: We did not explicitly or verbatim preregister the exclusion of participants with moderate political views because our preregistrations represent our focus on political ideology operationalized as two groups.

In our preregistrations, starting with Preliminary Study and throughout the rest of the main studies (Studies 1-3), we state our research questions and hypotheses with a focus on “politically conservative” and “politically liberal / progressive” participants. In these preregistrations, we also note clearly in the sample size subsection of the preregistration documents that we are interested in recruiting “ $n = \#$ conservative” and “ $n = \#$ liberal” participants, thus reiterating our focus on political ideology operationalized as two groups.

In our new study (Study 4), we explicitly add a statement to our preregistration’s exclusions subsection that we will exclude political moderates or independents: “For exclusions, we will exclude from analysis participants who are not strong in their political ideology (e.g., moderates or independents), participants who do not complete the survey, and participants who fail attention check questions.”

Finally, our theory and overall framing is focused on the differences between these two groups; we specifically recruit along these two ideological poles and our studies’ samples reflect our population of interest. In other words, we are not interested in moderates, independents, or the middle group between the two ideological groups. We are consistent across our studies on research question focus, independent variable operationalization, sample size recruitment, and theoretical interest centered on conservatives versus liberals. We report our exclusions of moderates and independents in the Supplementary Materials Appendix B “Index Measures – Political Ideology” (page 42) and Supplementary Materials Appendix A “Data Exclusions” (pages 33-35). We also provide exploratory linear regressions including political moderates in response to your final feedback point in this response letter (page 27 in this response letter).

(3) All studies in the manuscript deviate from the preregistered analyses and use t-tests and post-hoc Tukey HSD instead of the preregistered analyses.

RESPONSE: Given the interconnected nature of your feedback bullets (3) and (4), we provide our full response to both of these points under bullet (4).

(4) All preregistrations feature additional hypotheses that are never reported in the manuscript (pilot study: neutral ads; studies 1 and 2: a memory hypothesis and secondary analyses; studies 3 and 4: an expectation hypothesis and secondary analyses).

RESPONSE: This response addresses your feedback points (3) and (4) in a consolidated list.

We go through each of our original studies in this package, from Preliminary Study through Study 3, and revise the manuscript and/or add information into the Supplementary Materials to directly report preregistered analyses and provide complete hypothesis testing.

Below, we present a detailed study-by-study list. Each study lists the results that are reported, where in the manuscript or Supplementary Materials the results are reported, whether any

deviations are present between the preregistered analyses (and how we rectified this), and complete hypothesis testing for each study.

Preliminary Study

1. Confirmatory/preregistered omnibus one-way ANOVA testing for main effect of stereotype portrayal on attitudes (H1) was previously not reported and is now reported in Supplementary Materials Appendix B subsection “Preliminary Study: Testing Attitudes” on page 41:
 - *“Following our pre-registration, we first conducted a one-way ANOVA to test for a main effect of stereotype portrayal on attitudes and found no significant effect ($M_{cong} = 5.24$, $SD_{cong} = 1.02$, $M_{incong} = 5.24$, $SD_{incong} = 0.96$; $F_{(1, 194)} = 0.00$, $p = .952$). We find no support for our prediction that all consumers will prefer stereotype congruent portrayals more than stereotype incongruent portrayals.”*
2. Factorial ANOVA testing for interaction effect of stereotype portrayal and political ideology on attitudes (H2) is reported in Supplementary Materials Appendix B subsection “Preliminary Study: Testing Attitudes” on page 41-42. We note that the linear regressions and post-hoc Tukey HSD tests are exploratory:
 - *“Next, we ran a factorial ANOVA testing for an interaction effect of stereotype portrayal and political ideology on attitudes. Our results show a significant interaction effect ($M_{cons*cong} = 5.50$, $SD_{cons*cong} = 1.17$, $M_{cons*incong} = 5.03$, $SD_{cons*incong} = 1.42$, $M_{libs*cong} = 5.12$, $SD_{libs*cong} = 0.94$, $M_{libs*incong} = 5.34$, $SD_{libs*incong} = 0.59$; $F_{(1, 193)} = 5.39$, $p = .02$, $\eta^2_p = .03$). These results were robust in exploratory linear regressions with and without holding constant participants’ age, race, gender, income, and educational attainment ($b = -0.68$, $SE = 0.30$, $p = .02$; $b = -0.70$, $SE = 0.30$, $p = .02$). Conducting an exploratory post-hoc Tukey’s Honest Significant Differences (Tukey HSD) multiple pair-wise difference test, we found no significant pairwise differences, suggesting that our sample size is limited or the interaction effect is small despite being significant at $p < .05$. We address sample size limitations in our main studies, Studies 1-4. This interaction effect provides support for our prediction that political ideology will moderate attitudinal ratings of the stereotype congruent and incongruent ads.”*
 - Please note: This factorial ANOVA was not a preregistered analysis explicitly stated in the preregistration, but it tests a preregistered hypothesis (H2). Therefore, we do not present this analysis as exploratory, but we do note that the additional linear regression and TukeyHSD tests are exploratory.
3. One-way ANOVA testing for main effect of stereotype portrayal on time spent viewing advertisements (H3) is reported in Supplementary Materials Appendix B subsection “Preliminary Study: Testing Attitudes” on page 42:
 - *“Finally, we conducted a one-way ANOVA to test for a main effect of stereotype portrayal on time spent viewing the target advertisements, having predicted that stereotype incongruent ads will be more surprising (i.e., will be viewed for longer) than stereotype congruent ads. We found no differences in time spent (in seconds) viewing target advertisements between stereotype portrayal conditions ($M_{cong} = 693$, $SD_{cong} = 256$, $M_{incong} = 650$, $SD_{incong} = 280$; $F_{(1, 194)} = 1.25$, $p = .264$), thus failing to provide support for this prediction.”*

Study 1

1. Confirmatory/preregistered one-way ANOVA testing for a main effect of stereotype portrayal on attitudes (H1). This analysis, which was previously not reported, has been added to the Supplementary Materials Appendix B subsection “Study 1 Supplementary Results” under “Secondary Results” on page 43-44:

- *“Following our preregistration and to replicate Preliminary Study, we conduct a one-way ANOVA testing for a main effect of stereotype portrayal on attitudes. As in Preliminary Study, we find no significant differences in attitudes between stereotype portrayal conditions ($M_{cong} = 5.02$, $SD_{cong} = 1.00$, $M_{incong} = 5.09$, $SD_{incong} = 0.98$; $F_{(1, 956)} = 1.32$, $p = .252$) and fail to support our prediction that participants would overall favor stereotype congruent portrayals over stereotype incongruent portrayals.”*
2. Confirmatory/preregistered factorial ANOVA testing for an interaction effect of stereotype portrayal and political ideology on attitudes (H2). This was and remains reported in the manuscript on page 6. We have revised the reporting to underscore the confirmatory nature of this factorial ANOVA and the exploratory nature of the linear regression and post-hoc Tukey HSD follow-up tests:
 - *“Running a confirmatory factorial ANOVA, we found a significant interaction effect of stereotype portrayal and political ideology on attitudes ($M_{cons*cong} = 4.91$, $SD_{cons*cong} = 1.05$, $M_{cons*incong} = 4.85$, $SD_{cons*incong} = 1.12$, $M_{libs*cong} = 5.10$, $SD_{libs*cong} = 0.95$, $M_{libs*incong} = 5.29$, $SD_{libs*incong} = 0.80$; $F_{(1,954)} = 3.98$, $p < .05$, $\eta^2_p < .01$) (see Fig. 1A). This result held in an exploratory linear regression ($b = -0.25$, $SE = 0.13$, $p = .046$) which was marginally significant when holding participants’ age, race, gender, income, and educational attainment constant ($b = -0.22$, $SE = 0.12$, $p = .075$). Exploratory post-hoc TukeyHSD tests showed a significant difference between liberals evaluating congruent versus incongruent portrayals ($p < .05$), and a significant difference between liberal and conservative responses to incongruent portrayals ($p < .001$), such that liberals preferred incongruent portrayals significantly more than congruent portrayals, while conservatives favored stereotype congruent portrayals significantly more than liberals.”*
 3. Confirmatory/preregistered one-way ANOVA testing for a main effect of political ideology on downstream hiring (H4a). We replaced the t -test ($t_{(956)} = 5.39$, $p < .001$, Cohen’s $d = .35$) in the manuscript with confirmatory ANOVA results, see page 8 in the manuscript:
 - *“However, following our preregistration, we test for and found a significant main effect by political ideology ($F_{(1, 956)} = 29.10$, $p < .001$, $\eta^2_p = .03$), showing that conservative participants ($M_{cons} = 0.40$, $SD_{cons} = 0.49$) were more likely than liberal participants ($M_{libs} = 0.24$, $SD_{libs} = 0.43$) to choose the stereotype congruent Asian Male candidate (see Fig. 3A).”*
 4. One-way ANOVA testing for a main effect of stereotype portrayal on downstream hiring (H4b). We added this preregistered analysis to the Supplementary Materials Appendix B subsection “Study 1 Supplementary Results” under “Secondary Results” on page 44:
 - *“We also conduct a confirmatory one-way ANOVA testing for a main effect of stereotype portrayal on downstream hiring, having predicted a significant effect. We found no significant differences in downstream hiring selections between stereotype conditions ($M_{cong} = 0.31$, $SD_{cong} = 0.46$, $M_{incong} = 0.30$, $SD_{incong} = 0.46$; $F_{(1, 956)} = 0.04$, $p = .842$), thus failing to support this prediction.”*
 - Please note: This was not a preregistered analysis explicitly stated in the preregistration, but it tests a preregistered hypothesis (H4b). Therefore, we do not present this analysis as exploratory.
 5. Factorial ANOVA testing for an interaction effect of stereotype portrayal and political ideology on downstream hiring (H4c). This was and remains reported in the main text on page 8:
 - *“In Study 1, we found no interaction effect of political ideology and stereotype portrayal on job candidate choice ($M_{cons*cong} = 0.40$, $SD_{cons*cong} = 0.49$, $M_{cons*incong} = 0.39$, $SD_{cons*incong} = 0.49$, $M_{libs*cong} = 0.24$, $SD_{libs*cong} = 0.43$, $M_{libs*incong} = 0.23$, $SD_{libs*incong} = 0.42$; $F_{(1,954)} < 0.01$, $p = .98$).”*

- Please note: This was not a preregistered analysis explicitly stated in the preregistration, but it tests a preregistered hypothesis (H4c). Therefore, we do not present this analysis as exploratory.

Study 2

1. Confirmatory/preregistered factorial ANOVAs testing for interaction effect of stereotype portrayal and political ideology on outcomes. These were and remain reported in the main text on pages 6-10. No revisions were made to these results reporting, as no preregistration deviations exist.
2. Mediation analysis using PROCESS testing for moderated mediation, with mediator specified as expectation violation. These are newly added to the Supplementary Materials Appendix B, in subsection “Study 2 Supplementary Results” under “Mediation Results” and “Expectation Index” on page 45-47. We include Supplementary Figure S1 (Moderated Mediation Model) and Supplementary Table S9 (Expectation Index Question Items) on page 46-47.
 - a. *“In Study 2, we consider why perceptions diverged on the basis of political ideology and stereotype portrayal. We collected additional measures on expectations and norms (see Supplementary Table S9 for items). These measures were administered in the procedures of Study 2 immediately following the advertisements and prior to our dependent measures. Running 5,000 bootstrapped samples using the standard PROCESS Model 8 script in R (Hayes, 2017), we fitted a moderated-mediation model specifying stereotype portrayal as the independent variable, political ideology as the moderator, expectations as the mediator, and attitudinal perceptions as the outcome variable.*

We found that stereotype portrayal was a positive predictor of expectations ($b = 0.15$, $SE = 0.06$, $t = 2.35$, $p = .02$) but not attitudinal perceptions ($p = .42$), suggesting full mediation. Expectations ($b = 1.00$, $SE = 0.03$, $t = 30.51$, $p < .001$) was a positive predictor of attitudinal perceptions. Political ideology was a negative moderator of the link between stereotype portrayal and expectations ($b = -0.23$, $SE = 0.10$, $t = -2.28$, $p = .02$), but not of the link between stereotype portrayal and attitudinal perceptions ($p = .46$). In support of a moderated mediation, there was a significant indirect effect of stereotype portrayal through expectations on attitudinal perceptions conditional on political ideology (index = -0.23 , $SE = 0.11$, $CI [-0.44; -0.03]$), driven by a significant indirect effect among liberals ($b = 0.15$, $SE = 0.06$, $CI [0.03, 0.27]$) rather than among conservatives ($b = -0.08$, $SE = 0.09$, $CI [-0.25, 0.09]$). Figure S1 illustrates the moderated mediation model.

The moderated mediation also held when specifying economic perceptions as the outcome variable. Stereotype portrayal was a positive predictor of expectations, but not economic perceptions ($p = .17$). Expectations ($b = 36.62$, $SE = 5.44$, $t = 6.73$, $p < .001$) was a positive predictor of economic perceptions. Political ideology was a negative moderator of the link between stereotype portrayal and expectations, but not of the link between stereotype portrayal and economic perceptions ($b = -3.84$, $SE = 16.48$, $t = -0.23$, $p = .82$). Further, there was significant indirect effect of stereotype portrayal through expectations on economic perceptions at different levels of political ideology (index = -8.41 , $SE = 4.06$, $CI [-17.20; -1.05]$), with a significant indirect effect among liberals (index = 5.43 , $SE = 2.25$, $CI [1.32; 10.15]$) and a null indirect effect among conservatives (index = -2.98 , $SE = 3.31$, $CI [-9.74; 3.23]$) (see Figure S1).

Finally, when specifying behavioral selection as the outcome variable, the moderated mediation did not hold, as stereotype portrayal was a positive predictor of both expectations and behavioral selection ($b = -0.42$, $SE = 0.20$, $Z = -2.12$, $p = .03$). The index of moderated mediation also showed null significance (index = -0.03 , $SE = 0.03$, $CI [-0.09; 0.02]$).

- b. Please note: We had removed the reporting of the moderation mediation results after friendly feedback with colleagues who noted the mediation distracts from the main contribution of the paper (that of the interaction effect on outcomes).

Study 3

1. Confirmatory/preregistered factorial ANOVAs testing for interaction effect of stereotype portrayal and political ideology on outcomes. These were and remain reported in the main text on pages 8-10. No revisions were made to these results reporting, as no preregistration deviations exist.
2. Mediation analysis using PROCESS testing for moderated mediation, with mediator specified as expectation violation. These are newly added to the Supplementary Materials Appendix B, in subsection "Study 3 Supplementary Results" under "Mediation Results" on page 47:
 - a. *"We replicated our moderated mediation model for attitudinal perceptions from Study 2. Stereotype portrayal was a positive predictor of expectations ($b = 0.37$, $SE = 0.08$, $t = 4.96$, $p < .001$) and of attitudinal perceptions ($b = 0.24$, $SE = 0.07$, $t = 3.25$, $p = .001$). Expectations ($b = 0.90$, $SE = 0.04$, $t = 25.64$, $p < .001$) was a positive predictor of attitudinal perceptions. Political ideology was a negative moderator of the link between stereotype portrayal and expectations ($b = -0.81$, $SE = 0.11$, $t = -7.40$, $p < .001$), but not of the link between stereotype portrayal and attitudinal perceptions ($p = .52$). In further support of a moderated mediation, there was a significant indirect effect of stereotype portrayal through expectations on attitudinal perceptions conditional on political ideology index = -0.72 , $SE = 0.10$, $CI [-0.92; -0.53]$, driven by a significant indirect effect among both liberals ($b = 0.34$, $SE = 0.06$, $CI [0.21, 0.26]$) and conservatives ($b = -0.39$, $SE = 0.08$, $CI [-0.54, -0.24]$). Figure S1 illustrates the moderated mediation model. We found no significant moderated mediation for economic perceptions or for behavioral selections."*
 - b. Please note: We had removed the reporting of the moderation mediation results after friendly feedback with colleagues who noted the mediation distracts from the main contribution of the paper (that of the interaction effect on outcomes).

(5) In the pilot study, the preregistration reports the stereotype congruency measure to have 10 items (although 1 repeating), but the supplementary materials report using only 5 items for the composite measure of stereotype congruency.

RESPONSE: Thank you for raising this. Firstly, we note that we had a typo in the Pilot preregistration document, including listing the first item twice and not listing an item that was used. We created the Cronbach measure for stereotype alignment based on 5 items, which are listed below.

- 1) These advertisements illustrate general stereotypes that are commonly held by society.
- 2) The way the models are depicted in the ads does not align with common stereotypes. (reverse-coded)
- 3) These ads are unconventional. (reverse-coded)

- 4) The way the models are depicted in the ads matches common stereotypes.
- 5) The ads do not portray common stereotypes. (reverse-coded)

We did not include the items from the pre-registration that specifically focused on gender stereotypes, race stereotypes, or neutral stereotypes, as wanted to focus on congruent versus incongruent intersectional stereotypes. We were consistent in using the same 5-item stereotype alignment index in a subsequent pilot test for the three-factor design used in Study 4, which featured updated target advertisements that followed your suggestions on improving similarity in visual expression, body language, and context across conditions.

(6) In study 1, the preregistration specifies an irrelevant power analysis with 3 groups and a one-way anova, whereas the main pre-registered analysis features only 2 groups (stereotype-congruent and stereotype-incongruent) and the hypotheses feature either a comparison between two groups (H1 & H3) or a 2 (political ideology) * 2 (stereotypicality) interaction.

RESPONSE: For Preliminary Study (previously “Study 1”), this G*Power number of groups input of 3 was incorrectly inputted. Using the correct number of 2 groups, we find we would need at least 200 participants (screenshot below). Our clean sample of $n = 196$ in the Preliminary Study is very close to achieving this. For all main studies (Studies 1-4), we provide sample size calculations and preregistered recruitment goals on page 31-32 in the Supplementary Materials Appendix A subsection “Data Collection.”

(7) Uniquely to Study 2, it is unclear when and why the study teams decided to supplement the original data collection with a second wave. A month and a half passed between the first and the second wave, raising the concern that the additional round of data collection was initiated after the study team had already viewed the results, which would require an increased severity of the statistical tests (Lakens, 2023), for example by correcting the critical p of each of the statistical tests for multiple stops using Paugment (Sagarin et al., 2014). This is prudent for Study 2, given the small effect size observed.

RESPONSE: For Study 1 (previously named “Study 2”), we conducted Wave 1 in March 2023. Wave 1 had a preregistered *a priori* desired sample size of $n = 800$. We fell short of this sample size, recruiting $n = 684$ clean participants. We then submitted a preregistration to boost our liberal and conservative participants by a total of $n = 200$, as we hoped to get closer to our Wave 1 desired sample size of 800 total. We conducted Wave 2 in May 2023. The reason we conducted Wave 2 was to obtain the clean $N = 800$ across both studies (the $N = 800$ that we desired in our original Study 1 preregistration, having fallen short in the first round of recruitment). In Wave 2’s preregistration, we note that we will merge Wave 1’s clean data with Wave 2’s clean data, using an approach similar to mini meta-analysis (e.g., Goh et al., 2016).

Following your and Sagarin et al.’s guidance (2014), we add additional reporting of $p_{\text{augmented}}$ to Study 1 results (page 44 in the updated Supplementary Materials Appendix B, subsection “Study 1 Supplementary Results” under “ $p_{\text{augmented}}$ Results for Main Analyses”):

$p_{\text{augmented}}$ Results for Main Analyses

In Study 1, we conducted two pre-registered waves of participant recruitment using a mini meta-analysis approach in merging the data (Goh et al., 2016). Following Sagarin et al.’s guidance regarding sample augmentation (2014), we calculate and report $p_{\text{augmented}}$ for Study 1’s results below.

Attitudes. After running the first wave of participants (Study 1 $n_{\text{Wave1}} = 684$), we found no significant interaction effect of political ideology and stereotype portrayal on attitudes ($F_{(1, 680)} = 3.24, p = .072, \eta^2_p < .01$). Please note n_{Wave1} did not meet our *a priori* desired sample of 800 participants. We conducted a preregistered second wave of recruitment to increase the number of politically conservative and politically liberal participants and to achieve our desired sample size. With Wave 1 and Wave 2 ($n_{\text{Wave2}} = 274$), we obtained a full Study 1 sample of $n = 958$. For the full sample, the interaction effect was significant ($F_{(1, 954)} = 3.98, p = .046, \eta^2_p < .01$). Following Sagarin et al.’s guidance (2014), we calculated $p_{\text{augmented}}$ for the full sample and found $p_{\text{augmented}} = [.055, .072]$; Sagarin et al. note that $p_{\text{augmented}}$ range will always exceed .05.

Economic Perceptions. For the first wave of participants, we found a significant interaction effect of political ideology and stereotype portrayal on economic perceptions ($F_{(1, 680)} = 8.45, p = .004, \eta^2_p = .01$). For the full sample, the interaction effect remained significant ($F_{(1, 954)} = 6.23, p = .013, \eta^2_p < .01$). We calculated $p_{\text{augmented}}$ for the full sample and found $p_{\text{augmented}} = [.050, .053]$.

Downstream hiring. Running the first wave of participants, we found no significant interaction effect of political ideology and stereotype portrayal on downstream hiring ($F_{(1, 680)} < 0.01, p = .99$). For the full sample, the interaction effect was not significant ($F_{(1, 954)} < 0.01, p = .98$).

Following our preregistration, we test for a significant main effect by political ideology and found significant effect in our Wave 1 sample ($t_{(682)} = 5.16, p < .001$, Cohen’s $d = .40$) and in our full sample ($t_{(956)} = 5.39, p < .001$, Cohen’s $d = .35$). We calculated $p_{\text{augmented}}$ for the full sample and found $p_{\text{augmented}} = [.050, .050]$.”

We note these additional results reporting in the main text with Footnote 3 on page 6 in the manuscript: “We conducted two pre-registered waves of participant recruitment using a mini

meta-analysis approach in merging the data (Goh et al., 2016) (35). Following Sagarin et al.'s guidance regarding sample augmentation (2014) (36), we calculated and report augmented for Study 1's results in the Supplementary Materials Appendix B subsection "Study 1 Supplementary Results.""

(8) The preregistrations for studies 3-4, carrying the dates of 11/7 and 1/30 respectively, seem to have been completed only after data collection has already begun, 10/18 and 1/26, respectively. This violates the statements in the preregistration, saying "no data have been collected for this study", again reducing the value of these preregistrations.

RESPONSE: Thank you for bringing this to our attention.

We are not certain as to why Study 2 (previously "Study 3") was preregistered on AsPredicted *after* data collection had already begun. Our best guess is that we had preregistered using a Word document (following our approach used in previous studies and therefore indeed had *not* collected data for the study at the time of preregistration) and later learned about the best practice of using AsPredicted and created the preregistration on AsPredicted as soon as we learned to do so.

For Study 3 (previously "Study 4"), data collection did *not* occur before preregistration: we have an AsPredicted timestamp for Study 4 of January 18, 2024 and the raw data files show that data began collection on January 26, 2024. To your point, we did supplement the AsPredicted preregistration with a Word document preregistration uploaded to OSF on January 30, 2024; this was to specify a greater sample size and a potential mediator. You may notice in the raw data for Study 3 that data was collected on January 26, 2024 (survey test with n=20 participants to check survey flow) and then data collection was paused until February 12, 2024, which is after the addendum preregistration was uploaded to OSF.

Overall, we agree with your concerns about preregistration. We address these in our new Study 4, which adheres to the standards of data and preregistration transparency by using only AsPredicted preregistration with clear time stamps, planned participant recruitment methods, and specifying confirmatory versus exploratory analyses; further, data collection did not occur before preregistration was registered on AsPredicted.

The manuscript reports findings for ads generated by the authors. Study 4 features stimuli in two conditions that are roughly equivalent in all features but the manipulated feature of gender: models are presented in the same pose, using relatively similar visual expressions, in similar environments and situations, et cetera. This is not the case for the stimuli used in studies 1-3. For example, the I.T. support ad in the congruent condition features an East Asian man sitting in what appears to be a concentrated facial expression, whereas the ad in the incongruent condition features a Black Woman standing and smiling, perhaps conveying more warmth and less competence (and thus reinforcing population-level stereotypes). Such differences could offer several alternative explanations to the obtained results. To test the impact of stereotype congruity more cleanly, the target ads need to more closely resemble each other across conditions, as they do in Study 4.

RESPONSE: Thank you for raising this. Our new 3-way interaction study (Study 4) incorporates this feedback. We found images that control for visual expressions, body language, and environments as best as possible, given the constraints of using available stock images, using only real human images, and excluding use of any AI-generated images. This new set of

experimental stimuli were selected to enhance similarity in model facial expression, body language, and environment across conditions. Please see below for these images (they are also provided in Table 1 of the manuscript, on pages 23-24).

Study 4 Experimental Stimuli – Minority Model Race
Stereotype Congruent (L), Stereotype Incongruent (R)

Study 4 Experimental Stimuli – White Model Race
Stereotype Congruent (L), Stereotype Incongruent (R)

Further, we pilot tested these experimental stimuli for realism and stereotype alignment. Please see the Supplementary Materials Appendix B subsection “Study 4 Supplementary Results” under “Pilot Testing Advertisement Stimuli for Study 4” (page 48-49) for complete details on the pre-test methods, analyses, and results. Overall, our pilot study results suggest that participants did not detect any differences in realism of the advertisements across conditions and that they rated all ads as highly realistic; additionally, the experimental stimuli have stereotype alignment ratings that match the stereotype portrayal condition manipulation.

****One of the potentially interesting findings in this manuscript pertains to the differences between the results obtained in study 4 compared to the other studies. However, given the incidental nature of this (not previously hypothesized) difference and all the previous comments, the most convincing way to demonstrate such a difference and to bolster the robustness of the other findings would be to run a new, truly preregistered, study. Such a study could include a 2 (political ideology) by 2 (stereotypicality) by 2 (racial identity of targets) between-participant experimental array using ad stimuli that resemble each other as much as possible on all dimensions. If properly preregistered (in line with the 8 comments on preregistration above, including hypotheses and analyses) and assuming an adequate sample size, such a study would significantly contribute to the demonstration of the reported effects.**

RESPONSE: We agree that our findings from Study 3 using only White models in the target advertisements (previously named Study 4) are particularly interesting compared to Preliminary Study and Studies 1-2 that used only racial minority models in the target advertisements (previously named Studies 1, 2, and 3). We agree that a fully pre-registered study with a three-way factorial design is necessary to evaluate the robustness of our findings.

In this revision, we include a new and highly-powered ($n = 2,484$) study specifying a 2 (political ideology: conservative, liberal) x 2 (stereotype portrayal: congruent, incongruent) x 2 (model race: minority, White) design to confirm two-way interactions and explore three-way interactions. Please see pages 10-14 in the manuscript for the results of this new study.

Additionally, for this additional study, following your recommendation, we used updated experimental stimuli with target advertisements that resemble each other as much as possible across conditions (see the images pasted in the prior response in this Response Letter (Response Letter page 16), or Table 1 on page 23-24 in the manuscript for those stimuli, and page 48-49 in the Supplementary Materials Appendix B subsection "Study 4 Supplementary Results" under "Pilot Testing Advertisement Stimuli for Study 4" for the stimuli pre-test results).

Overall, running preregistered/confirmatory two-way factorial ANOVAs and exploratory three-way factorial ANOVAs, followed by appropriate simple effects analyses, we find a replication of our results from our prior studies: First, attitudes are significantly impacted by the interaction effect of political ideology and stereotype portrayal, such that liberals prefer incongruent portrayals significantly more than congruent portrayals, conservatives prefer congruent portrayals significantly more than incongruent portrayals, and conservatives prefer congruent portrayals significantly more than liberals. Second, we found a significant interaction effect of political ideology and model race on attitudes, such that liberals prefer advertisements featuring only minority models significantly more than advertisements featuring only White models, conservatives prefer white models significantly more than minority models, liberals prefer minority models significantly more than conservatives, and conservatives prefer White models significantly more than liberals. Third, we found a significant three-way interaction effect of all three factors on attitudes.

Regarding our other measures, we found no significant main effects, two-way interaction effects, or three-way interaction effects on economic perceptions. For downstream hiring, we found no significant two-way or three-way interaction effects; we did find a significant main effect of political ideology and of stereotype portrayal on hiring, such that conservatives were significantly more likely to hire the stereotype congruent software candidate than liberals, and that participants in the congruent condition were significantly more likely than participants in the incongruent condition to hire the congruent candidate.

Taken together and in large part, these results replicate the results of Preliminary Study and Studies 1-3, thus strengthening our overall empirical contribution and the robustness of our findings.

On a technical note, the raw data on the OSF contain potentially identifiable data – they include IP addresses and geographical coordinates for many of the observations. These should be removed.

RESPONSE: Thank you. We have removed all personal identification data (including IP Address, Location Longitude, Location Latitude, as well as workerID, PROLIFIC_PID, or participantId embedded data from the participant recruitment platforms TurkPrime, Prolific, and CloudResearch Connect) from the raw datafiles and uploaded anonymized versions of the raw datafiles to OSF.

Contextualization of the study:

The Introduction section includes many news items (e.g., NPR, The Atlantic). Had such sources been included in addition to academic sources there would have been no issue, but they cannot be relied on as a primary source. Ample relevant studies exist.

RESPONSE: In our original submission, we did incorporate many academic sources (see list below with original submission reference #s). In this revised introduction, we engaged more deeply with these academic sources and with additional academic sources (we have included a list of those additional sources below).

For example: *“Notable work has been conducted on each component of this intersection. For instance, research on political ideology has investigated the nuances underlying political disagreement, citing intolerance, between-group affect, heated dislike of political out-group members, and distinct cognitions and belief systems as important drivers of political polarization (11-16)”* (page 3 in manuscript).

Additionally: *“However, past research at the intersection of political ideology and social identity stereotypes has predominantly focused on how conservatives and liberals differ in their evaluations of traditional stereotypes (17-19), showing differences between liberals and conservatives in their preference, endorsement, or perpetuation of commonly-held and pervasive stereotypes. This body of research suggests that liberals may be more implicitly malleable in their responses to traditional stereotypes (17), conservatives are more likely to endorse race and gender-based stereotypes (18), and that liberals and conservatives differ in both explicit evaluations and implicit perceptions of a series of social identity-based stereotypes (19)”* (page 3 in manuscript).

Academic References in Original Manuscript (with original reference #s):

- ¹¹Angle, J. W., Dagogo-Jack, S. W., Forehand, M. R., & Perkins, A. W. (2017). Activating stereotypes with brand imagery: The role of viewer political identity. *Journal of Consumer Psychology, 27*(1), 84–90. <https://doi.org/10.1016/j.jcps.2016.03.004>
- ¹²Stern, C., & Axt, J. (2021). Ideological Differences in Race and Gender Stereotyping. *Social Cognition, 39*(2), 259–294. <https://doi.org/10.1521/soco.2021.39.2.259>
- ¹³Stern, C., West, T. V., & Rule, N. O. (2015). Conservatives negatively evaluate counterstereotypical people to maintain a sense of certainty. *Proceedings of the National Academy of Sciences, 112*(50), 15337–15342. <https://doi.org/10.1073/pnas.1517662112>

- ¹⁴Nosek, B. A., Smyth, F. L., Hansen, J. J., Devos, T., Lindner, N. M., Ranganath, K. A., Smith, C. T., Olson, K. R., Chugh, D., Greenwald, A. G., & Banaji, M. R. (2007). Pervasiveness and correlates of implicit attitudes and stereotypes. *European Review of Social Psychology*, *18*(1), 36–88. <https://doi.org/10.1080/10463280701489053>
- ¹⁵Bettencourt, B. A., Dill, K. E., Greathouse, S. A., Charlton, K., & Mulholland, A. (1997). Evaluations of Ingroup and Outgroup Members: The Role of Category-Based Expectancy Violation. *Journal of Experimental Social Psychology*, *33*(3), 244–275. <https://doi.org/10.1006/jesp.1996.1323>
- ¹⁶Damer, E., Webb, T. L., & Crisp, R. J. (2019). Diversity may help the uninterested: Evidence that exposure to counter-stereotypes promotes cognitive reflection for people low (but not high) in need for cognition. *Group Processes & Intergroup Relations*, *22*(8), 1079–1093. <https://doi.org/10.1177/1368430218811250>
- ¹⁷Prati, F., Crisp, R. J., & Rubini, M. (2015). Counter-stereotypes reduce emotional intergroup bias by eliciting surprise in the face of unexpected category combinations. *Journal of Experimental Social Psychology*, *61*, 31–43. <https://doi.org/10.1016/j.jesp.2015.06.004>
- ¹⁸Rudman, L. A., & Fairchild, K. (2004). Reactions to Counterstereotypic Behavior: The Role of Backlash in Cultural Stereotype Maintenance. *Journal of Personality and Social Psychology*, *87*(2), 157–176. <https://doi.org/10.1037/0022-3514.87.2.157>
- ¹⁹Matta, S., & Folkes, V. S. (2005). Inferences about the Brand from Counterstereotypical Service Providers. *Journal of Consumer Research*, *32*(2), 196–206. <https://doi.org/10.1086/432229>

^These are exclusive of the implicit and explicit stereotype literature that we cite and discuss, which we address in our response to your next feedback point.

Additional Academic References in Revised Manuscript (with updated reference #s):

- ¹¹Iyengar, S., & Westwood, S. J. (2015). Fear and loathing across party lines: New evidence on group polarization. *American Journal of Political Science*, *59*(3), 690-707. <https://doi.org/10.1111/ajps.12152>
- ¹²Iyengar, S., Lelkes, Y., Levendusky, M., Malhotra, N., & Westwood, S. J. (2019). The origins and consequences of affective polarization in the United States. *Annual Review of Political Science*, *22*(1), 129-146. <https://doi.org/10.1146/annurev-polisci-051117-073034>
- ¹³Finkel, E. J., Bail, C. A., Cikara, M., Ditto, P. H., Iyengar, S., Klar, S., et al. (2020). Political sectarianism in America. *Science*, *370*(6516), 533-536. <https://doi.org/10.1126/science.abe1715>
- ¹⁴Jost, J. T. (2017). Ideological asymmetries and the essence of political psychology. *Political Psychology*, *38*(2), 167-208. <https://doi.org/10.1111/pops.12407>
- ¹⁵Brandt, M. J., Reyna, C., Chambers, J. R., Crawford, J. T., & Wetherell, G. (2014). The ideological-conflict hypothesis: Intolerance among both liberals and conservatives. *Current Directions in Psychological Science*, *23*(1), 27-34. <https://doi.org/10.1177/0963721413510932>
- ¹⁶Crawford, J. T., & Pilanski, J. M. (2014). Political intolerance, right and left. *Political Psychology*, *35*(6), 841-851. <https://psycnet.apa.org/doi/10.1111/j.1467-9221.2012.00926.x>

The introduction says that “The theoretical and empirical attention given to stereotypes has historically been centered on implicit stereotypes (e.g., 20-21), and has slowly expanded to include focus on explicit stereotypes”. This is not an accurate description of the trajectory of research on stereotypes. The so-called “implicit revolution” in social

cognition started in the '90s with the application of cognitive priming research to social constructs such as stereotypes and the introduction of implicit measures such as the implicit association test by Greenwald and Banaji. These measures were introduced to uncover perceptions that were hypothesized to be undetectable with explicit measures. Research up to that point focused exclusively on explicit stereotypes, perhaps starting with the classic Princeton trilogy (Katz & Braley 1933) and all the way to the more contemporary stereotype content model by Fiske and colleagues, with many fantastic contributions by many scholars. The authors may want to significantly edit this point.

RESPONSE: Thank you for raising this important point. We've considerably edited this paragraph to more accurately reflect the evolution of stereotype research.

The revised paragraph now reads as follows (page 3-4 in manuscript):

“Further, updated research on explicitly divergent or stereotype incongruent portrayals of marginalized identities that reflects the changing portrayals in current films, brand advertisements, and mainstream media is necessary in advancing our understanding of stereotypes over time. The theoretical and empirical attention given to stereotypes has historically focused on explicit stereotypes (e.g., 26) and extended through contemporary frameworks (e.g., 27). As research developed, the field expanded significantly with increased attention to implicit social cognition (e.g., 28-29), focusing on implicit bias measurement and other measures beyond explicit self-report (e.g., 30). Recent research has examined the relationship between these explicit and implicit domains (e.g., 19, 31-34). In our work, we argue it is important to directly study explicitly stereotype congruent and stereotype incongruent portrayals, given the greater media coverage of explicit representations and the heightened attention that larger society is taking to how marginalized groups are portrayed. Indeed, in wider American society, explicit biases and preferences are frequently being expressed, oftentimes to staunch or extreme ends. Directly studying responses to explicit stereotype portrayals from opposing political ideologies reflects the current social moment. Moreover, focusing on explicit stereotypes and explicit reports of reactions to stereotype portrayal may reveal a new founded comfort or willingness among participants to openly express their potentially preferential or biased beliefs, in turn updating the research on explicit stereotypes. Thus, the present work investigates both explicitly stereotype congruent and explicitly stereotype incongruent portrayals of varying race and gender identities.”

We have added these additional sources to complement the revision:

Katz, D., & Braly, K. (1933). Racial stereotypes of one hundred college students. *The Journal of Abnormal and Social Psychology*, 28(3), 280–290. <https://doi.org/10.1037/h0074049>

Fiske, S. T., Cuddy, A. J. C., Glick, P., & Xu, J. (2002). A model of (often mixed) stereotype content: Competence and warmth respectively follow from perceived status and competition. *Journal of Personality and Social Psychology*, 82(6), 878–902. <https://doi.org/10.1037/0022-3514.82.6.878>

Greenwald, A. G., McGhee, D. E., & Schwartz, J. L. K. (1998). Measuring individual differences in implicit cognition: The implicit association test. *Journal of Personality and Social Psychology*, 74(6), 1464–1480. <https://doi.org/10.1037/0022-3514.74.6.1464>

The novelty of this work in examining the impact of political ideology on attitudes vis-à-vis perceptions of stereotype congruity and incongruity would be highlighted if relevant previous studies would be discussed more in-depth. For example, Stern et al. (2015) already characterized how liberals and conservatives differ in assessing stereotype-congruent and -incongruent stimuli (in a different context).

RESPONSE: Thank you for this recommendation. We did indeed reference Stern et al. (2015) in our original introduction.

We have emphasized this literature more in the revised discussion: *“Taken together, we show a significant and robust divergence between political ideologies, that the divergence in reactions is characterized by a consistency among liberals and a variability among conservatives. This variability among conservatives’ evaluations contends with prior work that both conservatives and liberals are equally intolerant and staunch in their views (15-16). Indeed, our work provides support towards the argument that conservatives effectively and considerately respond to certain stimuli. Just as how prior work has shown that conservatives typically penalize counterstereotypical people but will only do so when the counterstereotypicality is functional or provides utility (20), our work shows that conservatives respond favorably to portrayals of White models, perhaps because there is a functionality or usefulness for them in those representations. Importantly, our work advances past research on political ideology, stereotype congruency, and racial representation by providing a thorough investigation into the effect of these three factors on attitudes across a variety of target advertisements”* (page 16 in manuscript).

The introduction portrays liberals as advocating for progress and fighting social injustice, whereas conservatives favor tradition, which, at least the way the sentence is structured, is opposed to reducing social injustice (and hence, tradition is necessarily for social injustice). Some may say this is a biased characterization of conservative values and should be edited to reflect a more neutral view.

RESPONSE: We’ve revised several sentences throughout our introduction to avoid presenting any bias against conservative values. Instead, we carefully modify our language to present the liberal versus conservative ideologies and preferences in a more neutral style.

For instance, we’ve revised to present a more balanced view on page 2: *“For instance, while liberals often prioritize addressing perceived social injustices and advocate for change (i.e., “stay woke”), conservatives frequently emphasize preserving traditional values and institutions they view as foundational to society (i.e., “war on woke”) (4). These liberal-driven movements...”*

We’ve also edited a sentence on page 3: *“Popular mass media portrayals of marginalized groups in the U.S. (8), such as racial minorities and women, have incurred strongly divergent reactions from liberal- and conservative-minded viewers, wherein the former tend to advocate for more diverse representations, while the latter boycott often express concerns or penalize representations that may not reflect authentic characterizations or merit-based qualifications.*

Finally, we revised a sentence on page 3 to *“The same stimulus (i.e., an advertisement, a movie trailer, etc.) may incite strongly divergent actions such as praise and increased support from those who view it as positive progress, or criticism and penalizing “review bombing” backlash from those who see it as undermining important cultural traditions or standards.”*

Technical issues/suggestions:

The provided data files, although reported as raw, seem to be minimally processed in some way. The timestamps for the data entries do not feature the expected chronological order in most of the files. Please provide fully raw data files (and, ideally, the code used to process them).

RESPONSE: We have uploaded fully raw and newly anonymized (as noted in response to previous feedback from you) data files to OSF. We are not sure why the timestamps are not chronological, but this was how the data was downloaded from Qualtrics; it appears as though the downloaded data organizes by Finished and Consent columns – this was not any processing that we did, but seems to be the default that Qualtrics uses for downloads. We are happy to and plan to provide the analysis code if the paper is accepted.

Given the unbalanced sample and exclusion rates, it is important to provide a full description of the number of participants allocated to each condition to reliably evaluate the robustness of the findings.

RESPONSE: Please see table below (page 23 in this response letter), which we've also included in the Supplementary Materials Appendix A subsection "Sample Size by Condition" (page 35-36), detailing participants allocated to each condition. We also provide a complete and thorough reporting of preregistered sample sizes calculations/aims, participant recruitment timing, data exclusion criteria and associated counts, and non-participant rates for each study in Supplementary Materials Appendix A subsection "Sample Demographics" on page 37-40 in the revised manuscript.

Supplementary Table S1: Sample size by participant ideology and experimental conditions.

Study	N , Total Clean Study Sample (Post-Exclusions)	n , Sample by Participant Political Ideology (PI)	n , Sample by Stereotype Portrayal (SP) Conditions	n , Sample by PI x SP	n , Sample by Model Race (MR) Conditions	n , Sample by PI x MR	n , Sample by PI x SP x MR
Pilot Study	N = 220	N/A	n _{Cong} = 115 n _{Incong} = 105	N/A	N/A	N/A	N/A
Preliminary Study	N = 196	n _{Cons} = 61 n _{Libs} = 135	n _{Cong} = 100 n _{Incong} = 96	n _{Cons*Cong} = 29 n _{Cons*Incong} = 32 n _{Libs*Cong} = 71 n _{Libs*Incong} = 64	N/A	N/A	N/A
Study 1	N = 958	n _{Cons} = 423 n _{Libs} = 535	n _{Cong} = 476 n _{Incong} = 482	n _{Cons*Cong} = 204 n _{Cons*Incong} = 219 n _{Libs*Cong} = 272 n _{Libs*Incong} = 263	N/A	N/A	N/A
Study 2	N = 900	n _{Cons} = 351 n _{Libs} = 549	n _{Cong} = 438 n _{Incong} = 462	n _{Cons*Cong} = 171 n _{Cons*Incong} = 180 n _{Libs*Cong} = 267 n _{Libs*Incong} = 282	N/A	N/A	N/A
Study 3	N = 783	n _{Cons} = 377 n _{Libs} = 406	n _{Cong} = 380 n _{Incong} = 403	n _{Cons*Cong} = 185 n _{Cons*Incong} = 192 n _{Libs*Cong} = 195 n _{Libs*Incong} = 211	N/A	N/A	N/A
Study 4	N = 2,484	n _{Cons} = 1,161 n _{Libs} = 1,323	n _{Cong} = 1,256 n _{Incong} = 1,228	n _{Cons*Cong} = 582 n _{Cons*Incong} = 579 n _{Libs*Cong} = 674 n _{Libs*Incong} = 649	n _{Minor} = 1,255 n _{White} = 1,229	n _{Cons*Minor} = 579 n _{Cons*White} = 582 n _{Libs*Minor} = 676 n _{Libs*White} = 647	n _{Cons*Cong*Minor} = 279 n _{Cons*Cong*White} = 303 n _{Cons*Incong*Minor} = 300 n _{Cons*Incong*White} = 279 n _{Libs*Cong*Minor} = 348 n _{Libs*Cong*White} = 326 n _{Libs*Incong*Minor} = 328 n _{Libs*Incong*White} = 321

Note. For Pilot and Preliminary Studies, unlike for Studies 1-4, we did not seek a sample balanced by political ideology.

Relatedly, the study does not report the demographic characteristics of the samples. Although the study reports no a-priori hypothesis regarding the impact of such characteristics, the interpretation of the results would be vastly different if, for example, the samples were predominantly White versus a mixed racial/ethnic background. The same applies to the other demographic details collected, especially with regard to gender.

RESPONSE: We report participant race and gender demographics for all studies (Pilot Study, Preliminary Study, Studies 1-4) on pages 37-40 in Supplementary Materials Appendix A under “Sample Demographics”; please see Supplementary Table S2 through Supplementary Table S7 for each study’s breakdown of participant race and gender. Below we have pasted Supplementary Table S7 for quick reference to show the formatting and reporting style used in these demographic tables. Indeed, as is usually the case with recruiting samples from online platforms, the participants are predominantly White. We mention this as a limitation and future consideration in our updated discussion section. For instance, on page 16, we note: *“Of note, future work may consider exploring how the demographic make-up beyond political ideology of the participants may moderate results. For instance, recruiting only non-dominant identities (i.e., only racial minorities liberal and conservative participants or only female liberal and conservative participants) and assessing their reactions to congruent versus incongruent portrayals of their own non-dominant group may reveal findings important to understanding further intersections of identity and representation.”*

Supplementary Table S7. Study 4: Participant Race and Gender

Participant gender	Participant race	#	% of sample
Male	White	758	30.5%
	Black	90	3.6%
	Asian	74	3.0%
	Hispanic	75	3.0%
	American Indian or Alaska Native	1	0.0%
	Native Hawaiian or Pacific Islander	3	0.1%
	Other	13	0.5%
Female	White	1143	46.0%
	Black	114	4.6%
	Asian	75	3.0%
	Hispanic	84	3.4%
	American Indian or Alaska Native	3	0.1%
	Native Hawaiian or Pacific Islander	1	0.0%
	Other	21	0.8%
Other	White	15	0.6%
	Black	3	0.1%
	Asian	1	0.0%
	Hispanic	4	0.2%
	Other	6	0.2%

The manuscript reports the results of regression models in all studies. It might be interesting to include participants with moderate political views in these analyses to examine the effect and potentially increase the statistical power, particularly if only liberals show a difference between the stereotypicality condition.

RESPONSE: Thank you for this suggestion. In our response to your feedback point (2), we mentioned that our theoretical focus and operationalization of our research question is on differences in outcomes between politically conservative and politically liberal participants. We conduct and report the results of regression models of stereotype portrayal and political ideology (including politically moderate participants) on attitudes for Studies 1, 2, and 3 in the Supplementary Materials – please see Appendix B pages 44-45, 47, and 47-48, respectively. We find that moderates' attitudinal responses lie between conservatives' and liberals' (as one might expect). There were only a couple significant pairwise differences between moderates and their more extreme counterparts, present only in Study 3 (target advertisements featuring only White models); otherwise, moderates were largely no significantly different from liberals or conservatives.

Reviewer #2 (Remarks to the Author):

Nature Communications Psychology

Liberals and conservatives respond divergently to stereotype portrayals of racial minorities and women

The authors set out to examine liberal and conservative reactions to stereotypical and counter-stereotypical portrays of women and racial minorities. I had a few thoughts regarding the operationalization of stereotypical and counter-stereotypical portrayals, as well as a desire for a bit more initial setup of the research question. I hope my thoughts are constructive and help move their research forward.

RESPONSE: Thank you so much for your time in providing such constructive feedback. We thoroughly consider and respond to each point below.

One comment that kept arising for me was the conflation between increased diversity and counter-stereotypicality, or incongruency versus unaffiliated. The authors assumed that all diversity is inherently counter-stereotypical, when that simply can't be true. Just because I don't have a representation for all race/gender combinations does not make the representations I don't have inherently *counter-stereotypical*. For a stereotype to be counter-stereotypical, it should be explicitly counter to a known stereotype. For example, the authors use Asian women athletes as a counter-stereotypical exemplar. It seems as if the stereotype they are counter-acting is Asian women being nerds. But that feels like a stretch versus me simply not having a stereotypical representation of Asian women as athletes at all. They don't enter into the space as stereotypical or counter-stereotypical. I simply don't have a cognitive representation for them in that space. Relatedly, some of their examples were fascinatingly intersectional and other ones weren't. For example, there are several papers out there showing how Black women get a pass due to intersectional invisibility (Purdie-Vaughns & Eibach, 2008) when engaging in more dominant, masculine behaviors. To what extent is a Black woman software engineer seen as counter-stereotypical or there aren't strong representations due to the perceived masculinity of Black women (Hall et al., 2019; Livingston et al., 2012; Rosette et al., 2016)? These types of dynamics seem to be glossed over in the creation of the stimuli as

well as in the authors' theorizing, but likely influenced results. Another example of intersectional dynamics that seemed to be missing in nuance is why the tutoring services modeled by White men were deemed incongruent. If tutoring is meant to be more feminine, might participants have read gayness onto the men? If so, the example is counter-stereotypical in one sense, but stereotypical in another. The domain versus the actor within the domain gets a bit muddled and confused in terms of this stereotypical versus counter-stereotypical language. Now to be fair, I know the authors pretested the stimuli by asking people "The way the models are depicted in the ads does not align with common stereotypes." I still feel the stimuli are not clear depictions of the stereotypical versus counter-stereotypical dimension the authors hoped to test. I can say that the models don't align with common stereotypes but that does not mean I believe they are counter-stereotypical.

RESPONSE: Thank you for raising such an important point. We agree that our use of "stereotypical" versus "counterstereotypical" in the initial draft might lead to an assumption of two narrow camps of stereotypicality, which does not capture the full nuance in our intended depictions of racial minorities and women. Indeed, we operationalize our stimuli and our stereotype portrayal conditions as being either *stereotype congruent* or *stereotype incongruent* because this reflects the true nature of the operationalization: content (such as target advertisements) may portray traditional stereotypes, or they may not. And those that do not are not necessarily "counterstereotypical" so to speak – they may simply be incongruent with stereotypes, which includes the absence of stereotypical affiliation. Our original framework on stereotype portrayal thus consists of two groups: stereotypes present/stereotype congruent portrayals, stereotypes absent/stereotype incongruent portrayals.

Our original conceptualization of congruent versus incongruent stereotype portrayal is reflected in our pilot test measures. We created the Cronbach measure for stereotype alignment based on 5 items, which are listed below. Please note that none of these mention counterstereotypicality, instead just the absence or not aligning with commonly held stereotypes (i.e., stereotype incongruence).

- 1) These advertisements illustrate general stereotypes that are commonly held by society.
- 2) The way the models are depicted in the ads does not align with common stereotypes. (reverse-coded)
- 3) These ads are unconventional. (reverse-coded)
- 4) The way the models are depicted in the ads matches common stereotypes.
- 5) The ads do not portray common stereotypes. (reverse-coded)

In writing the initial submission, we shorthanded "stereotype incongruent" into "counterstereotype," thinking this would be a more straightforward way for readers to quickly read and understand our conceptualization. However, your point has encouraged us to revise our manuscript such that we remove mention of counterstereotypical, as it is misleadingly specific and narrow. Instead, we use our original framing of stereotype incongruent, as it captures the wider possibilities of depictions, including those of White male tutors, Asian female athletes, etc. For instance, we modify our language from "To evaluate this, we conducted four preregistered experiments ($N = 5,125$) with U.S. citizens in which we operationalized stereotype portrayal as either *stereotype congruent* (i.e., stereotypical, depicting commonly-held stereotypes) or *stereotype incongruent* (i.e., counterstereotypical, depicting divergence from commonly-held stereotypes)" to "...we operationalized stereotype portrayal as either *stereotype congruent* (i.e., depicting commonly-held stereotypes) or *stereotype incongruent* (i.e., absence of commonly-held stereotypes, including depiction divergence from commonly-held stereotypes)" (manuscript page 4).

This specification and clarification of “stereotype incongruence” is used throughout the introduction. For instance on page 4: *“Thus, the present work investigates both explicitly stereotype congruent and explicitly stereotype incongruent portrayals of varying race and gender identities”* and on page 2: *“Over the past few years, for instance, food product branding that is in line with stereotypes and film/tv cast lists that are not in line with stereotypes have incited sharp attention from liberals and conservatives, respectively (9, 10).”*

Similarly, we do not focus on “counterstereotypes” in our general discussion, with our first paragraph in that section featuring the following sentence: *“In this work, we examined how political ideology affects the way U.S. citizens evaluate portrayals of race and gender identities that are either in line and congruent with traditional stereotypes or divergent and incongruent with traditional stereotypes.”* (manuscript page 14).

Finally, we acknowledge that perhaps our operationalization is still imperfect, and incorporate your feedback into discussing the limitations of our work. For instance, in the general discussion, we include a few sentences noting: *“Further, future studies may consider dissecting and breaking down stereotype incongruence into two subcategories, strict counterstereotypes (i.e., depictions of individuals that directly counter commonly-held stereotypes) versus non-stereotypes (i.e., depictions of individuals that are neither blatantly stereotypical nor counterstereotypical; depictions that are unaffiliated with stereotypes), and investigating how conservative and liberal viewers respond to such depictions”* (manuscript page 16-17).

Secondly, in the introduction, the authors note that there hasn’t been much research on how “conservatives and liberals differ in their evaluations of traditional stereotypes.” However, they don’t fully motivate why we wouldn’t just expect the opposite (their hypotheses feel almost like a tautology at this point) nor why we should study this beyond “it hasn’t been studied yet”. I don’t find that all research that hasn’t been done should be done, so I encourage the authors to motivate why this work is important to do.

RESPONSE: Apologies for the confusion in this phrasing. In this phrase that you reference, we are noting that past research indeed *predominantly* focuses on how liberals and conservatives differ in evaluating *traditional* stereotypes (i.e., commonly-held stereotypes and stereotype congruent depictions of social identities). This is our build up to our current work, which incorporates and really focuses on stereotype incongruent portrayals of social identities. We are building on past work by including stereotype incongruent into empirical investigation on stereotypes, which not only reflects the current times that we illustrate in our first two introduction paragraphs, but also expands the study of stereotypes to compare congruency and incongruency in tandem. We’ve modified this paragraph in hopes to make our build-up, contribution, and the importance of our work clearer.

Following the general introduction, the two paragraphs on page 3 in our introduction focus on our contribution and literature positioning. We write:

Although understanding psychological processes underlying these polarized viewpoints could help to defuse these palpable real-world tensions, comprehensive research on the intersection of political ideology, EDI, and marginalized group stereotypes is limited. Notable work has been conducted on each component of this intersection. For instance, research on political ideology has investigated the nuances underlying political disagreement, citing intolerance, between-group affect, heated dislike of political out-group members, and distinct cognitions and belief systems as important drivers of

political polarization (11-16). Such polarization contributes to the way conservatives and liberals differentially perceive stereotypes of social identity groups. However, past research at the intersection of political ideology and social identity stereotypes has predominantly focused on how conservatives and liberals differ in their evaluations of traditional stereotypes (17-19), showing differences between liberals and conservatives in their preference, endorsement, or perpetuation of commonly-held and pervasive stereotypes. This body of research suggests that liberals may be more implicitly malleable in their responses to traditional stereotypes (17), conservatives are more likely to endorse race and gender-based stereotypes (18), and that liberals and conservatives differ in both explicit evaluations and implicit perceptions of a series of social identity-based stereotypes (19). Some research has studied reactions to counterstereotypical portrayals of marginalized identities, but these often do not study differences in reactions by political ideology (20-25), thus leaving open the critical question of how political ideology and stereotype portrayal interact.

Very little empirical work has simultaneously studied how political ideology affects reactions to explicit portrayals of marginalized identities that either maintain or diverge from traditional stereotypes, despite the considerable interconnection between political ideology and stereotype portrayal. As such, we seek to provide a thorough examination not only of how responses to stereotypical portrayals of race and gender may or may not differ from reactions to stereotype incongruent portrayals of the same subjects, but also of how these reactions depend upon the political ideology of the viewer. Our novel testing of the interactive dynamics of political ideology and stereotype portrayal on attitudes, economic attributions, and downstream behaviors will contribute a fuller understanding of the relationships between these important and socially contentious topics. Furthermore, we incorporate racial representation as an additional factor in understanding the effects of political ideology and stereotype portrayal on explicit evaluations and perceptions. Specifically, we vary the racial make-up in explicit stereotype congruent and explicit stereotype incongruent portrayals to assess the extent to which conservatives and liberals diverge in their direct evaluations of such representations.

Our third paragraph in this section on page 3-4, in which we consider explicit and implicit stereotypes, further motivates our investigation.

Smaller points

The way it reads is that awareness reinforces systematic inequalities, which doesn't quite make sense.

RESPONSE: We agree that this phrasing could be improved. We've rephrased to argue that greater attention is reinforcing beliefs and stirring increased behavioral responses: "*These liberal-driven movements and its backlash from conservatives reflect an intensifying, negative spiral: growing attention to the way individuals are represented in popular media reinforces beliefs and, in turn, leads to behavioral responses such as boycotts, cancellations, as well as accusations of "cancel culture" (5-7)*" (manuscript page 2).

• Also on page 2, the authors say that "review bombing" was part of negative backlash. But review bombing can also happen in a positive direction, where people purposefully flood a space with positive reviews to counteract backlash.

RESPONSE: We clarified use of review bombing as “penalizing review bombing” (manuscript page 2) so that this malicious act of posting negative user reviews online was clearly negative.

• A bit confused by the difference between Study 2 and Study 3. Is Study 3 the second wave of Study 2? How was Study 3 different from Study 2?

RESPONSE: These two studies are unique because Study 1 (previously “Study 2”) focused only on the interaction between stereotype portrayal and political ideology on attitudes, whereas Study 2 (previously “Study 3”) included an exploratory mediator.

(Regarding study renaming, please see the below Response Letter Table RL1 for the updated study names and our rationale for renaming the studies.)

This exploratory mediator for Study 2 was excluded from our initial submission due to friendly feedback from colleagues who noted the mediation distracts from the main contribution of the paper (that of the interaction effect on outcomes). We have since re-included the mediation results to the Supplementary Materials Appendix B subsection “Study 2 Supplementary Results” under “Mediation Results” on page 45-47.

Further, we continue to find value in the replication test that Study 2 provides for Study 1, since both studies test the same primary outcome measures and use the same racial minority target advertisements; as such, Study 2 serves as a robustness test to Study 1.

Response Letter Table RL1: Updated Study Names.

Initial Submission Study Names / #s	Revised Manuscript Study Names / #s	Rationale for Renaming
Pilot Study	Pilot Study	N/A
Study 1	Preliminary Study	This study is renamed to be Preliminary because it had a smaller sample size, measured attitudes, and the findings were replicated in Study 1. → Results are reported in Supplementary Materials Appendix B subsection “Preliminary Study: Testing Attitudes”
Study 2	Study 1	First main study with a complete suite of DVs: attitudes, \$, and hiring; sought large, balanced sample of Conservatives and Liberals.
Study 3	Study 2	This study maintains the complete suite of DVs with a large, balanced sample of participants.
Study 4	Study 3	This study maintains the complete suite of DVs with a large, balanced sample of participants.
N/A	Study 4	This is a new 2x2x2 study that did not exist in our initial submission. It also maintains the complete suite of DVs with a large, balanced sample.

• **Were the economic outcomes findings robust to outliers? I assume there were some with the large SDs.**

RESPONSE: We found no outliers in any of our studies. Please see below for the histogram of the economic index outcome for each main study, Studies 1-4. The large standard deviations in our results represent a wide, heterogeneous, but legitimate dispersion of economic index responses across the available range of \$0-\$500, as shown by these histograms.

We checked for outliers using visual methods (e.g., boxplots and histograms), using z-scores to check for responses that are more than 3 standard deviations from the mean, and using the interquartile range (IQR) to check for responses that are outside lower ($Q1 - 1.5 \times IQR$) and upper ($Q1 + 1.5 \times IQR$) bounds.

As a reminder, our economic index is the average of two questions, both of which bound responses within the range of \$0-\$500: *“Economic perceptions of monetary value are captured in an index, which consisted of a willingness to pay item (“Thinking about yourself... In total, how much would you be willing to pay for all of these products/services (in U.S. dollars \$)?” (Sliding bar from \$0 to \$500)”) and a worth to others item (“In total, how much do you think all of these products/services are worth to the average American (in terms of U.S. dollars \$)?” (Sliding bar from \$0 to \$500)”) (a $\geq .78$, measured in Studies 1-4).”* (Supplementary Materials Appendix B subsection “Index Measures” under “Economic Perceptions” on page 43).

• **The authors measured political ideology using social and fiscal questions. Should we expect fiscal conservatives to react to stereotype differences by identity? It seems like we should only expect effects for social conservatives, not fiscal ones.**

RESPONSE: Thank you for raising this interesting question. Because social and fiscal ideology are highly correlated (e.g., Jost et al., 2003) and because political ideology is increasingly an all-encompassing “mega-identity” (e.g., Mason, 2018), we would not expect different reactions based on the domain of conservatism. In fact, we agree with the scholars who demonstrate that the high correlation between social and fiscal ideology and would argue that fiscal conservatives are social conservatives and vice versa. Finally, because our paper focuses on participants with strong and consistent political ideologies across domains (i.e., we measured political ideology using political leaning, political party, social, and fiscal questions), we expect our outcomes to be robust despite any potential difference within each component of ideology.

Jost, J. T., Glaser, J., Kruglanski, A. W., & Sulloway, F. J. (2003). Political conservatism as motivated social cognition. *Psychological Bulletin*, 129(3), 339–375.

Mason, L. (2018). *Uncivil agreement: How politics became our identity*. University of Chicago Press.

In response to the specific questions asked by NCP:

RESPONSE: Thank you for transparently including these questions and responses in your review. We respond briefly to each below.

• Does the paper represent an advance in understanding which may influence thinking in the field? If you have concerns about the advance in relation to specific studies, we appreciate references to this work.

o It is unclear exactly how to think about where to go next with these findings. I didn't think it advanced theory as much as showed an addendum to a phenomenon we could have guessed.

RESPONSE: We believe our work is important in advancing past research on political ideology, stereotype congruency, and racial representation by providing a thorough investigation into the effect of these three factors on attitudes across a variety of target advertisements. Not only do we consider how political ideology moderates both stereotype congruent and stereotype incongruent portrayals simultaneously, but we also manipulate racial representation in these designs. We believe the revision does a stronger job in effectively communicating the importance of our work.

• Does the article presents an original study, new analysis, new model, or a direct or extended replication of previous work?

o Original study, although its more of a conceptual replication of past work and adding in one additional moderator (e.g., stereotypical versus counter-stereotypical)

RESPONSE: We believe our work is original in investigating stereotype congruent and stereotype incongruent portrayals simultaneously, while also adding in a racial representation manipulation in our study design.

• Are the data and analysis technically sound? Are they appropriate to answer the research question, e.g., are causal research questions addressed on the basis of causal, rather than correlational evidence?

o The analyses are fine. I had some questions about the operationalization, as outlined above.

RESPONSE: We responded to your concern in your first feedback point that mentioned operationalization. We believe our response details how we conceived of, originally intended to present, and further clarifies the operationalization of stereotype portrayal as two groups: one of which is stereotype congruence (i.e., depicting commonly held stereotypes) and the other is stereotype incongruence (i.e., the absence of commonly held stereotypes, which may include portrayals that deviate from commonly held stereotypes or portrayals that are unaffiliated with commonly held stereotypes).

- **Does the paper provide strong evidence for its conclusions?**
 - **Yes**

RESPONSE: Thank you for evaluating as such. We hope the revision and additional three-way experimental study further strengthens this.

- **Is the study question important to scientists for a sub-field of psychology?**
 - **Yes. Stereotypes are hugely relevant for many sub-fields within psychology**

RESPONSE: We sincerely agree with you on the importance of conducting empirical research on stereotypes. We also believe that our investigation of the interaction of stereotype portrayal and political ideology is broadly appealing to social, political, and other psychologists.

- **Are there any special ethical concerns arising from the use of animals or human subjects?**
 - **Nope!**

RESPONSE: We agree with this assessment. We have also made sure to anonymize our raw data files to protect our online participant samples.

- **Was the study preregistered and if so, did the authors follow the preregistration?**
 - **They did pre-register their studies**

RESPONSE: All studies are pre-registered, including our new three-way interaction study. We responded to deviations to preregistrations in this review letter and transparent provide additional details on secondary and exploratory analyses, complete hypothesis testing, sample size, etc. in the Supplementary Materials Appendix A and Appendix B.

Overall, love new work on stereotypes! The novelty and insights from the studies were difficult for me to discern. I believe the insights are there!

RESPONSE: Thank you so much for your encouragement. We believe our revised manuscript and study package offer a clearer contribution.

Hall, E. V., Hall, A. V., Galinsky, A. D., & Phillips, K. W. (2019). MOSAIC: A Model of Stereotyping Through Associated and Intersectional Categories. *Academy of Management Review*, 44(3), 643–672. <https://doi.org/10.5465/amr.2017.0109>

Livingston, R. W., Rosette, A. S., & Washington, E. F. (2012). Can an Agentic Black Woman Get Ahead? The Impact of Race and Interpersonal Dominance on Perceptions of Female Leaders. *Psychological Science*, 23(4), 354–358. <https://doi.org/10.1177/0956797611428079>

Purdie-Vaughns, V., & Eibach, R. P. (2008). Intersectional Invisibility: The Distinctive

Advantages and Disadvantages of Multiple Subordinate-Group Identities. *Sex Roles*, 59(5–6), 377–391. <https://doi.org/10.1007/s11199-008-9424-4>

Rosette, A. S., Koval, C. Z., Ma, A., & Livingston, R. (2016). Race matters for women leaders: Intersectional effects on agentic deficiencies and penalties. *The Leadership Quarterly*, 27(3), 429–445. <https://doi.org/10.1016/j.leaqua.2016.01.008>

Reviewed by Sa-kiera Hudson, PhD

Assistant Professor at University of California, Berkeley Haas School of Business
sa.kiera.hudson@berkeley.edu

I intentionally sign all my reviews and any and all typos can be blamed on COVID being an ongoing and exhausting pandemic, the systematic dismantling of LGBTQ rights, the weariness and heartbreak associated with constant bombing and killings, and police brutality in America. If my signature must be removed by journal policy, I request that the note [signature redacted] be included in its stead.

EDITORIAL POLICIES

We ask that you ensure your manuscript complies with our editorial policies and reporting requirements.

To that end, we require revised manuscripts to be accompanied by two completed items: a reporting summary that collects information on study design and procedure, and an editorial policy checklist that verifies compliance with all required editorial policies

- Nature Research Reporting Summary
- Editorial Policy Checklist

All points on the policy checklist must be addressed. Your revised manuscript can only be sent back to the referees if these checklists are completed and uploaded with the revision.

Notes: If you have submitted a Stage 1 Registered Report, Review, Primer, Comment, or Perspective you do not need to submit these forms. If you have already submitted these forms, you may disregard this request.

Communications Psychology is committed to improving transparency in authorship. As part of our efforts in this direction, we are now requesting that all authors identified as 'corresponding author' create and link their Open Researcher and Contributor Identifier (ORCID) with their account on the Manuscript Tracking System prior to acceptance. ORCID helps the scientific community achieve unambiguous attribution of all scholarly contributions. You can create and link your ORCID from the home page of the Manuscript Tracking System by clicking on 'Modify my Springer Nature account' and following the instructions in the link below. Please also inform all co-authors that they can add their ORCIDs to their accounts and that they must do so prior to acceptance.
<https://www.springernature.com/gp/researchers/orcid/orcid-for-nature-research>

If you experience problems in linking your ORCID, please contact the Platform Support Helpdesk.

This email has been sent through the Springer Nature Tracking System NY-610A-NPG&MTS

Confidentiality Statement:

This e-mail is confidential and subject to copyright. Any unauthorised use or disclosure of its contents is prohibited. If you have received this email in error please notify our Manuscript Tracking System Helpdesk team at <http://platformsupport.nature.com> . Details of the confidentiality and pre-publicity policy may be found here <http://www.nature.com/authors/policies/confidentiality.html>

June 24, 2025

Dear Editorial and Review Team,

Thank you for the continued review of our work, "Liberals and conservatives respond divergently to stereotype portrayals of race and gender" (COMMSPSYCHOL-24-0428A), and for deeming it suitable for publication in *Communications Psychology* pending final revisions.

In the remaining of this letter, we respond to each request made by all members of the review team. All of the comments from the review team appear in bold, and our responses appear in standard font. In this final round of revision, we believe we have appropriately addressed each request, including those from Reviewer #1 the complete set of Editorial Requests.

Thank you for your continued review of our research!

Your manuscript titled "Liberals and conservatives respond divergently to stereotype portrayals of race and gender" has now been seen by our reviewers, whose comments appear below. In light of their advice I am delighted to say that we are happy, in principle, to publish a suitably revised version in Communications Psychology.

We therefore invite you to revise your paper one last time to address the remaining concerns of our reviewers and a list of editorial requests. At the same time we ask that you edit your manuscript to comply with our format requirements and to maximise the accessibility and therefore the impact of your work.

EDITORIAL REQUESTS:

RESPONSE: Thank you, we have completed the Editorial Requests Table for our paper. It is included in our portal submission.

SUBMISSION INFORMATION:

In order to accept your paper, we require the files listed here <https://www.nature.com/documents/commsj-file-checklist.pdf> .

RESPONSE: We have cross-referenced this File Checklist to ensure that our final submission includes all relevant and required files.

OPEN ACCESS:

Communications Psychology is a fully open access journal. Articles are made freely accessible on publication. For further information about article processing charges, open access funding, and advice and support from Nature Research, please

visit <https://www.nature.com/commspsychol/open-access>

RESPONSE: We support open access and we are prepared to make these declarations and payments.

*** TRANSPARENT PEER REVIEW: Communications Psychology uses a transparent peer review system. On author request, confidential information and data can be removed from the published reviewer reports and rebuttal letters prior to publication. If you are concerned about the release of confidential data, please let us know specifically what information you would like to have removed. Please note that we cannot incorporate redactions for any other reasons.**

RESPONSE: We do not have any requests for information removal from our response letters.

*** CODE AVAILABILITY: All Communications Psychology manuscripts must include a section titled "Code Availability" at the end of the methods section. We require that the custom analysis code supporting your conclusions is made available in a publicly accessible repository at this stage; please choose a repository that generates a digital object identifier (DOI) for the code; the link to the repository and the DOI must be included in the Code Availability statement. Publication as Supplementary Information will not suffice.**

RESPONSE: We have made this update on page 23 in the manuscript, at the end of the main manuscript text, before the References section (per Editorial Request Table guidance): "The analyses herein were preregistered and the RStudio analysis code files are available in the Open Science Framework project repository, which is publicly accessible at the following link: <https://osf.io/cxpgk/> (identifier: DOI 10.17605/OSF.IO/CXPGK)."

*** DATA AVAILABILITY:**

RESPONSE: We have made this update on page 23 in the manuscript, at the end of the main manuscript text, before the References section (per Editorial Request Table guidance): "All study materials, anonymized raw participant survey data, and preregistrations are in the Open Science Framework project repository, which is publicly accessible at the following link: <https://osf.io/cxpgk/> (identifier: DOI 10.17605/OSF.IO/CXPGK)."

Best regards,

Jennifer Bellingtier

**Jennifer Bellingtier, PhD
Senior Editor
Communications Psychology**

**Patricia Lockwood, PhD
Editorial Board Member
Communications Psychology
orcid.org/0000-0001-7195-9559**

REVIEWER EXPERTISE:

**Reviewer #1 stereotypes, social psychology,
Reviewer #2 stereotypes, social psychology, intersectionality**

REVIEWERS' COMMENTS:

Reviewer #1 (Remarks to the Author):

**I was Reviewer #1 in the original submission.
The revised manuscript provides a very clear and robust set of results that are clearly described. I would like to applaud the authors for their efforts and their final outcome. Great job, I like this manuscript a lot, and will definitely cite it!**

RESPONSE: Thank you so much for your constructive feedback and continued review of the paper. We are pleased you found our revision satisfactory and our scientific contribution valuable!

**Two small issues that the authors might want to address before publication are"
(1) the "marginally significant" depiction should be corrected. Under NHST, results are either significant or insignificant. The manuscript could say that the descriptive direction is in line with, but it is not statistically significant.**

RESPONSE: Thank you. We have revised the two results which used the phrase "marginally significant" to better align with null hypothesis significance testing.

On page 10 in the manuscript, we revise a sentence by splitting the original sentence into two sentences and, in the second sentence, stating clearly that the exploratory regression with demographic covariates was not statistically significant:

- “This result held in an exploratory linear regression ($b = 0.25$, $SE = 0.13$, $p = .046$, $\eta^2_p < .01$, $95\% CI = [0.00, 1.00]$). In an exploratory linear regression holding participants’ age, race, gender, income, and educational attainment constant, we find similar directional results, though this result was not statistically significant ($b = 0.22$, $SE = 0.12$, $p = .073$, $\eta^2_p < .01$, $95\% CI = [0.00, 1.00]$).”

On page 12 in the manuscript, we updated the sentence to state directional similarity but no statistical significant for the linear regression controlling for participant demographics.

- “This finding was robust in a linear regression ($b = 39.72$, $SE = 15.91$, $p = .013$, $\eta^2_p < .01$, $95\% CI = [0.00, 1.00]$), and was directionally similar in a linear regression controlling for participants’ age, race, gender, income, and educational attainment, though this result did not show evidence of statistical significance ($b = 34.40$, $SE = 15.72$, $p = .029$, $\eta^2_p < .01$, $95\% CI = [0.00, 1.00]$).”

(2) The discussion talks about the malleability of responses from conservatives, but the data suggest that liberals vary their responses more than conservatives. Across studies, the range of the mean response of conservatives is half of that of liberals:

Conservatives: 4.69 (Study 4, incongruent minority) to 5.1 (Study 3, White, congruent)

Liberals: 5.3 (Study 2, minority, incongruent) to 4.21 (Study 4, congruent White)

It would be interesting to discuss this point.

RESPONSE: Thank you for raising this point. We revise the discussion as follows:

In the first paragraph on page 20 in the manuscript, we removed emphasis on conservatives and included mention of liberal malleability:

- “Importantly, our results show not just a consistency in attitudinal responses from liberals who exhibit stronger preference for incongruent portrayals than their conservative counterparts and, amongst themselves, significantly prefer incongruent over congruent portrayals (Studies 1- 4), but also a malleability in responses from both liberals and conservatives: liberals typically prefer incongruent portrayals and are willing to withhold favorable attitudes from congruent portrayals, while conservatives typically prefer congruent portrayals, but express higher attitudes towards incongruent portrayals than liberals under certain conditions (Studies 3 & 4).”

On page 20 in the manuscript, we revise this paragraph in a few ways. First, we removed “more” sensitive/malleable when describing conservatives’ attitudes, but retain this mention of conservatives because it is an important future direction to consider. Second, we incorporate mention of liberals’ flexibility in attitudes, and encourage future direction to consider this malleability as well. We also emphasize that the malleability from conservatives and liberals are different in nature:

- “Our findings along attitudes that liberals and conservatives are sensitive to the ways stereotype incongruency is portrayed and that their reactions are malleable under different conditions is worth noting and following up on. Both groups of participants’ attitudes change when portrayals feature models that they may inherently underscore or which contradict their different worldviews: conservatives are more reluctant to favor incongruent portrayals but show an openness towards endorsing incongruency when White models are featured, while liberals endorse portrayals of minority races and of incongruency but withhold this endorsement for solely White models. Future work will explore and test additional paradigm tests featuring socially dominant models, higher status models, or other depictions of political worldviews in various contexts.”

On page 21 in the manuscript, we retain our discussion of conservative malleability when model race is manipulated, as this is a particularly interesting result from our studies. Retaining this sentence about conservatives does not diminish the malleability of liberal responses, we think.

Further, on page 21 in the manuscript, we update one sentence to incorporate variability from liberals' responses:

- "Taken together, we show a significant and robust divergence between political ideologies, that the divergence in reactions is characterized by both consistency and variability among liberals and conservatives."

Finally, we revised the abstract to reduce emphasize on conservatives' variability in attitudes:

- "Representation in the media has become a polarizing issue dividing conservatives and liberals in the U.S. In four experiments ($N = 5,125$), we find that stereotype portrayal elicits divergent attitudinal, economic, and behavioral reactions from liberals and conservatives. Notably, these reactions differ when portrayals feature racial minority (Study 1, $n = 958$ & Study 2, $n = 900$) versus White models (Study 3, $n = 783$ & Study 4, $n = 2,484$). Our findings demonstrate consistent divergence in responses to stereotype congruent versus incongruent portrayals between liberals and conservatives, although the direction and magnitude of differences vary. Liberals and conservatives display both variability and consistency in their divergent evaluations: liberals endorse portrayals of minority races and of incongruency but withhold this endorsement for solely White models, whereas conservatives typically prefer congruent portrayals, but show an openness towards incongruency when White models are featured. Understanding these dynamics is crucial for navigating the current sociopolitical landscape, especially in contexts where representations of race and gender identities are contentious."

Reviewer 2 confirmed they had reviewed the revisions.

**** Visit Nature Research's author and referees' website at www.nature.com/authors for information about policies, services and author benefits****

This email has been sent through the Springer Nature Tracking System NY-610A-NPG&MTS

Confidentiality Statement:

This e-mail is confidential and subject to copyright. Any unauthorised use or disclosure of its contents is prohibited. If you have received this email in error please notify our Manuscript Tracking System Helpdesk team at <http://platformsupport.nature.com> .

Details of the confidentiality and pre-publicity policy may be found here <http://www.nature.com/authors/policies/confidentiality.html>